



# Differential InSAR for tide modelling in Antarctic ice-shelf grounding zones

Christian T. Wild[1], Oliver J. Marsh[1,2], and Wolfgang Rack[1]

[1]Gateway Antarctica, University of Canterbury, Private Bag 4800, Christchurch 8140, New Zealand
[2]British Antarctic Survey, High Cross, Madingley Road, Cambridge, CB3 0ET, United Kingdom

**Correspondence:** Christian T. Wild (christian.wild@pg.canterbury.ac.nz)

**Abstract.** Differential interferometric synthetic aperture radar (DInSAR) is an essential tool for detecting ice-sheet motion near Antarctica's oceanic margin. These space-borne measurements have been used extensively in the past to map the location and retreat of ice-shelf grounding lines as an indicator for the onset of marine ice-sheet instability and to calculate the mass balance of ice-sheets and individual catchments. The main difficulty in interpreting DInSAR is that images originate from a combination of several SAR images and do not indicate instantaneous ice deflection at the time of satellite data acquisition. Here, we combine the sub-centimetre accuracy and spatial benefits of DInSAR with the temporal benefits of tide models to infer the spatiotemporal dynamics of ice-ocean interaction during the times of satellite overpasses. We demonstrate the potential of this synergy with TerraSAR-X data from the almost stagnant Southern McMurdo Ice Shelf. We then validate our algorithm with GPS data from the fast-flowing Darwin Glacier, draining the Antarctic Plateau through the Transantarctic Mountains into the Ross Sea. We are able to match DInSAR to 0.84 mm; generally improve traditional tide models by up to -39% from 10.8 cm to 6.7 cm RMSE against GPS data from areas where ice is in local hydrostatic equilibrium with the ocean; and up to -74% from 21.4 cm to 5.6 cm RMSE against GPS data in feature-rich coastal areas where contemporary tide-models are most inaccurate. Numerical modelling then reveals a Young's modulus of $E = 1.0$ GPa and an ice viscosity of 10 TPa s when finite-element simulations of tidal flexure are matched to 16 days of tiltmeter data; supporting the theory that strain dependent anisotropy may significantly decrease effective viscosity compared to isotropic polycrystalline ice on large spatial scales. Applications of our method range from (i) refining coarsly-gridded tide models to resolve small-scale features at the spatial resolution and vertical accuracy of SAR imagery, to (ii) separating elastic and viscoelastic contributions in the satellite derived flexure measurement and (iii) gaining information about large-scale ice heterogenity in Antarctic ice-shelf grounding zones, the missing key to improve current ice-sheet flow models. The reconstruction of the individual components forming DInSAR images has the potential to become a standard remote-sensing method in polar tide modelling. Unlocking the algorithm's full potential to answer multi-disciplinary research questions is desired and demands collaboration within the scientific community.

*Copyright statement.* TEXT





# 1 Introduction

The periodic rise and fall of the ocean's surface is caused by the gravitational interplay of the Earth-Moon-Sun system and our planet's rotation. Knowledge of ocean tides is fundamental to fully understand oceanic processes, sedimentation rates and behaviour of marine ecosystems. In Antarctica, the tidal oscillation also controls the motion of ice sheets near the coastline and
ocean mixing in the sub ice-shelf cavity which modifies heat transport to the ice-ocean interface (Padman et al., 2018).

SAR satellites repeatedly illuminate Earth's surface and record the backscattered radar wave. While the SAR signal's amplitude depends on reflection intensity and is mainly characterized by the surface, the recorded phase holds information about the distance travelled by the signal (Massom and Lubin, 2006). Two-pass interferometry (InSAR) can be used to determine surface motion with sub-centimetre accuracy over vast remote areas. Recently, InSAR has been applied to measure surface velocity of
floating ice (Tong et al., 2018) and to observe tidal strain of landfast sea ice (Han and Lee, 2018). In grounding zone areas, where an ice-sheet comes in contact with the ocean for the first time and forms floating glaciers and ice-shelves, InSAR has become the state-of-the-art practice to measure the flux divergence of ice-flow velocity (Mouginot et al., 2014; Han and Lee, 2015) and thus the mass balance of many ice-shelves around Antarctica (Rignot et al., 2013). InSAR can also be used to identify vertical deflection due to ocean tides. Horizontal motion and vertical motion cannot be distinguished at this stage but the
unsteady tidal contribution can be extracted by differencing two separate InSAR pairs that originate from a triple or quadruple combination of three or four SAR images. This assumes that gravitational flow due to steady ice creep is time-invariant, and that its phase contribution can therefore be removed. The double-differential measurement of vertical displacement only is known as Differential InSAR (DInSAR). While DInSAR has often been applied to detect the grounding line movement around Antarctica (Konrad et al., 2018) the signal can also be used to measure spatial variability of ocean tides at very high ground
resolution. This second application is complicated by the fact that DInSAR interferograms show a combination of multiple stages of the tidal oscillation. Tidal migration of the grounding line as well as viscoelastic time delays in ice displacement, tidally-induced velocity variations and geometric effects on the surface flexure also complicate the correct interpretation of DInSAR interferograms to date (Rack et al., 2017; Wild et al., 2018).

Present-day displacement measurements associated with interferometric phase suffer from two limitations: As the absolute
number of waves in the received SAR signal cannot be measured, the phase can only serve as a measure of relative distance change between two images. Phase is, by definition, expressed as the fraction of a full wave cycle that has elapsed relative to the origin with values between $0$-$2\pi$ in radians. Measurements of relative ground displacements between satellite overpasses therefore require smooth phase unwrapping, the most crucial processing step in DInSAR. Leaps between adjacent cells above $1\,\pi$, e.g. introduced by layover or discontinuities in the initial SAR images, can cause jumps in the unwrapped phase and may
therefore bias the continuous motion field.

Due to these complications, only very few studies have attempted to derive a tide model from DInSAR: Minchew et al. (2017) developed an unprecedented spatially and temporally dense SAR data acquisition campaign for the Rutford Ice Stream, Weddell Sea. Their novel Bayesian method to unequivocally separate a complete set of energetic tidal harmonics from nontidal ice-surface variability is unique, but beyond the data availability for the remainder of Antarctica. Baek and Shum (2011) failed





to develop a full tide model from the ERS-1/2 tandem mission, but succeeded in detecting the dominant tidal constituent in Sulzberger Bay, Ross Sea. In this case, too short a time span (71 days) eliminated a change of the observed tidal amplitudes as the repeat-pass cycle of the SAR satellites masks the sensor's sensitivity to tidal variability. In the Ross Sea, the tidal oscillation is dominated by diurnal harmonics. An accurate inversion of TerraSAR-X data with an exact integer number of repeat-passes

to a complete set of tidal constituents is therefore not possible from DInSAR measurements alone.

Tide models can be consulted to predict both the timing and magnitude of the dominant harmonics. Numerous tide models of various spatial scales (global vs regional) and complexity have been developed (see Stammer et al., 2014, for an overview). While forward models integrate the equations of fluid motion subjected to a gravitational forcing over time, inverse models assimilate measurements of vertical displacement from laser altimetry, tide gauges and GPS (Egbert and Erofeeva, 2002;

Padman et al., 2003, 2008). Since the modelled physics is generally simple and gravitational forces are well known, tide model predictions are of high quality in areas where ice is freely-floating on the ocean (error = $\pm 0.9$ cm, Stammer et al., 2014). In coastal areas, in turn, tide models are prone to inaccuracies due to errors in model bathymetry, grounding-line location and insufficient knowledge of the ice water drag coefficient (Padman et al., 2018). Another source of error arises from the conversion of ice-shelf draft to ice-shelf thickness and subsequent estimation of water-column-thickness. This freeboard

conversion assumes that ice near the coastline is in local hydrostatic equilibrium, whereas stresses from the grounded ice clearly prevent a freely-floating state. A bias of the hydrostatic solution towards thicker ice (Marsh et al., 2014), and therefore a thinning water-column-thickness, may negatively affect the tidal prediction. In summary, the relatively coarse spatial resolution and underlying assumptions of contemporary tide models introduce inaccuracies especially in feature-rich coastal areas such as fjord-type outlet glaciers. Although average tide model accuracy has improved markedly in coastal areas over one decade, from

about $\pm 10$ cm (Padman et al., 2002) to $\pm 6.5$ cm (Stammer et al., 2014), they are still a magnitude larger than the sub-centimetre accuracy of DInSAR (Rignot et al., 2011).

In this manuscript, we show how the spatial benefits and high accuracy of DInSAR can be used to refine coarse resolution tide models to adequately resolve ocean tides along the feature-rich Antarctic coastline. First we introduce the necessary data set, describe the preprocessing and guide through the work flow. Second we test the algorithm for the Southern McMurdo Ice

Shelf (SMIS), a small and almost stagnant ice shelf with a simple grounding-zone geometry, and expand the study to the Darwin Glacier, a relatively fast-flowing outlet glacier within a complex fjord-like embayment. We validate our results with dedicated field measurements taken within the Transantarctic Ice Deflection Experiment (TIDEx) in 2016. We then demonstrate how this exercise can also be applied to reveal errors in interferometric phase unwrapping and answer fundamental questions about the physical properties of ice in Antarctic glaciology.

## 2   Methodology

### 2.1   Summary of SAR image processing

To develop the method, we use 11-day repeat-pass TerraSAR-X data in StripMap imaging mode. The satellite acquires X-band radar data (wavelength 3.1 cm, frequency 9.6 GHz) with a ground resolution of slightly below 3x3 m and images covering an



area of 30x50 km. We calculate vertical surface displacement due to ocean tides using the Gamma software package (Werner et al., 2000). We then correct the resulting DInSAR interferograms for apparent vertical displacement due to horizontal surface motion (Rack et al., 2017) using the method presented in Wild et al. (2018).

## 2.2 Tide models

The predictions of five tide models are validated: the regional barotropic models (1) Circum-Antarctic Tidal Solution (CATS2008a_opt) developed by Padman et al. (2008), (2) Ross Sea Height-Based Tidal Inverse Model (Ross_Inv_2002) developed by Padman et al. (2003), (3) Ross Sea assimilation model (Ross_VMADCP_9cm), (4) the fully global barotropic assimilation model (TPXO7.2) from Oregon State University developed by Egbert and Erofeeva (2002), and (5) the (t_tide) prediction of GPS data from freely-floating areas following the harmonic analysis of Pawlowicz et al. (2002). The t_tide software is a widely-
used toolbox for performing classical harmonic analysis of ocean tides. It can analyse any time series record and outputs the amplitude and phase of its dominant harmonics, along with a tidal prediction that is free from non-tidal effects. The isostatic deformation of the Earth's lithosphere underneath the moving water masses is modelled using TPXO7.2 load tide model (Egbert and Erofeeva, 2002), which itself is based on 13 tidal constituents and added to all tide model predictions except t_tide. In addition to the tidal motion underneath the floating ice, much of the ice-surface variability can be attributed to the Inverse
Barometric Effect (IBE, Padman et al., 2003). A +1 hPa anomaly of atmospheric pressure translates to a -1 cm drop on the ice-shelf surface. To correct for the IBE, we make use of atmospheric pressure records obtained by nearby automatic weather stations on Ross Island (Scott Base AWS) and the Ross Ice Shelf (Marilyn AWS). We validate these records with separate barometric measurements taken within the TIDEx campaign and find very good agreement.

## 2.3 In-situ data

We set-up a number of GPS receivers to measure ice-surface motion at millimetre accuracy and high temporal resolution. Here we use GPS data from the freely-floating part of the ice surface and develop local tide models using t_tide. GPS measurements from within the tidal flexure regions are only used as validation data sets. All GPS measurements were differentially corrected using static base stations. We also install an array of seven tiltmeters recording surface flexure over 16 days across the grounding zone to confine the physical properties of Antarctic ice. This is complemented by a dense network of point measurements of
ice thickness using the new autonomous phase-sensitive radar echo sounder (ApRES, Nicholls et al., 2015).

## 2.4 Combining DInSAR and tide models

A tide model must perfectly predict the DInSAR observation in an area that can be expected to experience the full tidal forcing. We first adjust the tide model output to match the highly-accurate DInSAR measurements using a least sum of squares routine (Wild et al., 2018). Second we build on earlier work by Han and Lee (2014) and develop an empirical displacement
map showing tide-deflection ratio throughout the satellite image ($\alpha$-map). By feeding the $\alpha$-map with the adjusted tide model output, the 'point-forecast' is then spatially extended to predict the mean vertical displacement for every pixel at the times





of SAR data acquisition. We then perform the double-differences of the empirical model corresponding to the SAR image combinations used to generate the DInSAR images. The original DInSAR satellite measurements are subsequently removed from the mean DInSAR images to calculate their misfits, $\mu$. We now compute the least-squares solution to the equation $Ax = b$ such that the 2-norm $|b - Ax|$ is minimized. Here, $A$ is the $m \times n$ DInSAR matrix of SAR image combinations with $m$ rows of

SAR images and $n$ columns of coherent DInSAR interferograms; $b$ is a vector of $\alpha$-prediction misfits and $x$ the least-squares solution of this over-determined system. The values of $x$ correspond to how much an $\alpha$-prediction deviates from the 'real' vertical displacement at the times of SAR data acquisition. We therefore subtract these offsets, $\theta$, from the $\alpha$-prediction maps.

   We demonstrate the workflow in one spatial dimension with an example of the Southern McMurdo Ice Shelf ($78°15'$ S, $167°7'$ E, SMIS). In this study area, we derived 9 DInSAR images from 12 TerraSAR-X scenes in 2014 (Rack et al., 2017). We choose

a pixel on the freely-floating end of a profile through the ice-shelf grounding zone to represent the unrestricted ice shelf movement and calculate the percentage vertical displacement of every other pixel from this location. Averaged over the 9 DInSAR interferograms this pixel retains 100% vertical displacement (red areas in Fig. 1), while grounded areas experience zero net uplift (purple areas). Individual pixels on the freely-floating part of the SMIS may show $\alpha$-values slightly above 100%.

   We now extract the $\alpha$-values along the profile from the $\alpha$-map. This $\alpha$-profile can be multiplied with the individual DIn-

SAR measurements on its freely-floating end which results in empirically derived $\alpha$-predictions (Fig. 2 center). These mean predictions do not perfectly replicate the DInSAR measurements (Fig. 2 top). Their misfits, however, show a very systematic pattern (Fig. 2 bottom). It is desirable to find a combination of offsets that have the least deviation from the $\alpha$-predictions. We therefore hypothesize that this rather systematic signal can be reconstructed using a least-squares strategy. Here, the linear system is under-determined with 9 DInSAR equations and 12 unknown SAR offsets. We solve this system simultaneously by

finding the combination of offsets that result in a minimal sum of squares. The reconstructed offsets must then be removed from the $\alpha$-prediction for the times of SAR data acquisition (Fig. 3 top). The computed least-square offsets generally replicate the pattern of the misfits (Fig. 3 center) and result in smooth displacement profiles in the ice-shelf grounding zone (Fig. 3).

## 3   Results

In this section we apply the workflow in two spatial dimensions to the Darwin Glacier ($79°53'$ S, $159°00'$ E). A dedicated

field campaign was conducted in its grounding zone in 2016 and *in-situ* data is available. In contrast to the simple geometry at the SMIS, the Darwin Glacier consists of a feature-rich embayment that is constrained by steep topography at its margins. Additionally a buttressing ice rise to the Ross Ice Shelf restricts outflow in the North.

### 3.1   Reconstruction of displacement maps during satellite overpasses

From the interferogram dataset we identify a corridor of only about 2 km width along the centerline where the glacier can be

assumed to be freely floating (Fig. 1). This area is expected to experience the full oscillation predicted from tide models. We run five tide models to predict the tidal oscillation at the GPS station 'Shirase' over the time-span of SAR data acquisitions. Here we use atmospheric pressure data from the automatic weather station 'Marilyn' which is located about 120 km away on

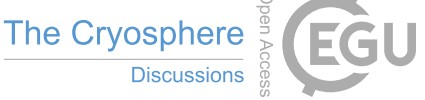



the Ross Ice Shelf to correct for the inverse barometric effect. This record correlates well (Pearson's correlator 0.989) with a mean of seven barometers installed over 14 days across the Darwin Glacier during the TIDEx campaign. All tide-model predictions show a clear fortnightly occuring spring-neap tidal cycle which is superimposed by a dominant diurnal signal (Fig. 4). The approximately fortnightly occurring spring/neap tides are largely owed to the difference in wavelength between the K1 and O1 constituents. The K1 tide has a period of 23.93 hours and is the dominant tidal constituent in the Ross Sea. (Padman et al., 2018).

We apply t_tide to our 16 day record of the 'Shirase' GPS to test the potential of short-term GPS surveys to improve current Antarctic tide-models. While the analysis captures the dominant diurnal constituents, fortnightly harmonics could not be retrieved adequately from this time series alone. The t_tide prediction is therefore the least accurate tide model and requires the largest adjustment to match DInSAR (Tab. 1). Although all the corrected tide model outputs now replicate our DInSAR measurements, their rate of tidal change is affected by the adjustment. Offsets computed for the Ross_Inv_2002 tide model are generally below 10 cm, whereas other tide models require adjustments of up to 13.3 cm (Tab. 1). This agrees well with the findings of Han et al. (2013), who find that the Ross_Inv_2002 model is the optimum tide model for the Terra Nova Bay with a 4.1 cm RMSE against 11 days of tide gauge data. We therefore choose Ross_Inv_2002 for numerical modelling purposes to minimize any effects on a viscoelastic model, but use TPXO7.2 to reconstruct vertical displacement at the times of satellite overpasses as it fits best our GPS measurements. We refer to the Appendix for a validation of individual tide model output with GPS data from 'Shirase' (Fig. A1).

After the adjustment, modelled tidal amplitudes range from -0.966 m to 0.781 m over the whole SAR period (Fig. 4). Mean residual error to the 45 DInSAR measurements at the tide-model location 'Shirase' is just 0.84 mm, which is within interferogram noise. We attribute this accuracy to the exceptionally high phase coherence of the TerraSAR-X data set. The reconstruction algorithm results in 12 smooth vertical displacement maps for the times of SAR data acquisition (Fig. 5).

### 3.2 Validation with GPS measurements

We now validate these reconstructions with available field data. As both GPS records overlap with the acquisition of SAR image 11, we extract the vertical displacement along the glacier's centerline and plot the profiles against the two GPS point measurements. The GPS measurement at 'Hillary' is 0.169 m, which is close to the reconstruction of 0.156 m. The 'Shirase' GPS measurement is 0.566 m, which is slightly above the reconstruction of 0.522 m. We attribute the deviations of 1.3 and 4.4 cm, respectively, to a combination of interpolation artefacts, temporal smoothing of the GPS data and residual errors of the least-squares algorithm. The overall shape of the vertical displacement is well reproduced as observed with both GPS measurements (Fig. 6).





### 3.3 Applications

#### 3.3.1 Tide-model refinement

A map of tide-deflection ratio ($\alpha$-map) can be combined with the tide model to predict an average time-series of vertical displacement between the times of SAR image acquisition. With this approach, the coarse grid of tide models is refined to resolve small-scale features of vertical tidal displacement throughout the embayment. The $\alpha$-value for the pixel containing the 'Hillary' GPS station is 46.06%. We use this value and linearly scale the tide-model output for the location of the 'Shirase' GPS to predict vertical tidal motion within the flexure zone. This scaling maintains the tide model's high correlation (Pearson's correlator 0.95) with the 'Shirase' record, but improves the RMSE between the TPXO7.2 output and the 'Hillary' record from 21.4 cm to 5.6 cm, which corresponds to an improvement of -74% (Fig. 7).

#### 3.3.2 Ice heterogenity

With the twelve reconstructed displacement maps at hand, it is now possible to perform any image combination. We mosaic the 45 double differences corresponding to DInSAR combinations and then calculate again the misfit between each modelled and observed interferogram for every pixel. The standard deviation of these misfits is shown in Fig. 8, with the majority of the glacier surface below noise level of interferograms ($\sigma < 1.0$ cm). We identify a narrow band with higher standard deviations ($\sigma \approx 2.0$ cm) from the inner shear margin of the Darwin Glacier extending along flow-direction onto its ice shelf. Standard deviations are largest out on the Ross Ice Shelf ($\sigma > 4.0$ cm) and a result of poor coherence between SAR images in this area.

#### 3.3.3 Detection of errors in phase unwrapping

We now apply the algorithm to the SMIS data set and calculate misfits of 9 DInSAR interferograms. Resulting standard deviations are generally smaller in this area ($\sigma < 0.3$ cm) and smoothly distributed throughout the map. We identify two regions of jumps between adjacent cells at the SMIS. Both extend from the center of an ice rise towards the dry land (cyan and green areas in Fig. 8) with $\sigma \approx 0.4$ cm and $\sigma \approx 0.5$ cm, respectively. We interpret these rapid increases as a proxy for errors in the DInSAR measurements, as the modelled least-square interferograms originate from a curvature-minimizing polynomial interpolation. We re-evaluate the remote-sensing part of the analysis and find discontinuities in DInSAR interferograms ID:1 and ID:8 that match the course of the two jumps in the standard deviation map. These discontinuities, in turn, are a result of using a minimum cost-flow algorithm on a triangular network for unwrapping interferometric phase differences to relative surface displacement.

### 3.4 Finite-element modelling of viscoelasticity

We hypothesize that any non-linear signal due to viscoelastic ice properties is significantly reduced or even completely lost during the averaging step to compute the $\alpha$-map. This signal can then be reconstructed by finding the offsets to match observations made with DInSAR. We therefore subtract the $\alpha$-prediction again from the 12 reconstructions to extract the theorised



viscoelastic signal (Fig. 9). This signal is negligible at times during neap tide (SAR 4 and 9) but well pronounced for SAR images acquired during spring-tide periods (SAR 1,6,7,10 and 11).

In order to further explore this pattern, we now make use of the tiltmeter array and ApRES network of ice-thickness measurements at the Darwin Glacier (Fig. 10). We match the numerical solutions from two finite-element models to seven tiltmeter

records, with the goal to derive information on the physical properties of Antarctic ice. Thereby, the Young's modulus, $E$, is a measure of ice stiffness and controls the width of the flexure region. The value for ice viscosity, $\nu$, influences the timing of the flexural response within the grounding zone (Wild et al., 2017). Two numerical models of ice-shelf flexure are employed. The elastic approximation (Holdsworth, 1969; Vaughan, 1995; Schmeltz et al., 2002) as formulated by Walker et al. (2013):

$$kw + \nabla^2 (D\nabla^2 w) = q, \tag{1}$$

where $w(t)$ is the time-dependant vertical deflection of the neutral layer in a plate, $\nabla^2$ is the Laplace operator in 2-D space and $k = 5 \text{ MPa m}^{-1}$ a spring constant of the foundation which is zero for the floating part. The applied tidal force $q(t)$ is defined by:

$$q = \rho_{sw} g [A(t) - w], \tag{2}$$

with $g = 9.81 \text{ m s}^{-2}$ the gravitational acceleration and $A(t)$ the time-dependant tidal amplitude given by the adjusted Ross_Inv_2002

tide model. We choose this model, in contrast to the TPXO7.2 model, for finite-element simulations to minimize any potential effects of tide-model adjustment on viscoelasticity (Tab. 1). The stiffness of the ice shelf is given by (Love, 1906, p. 443):

$$D = \frac{EH^3}{12(1-\lambda^2)}, \tag{3}$$

where $E$ is the Young's modulus for ice, $H(x,y)$ our ice thickness map derived from ApRES point measurements and $\lambda = 0.4$ the Poisson's ratio. We compare the elastic model with the viscoelastic approach developed by Walker et al. (2013):

$$\frac{\partial}{\partial t} \left[ kw + \nabla^2 \left( D\nabla^2 w \right) \right] + \frac{Ek}{2\nu(1-\lambda^2)} w = \frac{\partial}{\partial t} q + \frac{E}{2\nu(1-\lambda^2)} q, \tag{4}$$

where $\nu$ is ice viscosity. The following boundary conditions are applied for both models: the upstream boundary of the model domains on the grounded portion are anchored rigidly ($w = 0, \nabla^2 w = 0$), the downstream boundaries on the freely-floating ice shelf are set free. The location of the tide model computation is constrained to be equal to the tidal oscillation ($w = A(t), \nabla w = 0$) and the grounding line is represented by a fulcrum ($w = 0$). Both models are implemented in two spatial dimensions to

capture effects of complex grounding-line configuration on ice-shelf flexure (Wild et al., 2018). We then solve the models using the commercial finite-element software COMSOL Multiphysics. As tiltmeters measure slope, $w'$, along their longitudinal axis, we derive the models' solutions for vertical displacement, $w$, with respect to the $x$ and $y$ directions. This allows us to retrieve surface slopes components in easting and northing direction and to rotate them into the individual orientations of the tiltmeter sensors. With our 16 day tiltmeter records it is only possible to capture their diurnal and semi-diurnal components

with confidence. Fortnightly and monthly harmonics have been removed from the tiltmeter time series and we focus further analysis only on the K1 component. Therefore, we now make extensive use of the t_tide program to automatically extract the





modelled K1 harmonics from the modelled surface slopes and compare them against the K1 constituents from the tiltmeters. A Young's modulus of $E = 1.0$ GPa and an ice viscosity value of $\nu = 10$ TPa s fits best our measurements (Fig. 11). The viscoelastic model gives an average RMSE of 0.00118845 $^\circ$ to the seven tiltmeters and improves on the elastic approximation with an average RMSE of 0.00147136 $^\circ$ by $\approx -20\%$.

## 4  Discussion

### 4.1  Seasonal bias in $\alpha$-map

Due to the alignment of the satellite overpasses with the dominant diurnal tidal constituents in the Ross Sea, the observed stage of the tidal oscillation varies only slowly throughout the year. In the austral winter months, SAR images are acquired during stages of low tide, whereas satellite overpasses concur with stages of high tide during the austral summer months. The first 8
snapshots of our TerraSAR-X data set for the Darwin Glacier show conditions at low tide and only the last 4 are acquired during high tide. Our $\alpha$-map, in turn, ignores this seasonality and may therefore have a low-tide bias. As a result, the contribution of a tide induced landward migration of the grounding line may be affected by the averaging process. The seasonal bias would then modify the scaling of the tide-model within the flexure zone. This theory is supported by the finding that low-tide stages in the 'Hillary' GPS record are matched closely by the scaled tide model, but peaks during high-tide stages are still over estimated
(Fig. 4)

### 4.2  Viscoelasticity between snapshots

Similarly, the linear scaling using an $\alpha$-map only modifies the predicted tidal amplitude, but neglects a viscoelastic time delay in the flexural response towards the grounding line. Wild et al. (2017) found that viscosity is most pronounced in the diurnal tidal components. Harmonic analysis of our GPS records reveals that the diurnal K1 and O1 constituents at 'Hillary' are
20 lagging approximately 20 mins behind 'Shirase'. This signal is currently disregarded in the scaling work flow as ice is treated as a perfect elastic material that transfers tidal motion instantaneously in the flexure zone. This assumption, however, allowed us to improve the accuracy of the tide-model prediction by -74%. Currently, the viscoelastic signal can only be reconstructed for the times of SAR data acquisition. Including viscoelasticity between times of satellite overpasses may therefore be only a small, but systematic, opportunity for further refinement. When separating the viscoelastic contribution from the reconstructed
maps of vertical displacement at times of satellite overpasses, we assume that an $\alpha$-prediction corresponds to an instantaneous elastic response. This is justified by viscoelasticity being most pronounced when rates of tidal change are maximal. SAR images acquired during periods of spring tides at the Darwin Glacier show a significant viscoelastic contribution that diminishes during neap tide periods.

When predicting rates of tidal change using the adjusted Ross_Inv_2002 tide model, we identify a threshold of $\dot{A} \approx$
$\pm 0.05$ m h$^{-1}$ (SAR times 1, 2, 7, 8, 10, 11 in Tab. 1) above which viscoelasticity is well represented in the reconstructed vertical displacement maps (panels 1, 6, 7, 8, 10 and 11 in Fig. 9). Image 6 is thereby an exception, as the used Ross_Inv_2002



tide model was adjusted largely (-0.082 m), which affects the viscoelastic model. These independent observations support our suggested threshold of $\pm 0.05$ m h$^{-1}$ for the separation of elastic and viscoelastic signals, as derived from tiltmeter data on the Southern McMurdo Ice Shelf presented in an earlier study (Fig. 8 in Wild et al., 2017). The advantage of separating the elastic from the viscoelastic contribution to the tidal flexure pattern is the large potential for improving current inverse mod-

5 elling techniques to determine grounding-zone ice thickness from DInSAR measurements alone. Hereby, an elastic model is currently employed to optimize grounding-zone ice thickness to match the surface flexure from DInSAR. This is because an elastic model for tidal flexure is only forced by the 'apparent' tidal amplitude (Eq. 1), which can be measured directly from the interferogram on the freely-floating area. A viscoelastic model additionally incorporates the time derivative of the tidal forcing (Eq. 4) and hence captures the rate of tidal change. This information, in turn, can not be deduced directly from the

10 interferogram which makes the usage of auxiliary tide models inevitable. Tide models, however, have shown to be prone to large inaccuracies around Antarctica making a successful inversion of viscoelastic flexure models highly elusive. The applicability of an elastic model varies from location to location as effective viscosity is dependant on ice temperature and shear stress (Marsh et al., 2014). Our method to separate the two contributions to the flexure pattern may therefore help to remove the viscoelastic contamination and allow purely elastic inverse modelling. Furthermore, the threshold of $\pm 0.05$ m h$^{-1}$ is inval-

15 ueable to determine which satellite data acquisitions should be used for this calculation. This analysis, however, goes beyond the scope of this manuscript and will be published elsewhere.

### 4.3 Large-scale ice anisotropy

Fast-moving glacial environments like the Darwin Glacier are subject to large deformation by flow convergence and divergence, ice compression and extension, lateral shearing at the margins accompanied by fracture under tension and rapid thinning at

20 the ice-ocean interface. With cumulative deformation, a crystallographic fabric evolves that reflects the glacier's flow history (Alley, 1988), and with it strain-dependent mechanical anisotropy of ice. The standard deviation map, Fig. 8, shows a narrow band of larger misfits extending from the Darwin Glacier's inner shear margin out towards the freely-floating ice shelf. As preferred crystallographic orientation develops with strain, effective viscosity decreases of about a factor of ten compared to initially isotropic polycrystalline ice (Hudleston, 2015). Our analysis of tiltmeter data reveals a five-fold reduced viscosity at

25 the very dynamic Darwin Glacier compared to an earlier study at the almost stagnant Southern McMurdo Ice Shelf (Wild et al., 2017). We theorise that this microscopic process explains the macroscopic response observed here, and accounts for the measured glacial heterogenity within the embayment. Large scale observations of ice anisotropy, in turn, are currently the missing key to improve parametrisations to account for polar ice anisotropy in ice-sheet flow modelling (Gagliardini et al., 2009).

### 30  5   Conclusions and Outlook

Here we present the first data fusion of DInSAR with traditional Antarctic tide-modelling to predict spatial variability of tidal motion. The principal value of using DInSAR and tide models in tandem lies in the spatio-temporal benefits of resolving small



features over large regions. Their symbiosis not only improves current accuracies of the predicted tidal amplitudes in coastal regions generally, but also avoids issues related to the timing of the tidal wave and the sun-synchronous satellite orbit when attempting to derive tide-models from SAR data alone. The method presented in this paper improves traditional tide modelling in average by -22% from 11.8 cm to 9.3 cm RMSE against 16 days of GPS data. The GPS station 'Shirase' on the freely-

5 floating part of the Darwin Glacier has proven invaluable to determine which tide-model has to be used to best reconstruct the vertical displacement during satellite overpasses. For the Darwin Glacier, the TPXO7.2 tide model predicts best the tidal oscillation. With using DInSAR measurements to adjust the TPXO7.2 tidal prediction, its RMSE could be improved by -39% from 10.8 cm to 6.7 cm which exceeds the average tide model improvement of -35% within the last decade (Stammer et al., 2014).

Our GPS record from 'Shirase' is too short to develop a local tide model that improves already available Antarctic tide models. A longer record is required to adequately resolve a full set of tidal constituents. The GPS record from 'Hillary' could not be used for this purpose as it was recorded within the tidal flexure zone. Comparison of its measurements with predicted vertical displacement from feeding the $\alpha$-map with the adjusted TPXO7.2 tide model shows a -74% improvement over using the tide-model output alone. This independent validation supports the finding that DInSAR is very useful for refining tide models

in Antarctic grounding-zones. Accurate prediction of ocean tides in coastal areas is crucial as the majority of Antarctica's ice is discharged through large outlet glaciers.

Numerical modelling of ice dynamics in Antarctic grounding zones commonly assumes that ice is isotropic and homogeneous i.e. of same density and rheological properties throughout. Our analysis reveals that this assumption is valid for the Southern McMurdo Ice Shelf, an almost stagnant area with a simple grounding line, but invalid for the Darwin Glacier, a

20 fast-flowing outlet glacier with complex shear margins causing non-negligible ice heterogenity within the embayment.

Further work is required (1) to incorporate viscoelasticity to continue refining predictions of tidal motion between times of satellite overpasses, (2) to develop an automated method to monitor grounding-line migration due to ocean tides and (3) to perform inverse modelling of tidal elastic flexure to indirectly measure ice-thickness from SAR data.

*Code availability.* The code is freely available to the scientific community. Collaboration is anticipated and desired.

*Data availability.* TerraSAR-X data presented in this paper are subject to license agreements. GPS/tiltmeter and ApRES data are available upon request.

*Code and data availability.* No data sets, nor software, are part of this study.





*Sample availability.*  No samples were collected for this study.

## Appendix A:  GPS evaluation of tide models

The quality of the used tide-model to correctly reconstruct tidal displacement at the times of SAR data acquisitions, is also crucial to accurately predict spatial variability in tidal motion for all times. Here, we assume that a freely-floating area on the ice-shelf experiences the full oscillation as predicted from a tide model. In this area, however, tide-model output deviates from our DInSAR measurements. This indicates either that the area under investigation is prevented from a freely-floating state by lateral stresses within the embayment, or that the tide-model prediction is inaccurate for this area. We circumvent this ambiguity by making use of the high vertical accuracy of DInSAR and correct the tide-model prediction to match our satellite measurements. This raises the question of whether the adjustment improves or worsens the match to a 'real' tidal motion ? We therefore independently evaluate the pre- and post adjustment tide-model predictions and calculate their RMSE to 16 days of GPS data from the freely-floating area (Tab. A1). The adjustment improves all traditional tide model predictions by up to -39% for TPXO7.2, and only worsens the RMSE for the t_tide output by +11%, indicating that a harmonic analysis of GPS data can not be improved by using DInSAR for correction purposes. We choose TPXO7.2 for further processing as it displays the overall smallest RMSE (6.7 cm) and replicates the small-scale variability observed during the neap-tide period in the second half of our GPS record.

*Author contributions.*  All authors conceived the study and conducted fieldwork. CW developed the algorithm, performed the data analysis and wrote a first version of the paper. OM and WR processed the SAR images for the Darwin Glacier and Southern McMurdo Ice Shelf, respectively. All authors finalized and approved the manuscript.

*Competing interests.*  The authors declare that the research was conducted in the absence of any commercial or financial relationships that could be construed as a potential conflict of interest.

*Disclaimer.*

*Acknowledgements.*  We thank Antarctica New Zealand for logistical support and the Scott Base staff for their dedication to the Transantarctic Ice Deflection Experiment. D. Price, M. Ryan, D. Floricioiu and E. Scheffler also contributed to fieldwork. We acknowledge the German Aerospace Agency (DLR) for providing TerraSAR-X data through project HYD1421. Landsat-8 images courtesy of the US Geological Survey. AWS data from Scott Base is provided by NIWA, and for Marilyn station by AMRC, SSEC, UW-Madison. The authors enjoyed





fruitful discussions with H. Purdie and M. Sellier. CW also thanks R. Mueller and L. Padman for sharing oceanographic expertise. Reviewers and Editor later.



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



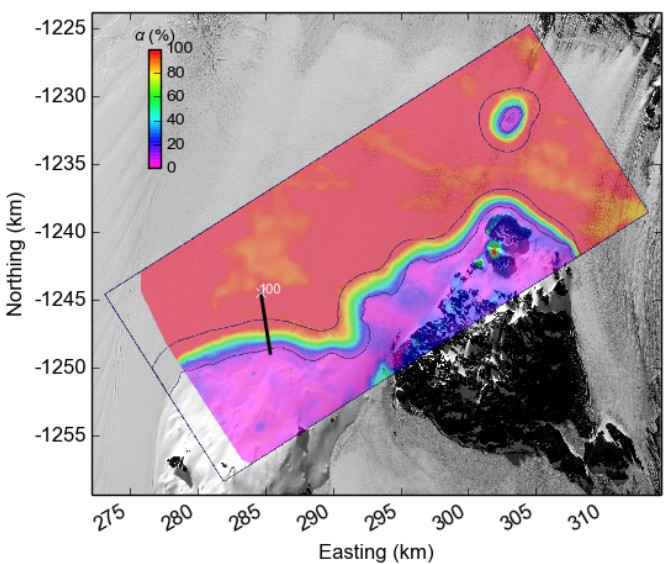

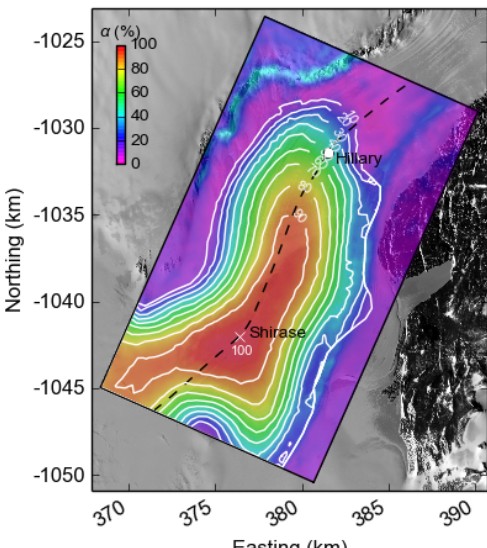

**Figure 1.** $\alpha$-maps of percentage vertical displacement due to ocean tides. Red colors highlight areas that can be assumed to be freely-floating. The white crosses show the tide-model locations that also serve as a common reference point across the images. The solid black line is the location of the profiles shown in Figures 2 and 3 on the Southern McMurdo IceShelf (left). The dashed black line shows the location of the profiles along the Darwin Glacier's centerline shown in Fig. 6 (right). The GPS station 'Shirase' and and 'Hillary' in the tidal-flexure zone. White contours delineate areas of constant vertical displacement. The map background is contrast-stretched Landsat 8 panchromatic imagery. The geographic projection is Antarctic Polar Stereographic with easting and northing coordinates shown in kilometers.





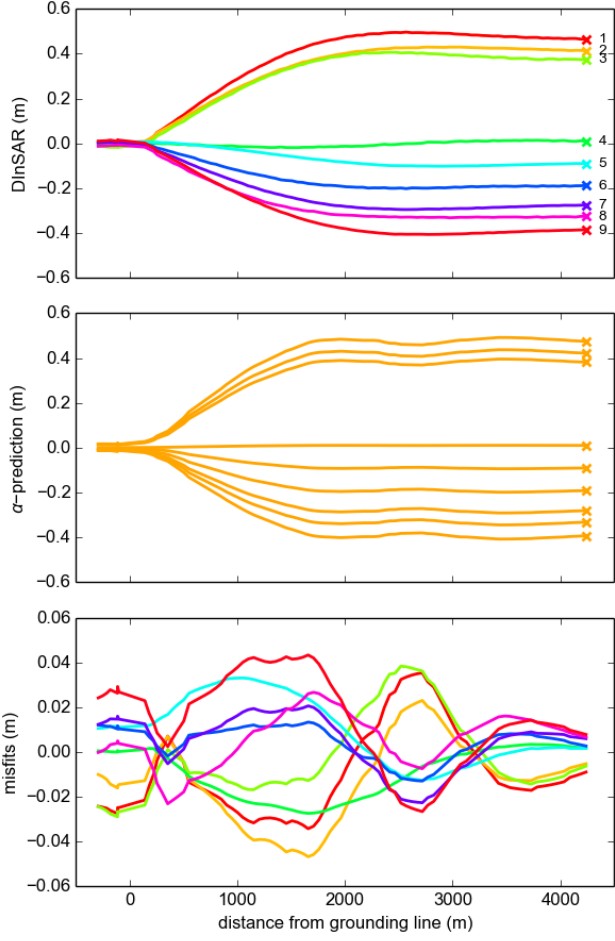

**Figure 2.** Vertical displacements along a profile through the grounding zone of the Southern McMurdo Ice Shelf, as (top) measured with 9 DInSAR interferograms, (center) predicted from an empirical displacement model ($\alpha$-map) and (bottom) their difference.



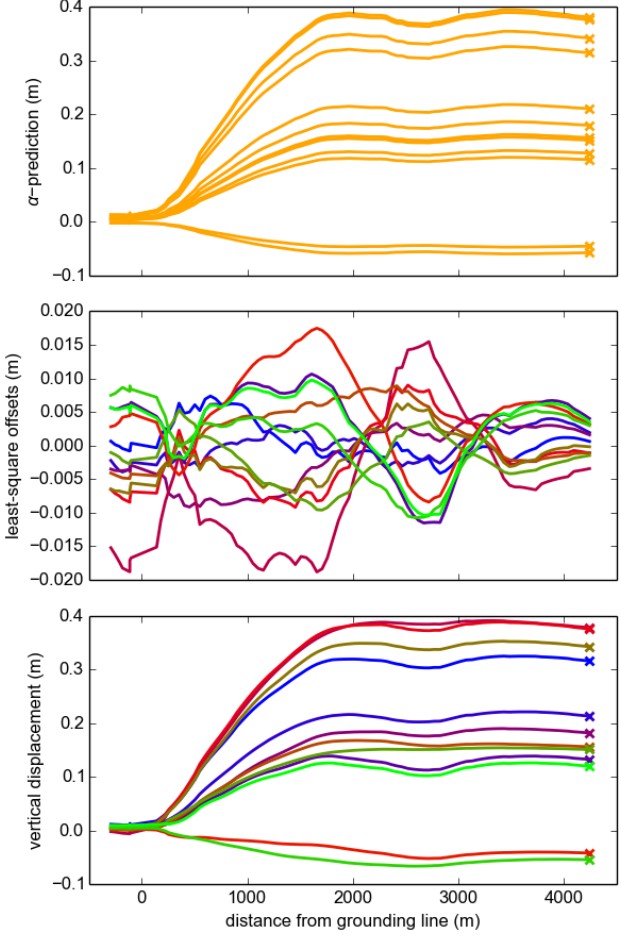

**Figure 3.** Reconstruction of vertical displacement along the profile during the 12 times of satellite overpasses on the Southern McMurdo Ice Shelf. (Top) a combination of an empirical displacement model with adjusted CATS tide-model output, (center) their least-square adjustment and (bottom) the final vertical displacement profiles during the times of SAR data acquisition.





**Figure 4.** The tidal oscillation at the Darwin Glacier as predicted by four tide models and a harmonic analysis of GPS data from the freely-floating area. The tide-model outputs are adjusted to match DInSAR observations using a least-squares fitting technique published in Wild et al. (2018). Black vertical lines coincide with times of SAR data acquisitions. Values for the prevailing tidal amplitudes and their adjustment at these times are given in Table 1. Gray shaded areas delineate the duration of the TIDEx campaign, when GPS data was acquired for validation (Figs. 7 and A1).



**Table 1.** SAR imagery used for the Darwin Glacier, least-squares adjustment ($\theta$ in m) for 5 tide models, tidal amplitude ($A$ in m) as predicted with the TPXO7.2 tide model and rate of tidal change ($\dot{A}$ in m h$^{-1}$) as predicted with the Ross_Inv_2002 tide model.

| SAR: | Date 13:57 (UTC) | $\theta_{\text{CATS}}$ | $\theta_{\text{RossInv}}$ | $\theta_{\text{Ross9cm}}$ | $\theta_{\text{TPXO7.2}}$ | $\theta_{\text{t\_tide}}$ | $A_{\text{TPXO7.2}}$ | $\dot{A}_{\text{RossInv}}$ |
|---|---|---|---|---|---|---|---|---|
| 1 | 25/05/16 | 0.109 | 0.098 | 0.133 | 0.101 | -0.004 | -0.341 | -0.059 |
| 2 | 05/06/16 | -0.061 | -0.066 | -0.097 | -0.039 | -0.030 | -0.666 | -0.057 |
| 3 | 16/06/16 | 0.029 | 0.035 | -0.050 | 0.034 | 0.013 | -0.409 | -0.007 |
| 4 | 27/06/16 | 0.032 | -0.012 | -0.049 | -0.009 | -0.111 | 0.002 | -0.022 |
| 5 | 08/07/16 | 0.054 | -0.022 | 0.107 | 0.008 | 0.122 | -0.271 | -0.037 |
| 6 | 19/07/16 | -0.086 | -0.082 | 0.035 | -0.088 | 0.206 | -0.661 | -0.005 |
| 7 | 30/07/16 | -0.091 | 0.015 | -0.026 | -0.045 | 0.074 | -0.687 | 0.080 |
| 8 | 10/08/16 | -0.078 | 0.001 | -0.035 | -0.040 | -0.080 | -0.276 | 0.072 |
| 9 | 26/10/16 | -0.073 | -0.023 | -0.053 | -0.046 | -0.244 | -0.132 | 0.011 |
| 10 | 06/11/16 | -0.044 | -0.009 | -0.088 | -0.023 | -0.172 | 0.087 | 0.096 |
| 11 | 17/11/16 | 0.099 | 0.025 | -0.002 | 0.084 | 0.059 | 0.522 | 0.052 |
| 12 | 28/11/16 | 0.109 | 0.041 | 0.124 | 0.062 | 0.168 | 0.398 | -0.029 |
| | mean absolute $\theta$ | 0.072 | 0.036 | 0.067 | 0.048 | 0.107 | - | - |



**Figure 5.** Reconstructed vertical displacement maps in the grounding-zone of the Darwin Glacier. The images show surface displacement due to ocean tides at the 12 times of SAR data acquisition. Dashed black lines along the glacier's centerline correspond to the profiles shown in Fig. 6. The white cross marks the tide-model location. The green triangle and dot in the lower center panel mark the locations of the two GPS stations 'Shirase' (freely-floating) and 'Hillary' (within the tidal flexure zone). Finite-element mesh in gray, mean course of the grounding line as determined from 45 DInSAR images in black. Note the ice rise in the bottom left corner. The map background is contrast-stretched Landsat 8 panchromatic imagery.



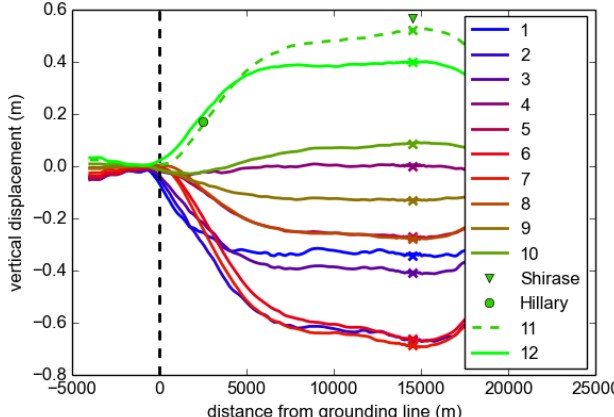

**Figure 6.** Profiles through the reconstructed maps of vertical displacement along the Darwin Glacier's centerline. The crosses mark the location where the adjusted tide-model output is applied to the $\alpha$-map. The green triangle and dot mark the locations of the two GPS stations 'Shirase' and 'Hillary' and are only used to validate the dashed green profile 11.



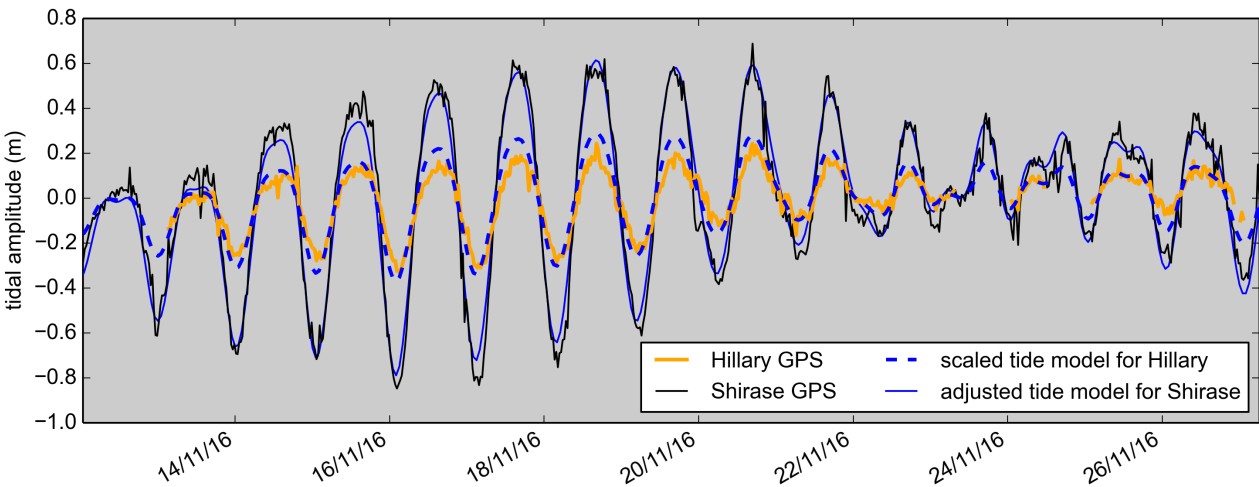

**Figure 7.** Time-series of vertical tidal displacement on the freely-floating part of the Darwin Glacier ('Shirase') and within the flexure zone close to the grounding line ('Hillary') . The (solid blue) corrected tide-model output for the 'Shirase' location is at first compared to (black) its corresponding GPS record. The (dashed blue) extended tide model is scaled and shows an empirical prediction for (orange) the GPS record at the flexure-zone station 'Hillary'. The length of the record corresponds to the gray-shaded area in Fig. 4.





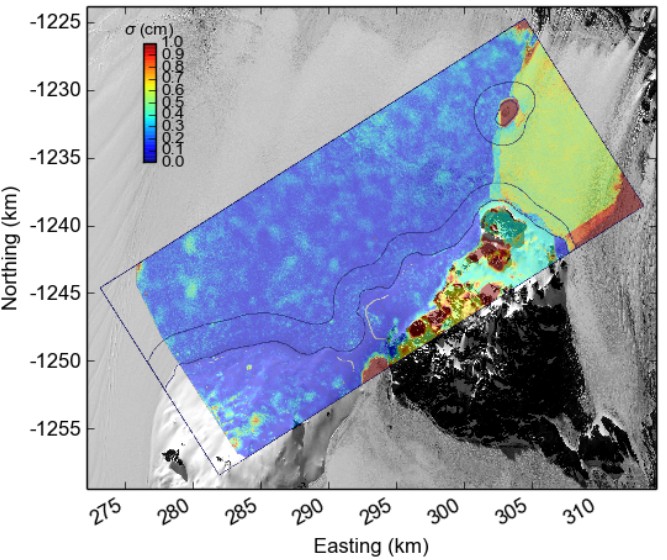
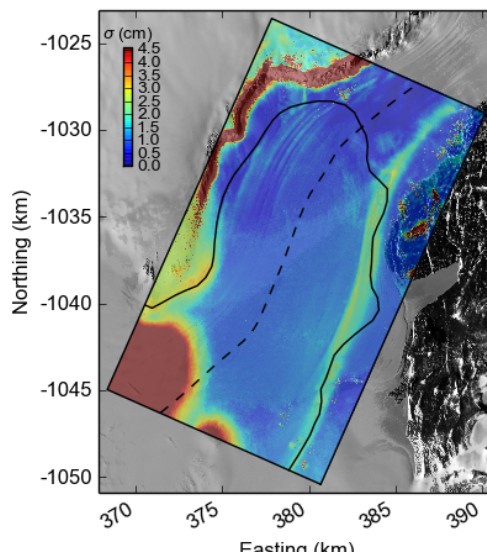

**Figure 8.** Standard deviations of misfits between modelled and observed DInSAR interferograms. Mean course of the grounding line as determined from DInSAR images in black. Note the ice rise in the upper right corner and two jumps in standard deviations between this ice rise and the dry land for the Southern McMurdo Ice Shelf. Also note the band of higher standard deviations from the Darwin Glacier's shear margin from Diamond Hill towards the ice rise in the bottom left corner and the high standard deviations. The map background is contrast-stretched Landsat 8 panchromatic imagery.





**Figure 9.** Spatial distribution of 12 least-square offsets that minimize the sum of misfits between 45 maps of $\alpha$-predictions and their corresponding DInSAR measurements. These offsets can be interpreted as the viscoelastic contribution to the reconstructed vertical tidal displacement at the times of SAR data acquisition. Dashed black line corresponds to the glacier's centerline, the solid black line shows the Darwin Glacier's mean grounding line as determined with DInSAR.





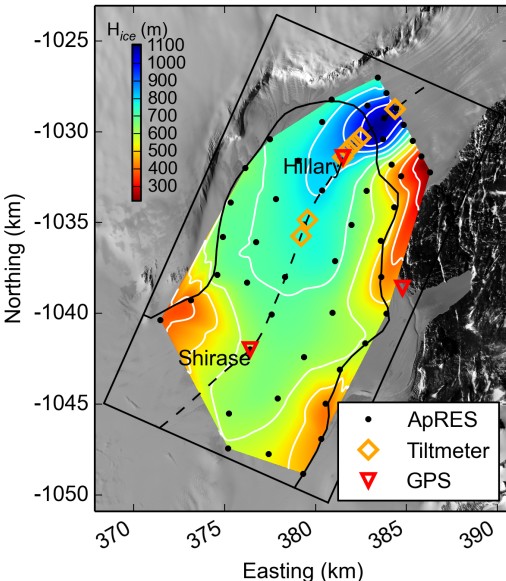

**Figure 10.** Measured ice-thickness map in the grounding-zone of the Darwin Glacier. Black dots show locations of high-precision ApRES measurements. Orange rectangles mark seven tiltmeter sensors that are orientated along the glacier's centerline. Red triangles show locations of GPS stations on the moving ice (Shirase and Hillary) and the location of the GPS base station on stagnant ice which is used for differential correction of the measurements. White contours correspond to a 100 m change in interpolated ice thickness. The map background is contrast-stretched Landsat 8 panchromatic imagery.





**Figure 11.** Surface flexure of the K1 tidal constituent along the Darwin Glacier's centerline as (orange) measured with an array of seven tiltmeters, (magenta) modelled using a viscoelastic rheology and (black) modelled with the elastic approximation. The orange dashed lines correspond to the uncertainty range of the K1 phases as determined from harmonic analysis of the individual tiltmeter records.



**Figure A1.** Validation of the tidal predictions of 5 tide models with a GPS record from the freely-floating 'Shirase' station. The tide-model outputs are adjusted to match DInSAR observations using a least-squares fitting technique published in Wild et al. (2018). Root-mean-square-errors before and after this adjustment are presented in Tab. A1. The length of the records corresponds to the gray-shaded area in Fig. 4.





**Table A1.** Root-mean-square-errors in m between tide-model output and GPS data from 'Shirase' before and after the adjustment to match DInSAR.

| Tide-model: | RMSE before: | RMSE after: |
|---|---|---|
| CATS2008a_opt | 0.117 | 0.087 |
| Ross_Inv_2002 | 0.112 | 0.091 |
| Ross_VMADCP_9cm | 0.135 | 0.127 |
| TPXO7.2 | 0.108 | 0.067 |
| mean | 0.118 | 0.093 |
| t_tide | 0.127 | 0.141 |