# Peer review of "Differential InSAR for tide modelling in Antarctic ice-shelf grounding zones"

_The Cryosphere, 2018_

## Referee Comment (RC1) · Anonymous Referee #1 · 27 Feb 2019

In this manuscript, the authors use differential SAR interferometry (DInSAR) to refine tide models on 2 test sites. By adjusting the DInSAR, they improve traditional tide models against the GPS data and can retrieve interesting information about the ice rheology. In my opinion, the methodology for both, remote sensing and model is solid, the results are interesting and the paper is well-written. I would recommend publication after correction of the minor comments below:

Minor comments:

In my opinion, the InSAR part should have a stronger description, not necessarily the technique that is already dealt with in Rack et al. 2017, but more regarding the number of acquisitions and how they were combined in time to form DInSAR. For example, Table 1 with the acquisition dates is not mentioned in the main text. Indeed, I am confused

with the number of SAR acquisitions and the number of DInSAR that were formed. Did you form 45 DInSAR observations from 12 SAR acquisitions? If so, this would indicate that you did not use consecutive acquisitions only, but tried every available combination. If it is the case, what is the advantage in using the additional combinations, that are not independent (from the consecutive pairs)?

Figure 5 shows reconstructed vertical displacement maps that look like interferograms, but they correspond to the tide displacement at the time of the acquisitions, not what we are observing through double differential interferometry. I believe that, at least, one illustration with direct comparison between DInSAR and reconstructed differential displacement would be useful for the reader. I also wonder why grounded portions of the glacier are moving in with tide, especially the speed slope on the top left corner. I would assume that the grounded portion should not move up and down with the tides.

3.3.2 Ice heterogeneity

As mentioned previously, as the grounded portion are vertically moving in your reconstruction, it translated in large standard deviations in Figure 8 on the top left corner where the steep slopes are found. I also notice large misfits (saturated in red) in the bottom left that are not mentioned in the text. Is the misfit due the absence of ice thickness measurements in this corner? Some comments would be welcome.

3.3.3 Detection of errors in phase unwrapping.

It is an interesting way of catching unwrapping issues. Would it make sense to include a new misfit map after the phase jumps are corrected?

3.4 Finite-element modelling of viscoelasticity

You mention that a Young's modulus of E=10GPa and an ice viscosity of 10 TPa s fits best your measurements. It would be nice to include uncertainties to appreciate how well these parameters are constrained.

4.3 Large-scale ice anisotropy

Results show that the shear margin of Darwin glacier are softer than surrounding ice (viscosity reduced by five-fold). The authors hypothesize that the change is due to ice anisotropy (from preferred crystallographic orientation). Some other hypothesizes have been proposed such as heating due to the important shear. It would be a nice if the authors could expand on this. Conclusions would remain unchanged but the origin of the softening would be different.
* * *

---

## Referee Comment (RC2) · Laurence Padman (Referee) · 25 Mar 2019

**GENERAL COMMENTS**

This paper describes the use of multiple DInSAR images to improve tide models in complex regions around Antarctica's coastline. The manuscript extends the work already presented in Wild et al. (2018; Frontiers in Earth Science), particularly by bringing in new data from GPS and tiltmeters on Darwin Glacier.

Overall, I found the work to be a valuable addition to discussions of tides in areas that are difficult to model, and the manuscript is mostly well organized and written. However, I don't think the authors are as clear about the problem(s) they are trying to solve as they need to be (see MAJOR COMMENT 1) and some areas of the text are too

complex, jargon-filled or reliant on prior studies to be easily followed (See SPECIFIC COMMENTS).

– Laurie Padman

MAJOR COMMENTS

1. The authors need to more clearly lay out sources of errors in tides, before they get to Results. Looking at the Abstract, p.1, l.11, for example, the impression is that the improvement in the "feature-rich coastal areas" is somehow in the tide models themselves, whereas it is mostly just because the DInSAR-derived flexure model is being used to scale the tides. Part of this is deciding what you mean by "tide model" vs "ice surface height response to tides". Maybe a clearer sequential description of the issue is needed. It might be that you decide just to be specific about what you mean by "tide model"... "In this paper, we use the term 'tide model' to refer to ..." But then, when you define improvements that are mostly associated with taking flexure into account, you still need to emphasise that the improvement is because of the inclusion of flexure, not a true improvement in a tide model.

2. It was not totally clear to me where in the processing chain the IBE is applied. It should probably be treated like a tide; therefore, added to the model-predicted tide before scaling to GPS data and/or DInSAR alpha-maps. However, the authors also need to know that the IBE is *not* simply -1 cm per +1 hPa: there is a dependence on time scales and, also, the ocean physics of how the water moves around under forcing by surface atmospheric pressure. These issues are discussed a bit by Padman et al. (2003), but also do some searching on more recent IBE issues; e.g., "MOG2D" and Rui Ponte's papers.

3. As far as I can tell, the method used to optimize the tide "model" relies on scaling the total tide prediction (by the alpha-map) to optimally fit a GPS or multiple DInSAR fields. That may, in fact, be the best that's possible when the GPS record is short or given the limits on DInSAR availability. However, it misses the strong possibility that different

errors are present in different tidal constituents. For example, in the Ross Sea, O1 and K1 are both large (but also see the next comment). If O1 in the tide model is accurate but K1 isn't, then you get a different expectation between DInSAR double-differenced values than you get by assuming errors are the same, in fractional terms.

4. (a) The authors appear to treat K1 from 15 days of GPS as a reasonable estimate, noting that the short record just means you can't resolve the fortnightly tides. However, K1 is close in frequency to P1 (6-month modulation period), and P1 has an amplitude of 15-20% of K1. On 15 days of data, t-tide just finds a "K1" that is really "K1+P1". So, over a 6-month period, the "K1" tide from analyses of short records can vary a lot (by 30-40% from minimum to maximum) without the tidal physics actually changing. You can reduce the error from this by using inference, where the inference parameters come from a longer GPS or maybe your favorite tide model. But then you'd also need to include P1 in the prediction scheme. (b) The same issue applies to semidiurnals S2 and K2. (c) Also note that models can differ systematically between diurnal and semidiurnal constituents; i.e., one model might have good diurnals and poor semidiurnals, while another has the opposite.

5. These issues may help explain why different models best suit your needs for the model scaling. E.g., perhaps the best model (for diurnal-dominated Ross Sea sites) for DInSAR analyses is one where O1, K1 and P1 amplitudes are all "wrong" but highly correlated, so that the alpha-map scaling approach works well, whereas an overall more accurate model (for correlating with GPS) has compensating errors between different constituents.

6. It wasn't clear what happens to true tide-model *phase* errors, which could cause as big an overall error in DInSAR fields as you get from amplitude errors.

7. I was surprised to see the Minchew et al. (2017) Rutford study get so little attention here (p.2, l.31-34). It's true they needed a dedicated SAR "mission" to get an empirical model out of it, so it can't yet be applied everywhere. But it'd be good to have more
information on what they learned, e.g., "we'd need >100 DInSAR fields", "accuracy of retrieved 'total' tide is less than we get from our approach", "even with 100's of DInSAR, can't empirically determine tidal harmonics S2, . . ."

SPECIFIC COMMENTS

The authors might be interested in Appendix B in Rignot et al. (2000, JGR-Oceans), where Doug MacAyeal was, I think, the first to lay out the idea of inverting InSAR to tidal constituents.

p.2, l.34 to p.3, l2: I don't like the structure, "Baek and Shum (2011) *failed* to . . ." Maybe invert the sentence, "Baek and Shum mapped the dominant tidal constituent (O1) in . . ., but data limitations prevented them from developing a full tidal model." Note that even getting O1 right is valuable; it indicates ways in which ocean tide models need to be changed, and these changes will filter into other constituents. Also, *no* method (even yours) will lead to a "full tidal model".

p.3, l.10-13: This misrepresents Stammer et al. (2014). The very small error (<1 cm) cited by Stammer et al. is for the deep ocean (nothing to do with ice), *under* the TOPEX/Poseidon and Jason satellite orbits, which were chosen specifically to resolve tides. (Padman et al., 2018) gives a summary.) The high accuracy comes mainly from assimilation of the high-quality T/P/J data sets, and only secondarily from the "physics" issues such as the long wavelength of tides in deep water and other issues like friction that are also more easily dealt with (or smaller) in deep water. Note that errors along the shallower margins feed into deep ocean tides in purely dynamic ("forward") model solutions so that, without accurate data to assimilate, deep-water tides will be much less accurate.

p.4, l.9: It's really unfair to cite Pawlowicz et al. for t_tide, but not to include "which is based on Foreman (1977)." Foreman did the harder part of the coding; to a significant extent, Pawlowicz mainly converted FORTRAN to Matlab code.

p.4, l.27: "A tide model *must perfectly predict* . . ." Why? Clearly they don't, even after tuning.

p.4, l.28-29: This cite to what Wild et al. (2018) did is too terse. At a minimum, at this stage we should know whether (i) the model is resolving individual tidal constituents, or just scaling a prediction of total tide, and (ii) whether just tide model amplitude is considered to contain errors (correlated between different constituents?) or whether possible phase errors can exist and be accounted for?

p.6, l.7-9: Need more details on t_tide analysis. Especially, tell us if you used inference to separate K1 and P1, and S2 and K2. See MAJOR COMMENT 4. An analysis of 15 days without inference usually predicts tides in the same 15-day window really well, but it cannot be used to extrapolate to other time periods.

p.6, l.25-28: As I read this, I didn't understand why the Hillary comparison was called "close to" while the Shirase one was "slightly above". Hillary is also "slightly above".

p.7, l.7-9: See MAJOR COMMENT 1; This improvement is large, and impressive, and important to know about. But the primary reason for the improvement is taking into account the DInSAR-measured flexure, not what I infer from "tide model improvement". It's a terminology issue, but confused here because you talk about "scal(ing) the tide-model output" which implies that the tide model really refers to the model without ice mechanics taken into account.

p.7, l.24-26: This is a jargon-rich statement that really doesn't tell me why discontinuities occur.

p.8, l.29 to p.9, l.1: See MAJOR COMMENT 4; You can't directly compare K1 from a long record (say, >6 months) with K1 from a short record, unless you used a reliable inference scheme to separate K1 from P1.

p.9, l.7-9: This statement about orbit alignment with high tide might be true; I need to check it. However, typically the long-term diurnal variability is semi-annual.

p.9, l.20-21: From this I assume that phase is not considered in any of the processing chain taking advantage of tide models to improve interpretation of DInSAR. That's okay, but it needs to be much more explicit in the "Methods".

p.9, l.26-28: This statement needs to be defended with the right figure. You kind-of get into it in the next paragraph, but it should be an easy figure of "error due to viscoelasticity" vs "A dot".

p.10, l.9-16: This, for me, is very dense text and hard to interpret for my own interests. Can it be simplified?

TECHNICAL CORRECTIONS

1. The authors often hyphenate words that don't need it; e.g., "ice-shelf" and "tide-model". This makes sense in compound expressions like "ice-shelf height" and "tide-model physics", although even there it isn't usually needed. But it is wrong if used as in "the ice shelf moves …"

2. My preference is for using past tense for anything that *was* done: "We measured …", "We analysed …" etc. Then present tense for things that are determined to be true, e.g., "The ice shelf flexes with the tides …"

3. Authors routinely misspell "heterogeneity" and related.

4. Use of "Theta" for a least-squares adjustment value is confusing to me, since Theta is often used for *phase*. Why not "delta-h" or something else more related to height? If you continue to use Theta, then give the units (m) where it appears, e.g., legend on lower panel of Fig.4 and in Fig.A1.

Fig.1 caption: (a) This isn't really the "% displacement due to ocean tides", which implies, to me at least, that (e.g.) "30%" would mean that 30% is due to tides, and the other 70% is not-tides. I think you mean that it is the fraction of the tide-model signal that appears in the flexurally-constrained ice surface elevation signal. (b) Sentence starting "The GPS station" is not a sentence. (also two adjacent 'and's)

Fig.2: Why not use same colouring for middle panel as for top and bottom panels?

Fig.3: Why not use same colouring for top panel as for lower two panels?

Fig.5: Font size is incredibly small! First, eliminate all repeated text. All panels use the same Easting and Northing limits, so you can delete most of those, and close up the gaps between panels.. Then, colour scales and bars are unreadable. But they are same, right (?) so only need to appear once, presumably *off to the side of* the matrix of panels. And reduce number of color labels on the color bar to create some space. "A" values should have a white background: black on mid-gray is fuzzy. Same for SAR-pass IDs (1-12). Dashed lines and symbols are hard to see. Maybe, like the colorbar, put labels outside the matrix and have arrows pointing in to the sites on one panel.

Fig.7: Make the legend just single column. Then, add the lines in the order they appear in the processing chain. So . . . "Shirase GPS", "Adjusted model for Shirase", "Scaled model for Hillary", "Hillary GPS".

Fig.8: (a) since you mention "Diamond Hill", point it out on the map. (b) Color bars: increase bar and text size; reduce number of color labels; provide a white background for the region of each panel devoted to the color bar.

Fig.9: All the same comments as for Fig.4.

(new figure request): See earlier comment, "p.9, l.26-28: This statement needs to be defended with the right figure. You kind-of get into it in the next paragraph, but it should be an easy figure of "error due to viscoelasticity" vs "A dot"."

---

## Author Comment (AC1) · 16 Jul 2019

**Reply to Anonymous Referee #1 on "Differential InSAR for tide modelling in Antarctic ice-shelf grounding zones"**

Summary:
The reviewer has largely understood the manuscript, picks up its main information and has generally good comments to improve the quality of the paper. The only misunderstanding seems to be that DInSAR is adjusted to match the tide model output (as indicated in the beginning of the reviewer's second sentence). With the relatively high vertical accuracy of DInSAR (<1cm) compared to tide models (approx 10cm) we consider DInSAR the absolute truth, and only adjust the tide-model output on the freely-floating part of the ice shelf to match DInSAR (we added this sentence to the introduction). The adjusted tide-model output is later scaled in the flexure zone using an alpha-map. We thank reviewer #1 for providing a very constructive feedback and the suggestion to include a Figure with a direct comparison between DInSAR images and reconstructed differential displacements in the paper, as well as the inspiration to include a Table dealing with a more detailed uncertainty analysis on the ice rheology values.

Minor comments:
1) DInSAR combinations
For the SMIS, 12 SAR images from three different satellite tracks were used to produce 9 DInSAR images. For the Darwin Glacier, 12 SAR images from one satellite track were available to produce a total of 45 DInSAR images (see Table). The combinations were generally chosen so that a later image is always subtracted from an earlier image. For image triples, the central image was taken as a common reference/master image. Additionally the data gap between SAR 8 and 9  at the Darwin Glacier was taken into account (no 8-9 combination as loss of coherence). The advantage of using every other remaining combination is that more double-differential measurements of tidal amplitude are available for the least-squares fitting algorithm than only using consecutive pairs alone. The system of linear equations is then overdetermined (instead of underdetermined). We have added these statements were appropiate in the main text and include a table of DInSAR combinations for the Darwin Glacier.

2) Figure 5
The reviewer is right that these images don't show wrapped interferometric phase and rather unwrapped vertical tidal displacement. The reason for displaying these maps with fringes is mainly to show the reader that the algorithm can reproduce complex flexural patterns within the grounding zone.  We included a selection of 3 measured versus modelled DInSAR images as proposed by the reviewer.
The reviewer is also right that one would expect no tidal change on the grounded parts (ie same color in all panels). As the signal to noise ratio is increasing drastically from the grounding line in upstream direction, the algorithm will systematically be biased by noise in the interferograms. Areas where no tidal signal can be expected (as in the top corner or on rocks) will therefore show variability where there is none in reality. We have included sentences where appropiate.

3) Ice heterogeneity
The red band of relatively high standard deviations in the top left corner of the Figure follows the course of rocky cliffs. In these areas, most DInSAR measurements lost coherence and voids were dominated by noise. Similarly,  the red area in the bottom left corner coincides with the shear zone with the fast-flowing Ross Ice shelf (the Byrd Glacier is adjacent). In this area a loss of coherence is also problematic. We comment on it in the revised paper.

4) Phase unwrapping
The purpose of Figure 9 is to show the reader the application of detecting unwrapping issues at the SMIS (jumps in the standard deviation) which we were able to avoid at the Darwin Glacier (smooth standard deviations on the floating part). As the main purpose of the paper is not on phase unwrapping strategies but rather on improving tide model output, we have decided against including a corrected standard deviation map of the SMIS as the paper is quite heavy on figures and tables already.

5) Finite-element modelling

[Figure]

*Illustration 1: Mean Root-mean-square-error to seven K1 harmonics as determined from harmonic analysis of tiltmeter data in the grounding zone of the Darwin Glacier. Only the Young's modulus (E) can be varied in an elastic model (black curve), the dots represent viscoelastic model performance with viscosity values corresponding to values in the legend. The smaller the mean RMSE the better the match of the model.*

The reviewer is right that uncertainties for the ice rheology should be provided. (I assume E=10 GPa is a typo and should have been E = 1 GPa). We have therefore performed a thorough uncertainty analysis, but point the reviewer/reader to Wild et al., 2017 (Journal of Glaciology) for a more detailed sensitivity analysis on varying the ice rheology. Uncertainties arise from the quality of the harmonic analysis of the individual tiltmeter records using t_tide. Both amplitude and phase of the K1 signals are determined within error bounds, which have been accounted for in the revised paper. We note that the main hypothesis coming out of the present paper is the reduction of ice viscosity in the shear zone and state that a finer resolution of the viscosity value (12.9, 13.0, 13.1,etc) has been chosen to tune the model than for the Young's modulus (0.5, 1.0, 1.5, etc). This is supported by the fact that including viscoelasticity in our model simulations generally reduces the mean RMSE to tiltmeter data more than changing the Young's modulus between 0.5 and 2.0 GPa. The uncertainty range for the Young's modulus and the viscosity value are calculated as the mean absolute deviation from the best E and best nu in Table 3.

6) Large-scale ice anisotropy
We have included a paragraph about other mechanisms that soften ice (damage, shear heating, tidal stresses) and note that none of them explains the spatial heterogeneity and differences between Darwin Glacier and SMIS that we observe here.

7) Other related changes to the manuscript
I have found an error in the calculation of the mean error of the adjusted tide models to all 45 DInSAR measurements. The originally stated error of 0.84 mm was calculated without taking the sign of the residual errors into account (a mean of values around zero will always be close to zero). For this reason, the mean absolute error was calculated and the error corrected to 7 mm (which is still within interferogram noise)

[Figure]

*Illustration 2: Mismatch between 45 DInSAR measurements of tidal surface displacement in the freely-floating part of the Darwin Glacier. Dots correspond to model predictions before the adjustment to DInSAR, the green crosses correspond to the residual errors after the adjustment. The green crosses average out to a mean absolute error of 7mm, which is now changed in the paper*

---

## Author Comment (AC2) · 16 Jul 2019

**Reply to Referee #2 (Laurence Padman) on "Differential InSAR for tide modelling in Antarctic ice-shelf grounding zones"**

Summary:
The reviewer is clearly an expert in tide modelling and provides invalueable insights in tidal dynamics, strengths and weaknesses of harmonic analysis as well as into the literature. The reviewer is right to point out that not the actual tide model physics are improved, but only (i) the tide model outputs are adjusted on the freely-floating part of the ice shelf by using DInSAR measurements (they are considered the absolute truth) and (ii) the adjusted tide model outputs are then scaled in the flexure zone by using 'the fraction of the tide-model signal that appears in the flexurally-constrained ice surface elevation signal' (aka alpha-map). The reviewer, however, seems to have misunderstood (1) the role of the two GPS records Shirase and Hillary in the analysis. Both of these records have only been used for validation purposes and don't feed into the tide adjustment/scaling algorithm. Similarly (2), it is true that t_tide only captures K1 and S2 as K1+P1 and S2+K2 without using inference, however, these harmonics have only been used within a short 16 day window and not in model simulations over an entire year (the reviewer agrees that 'an analysis without inference predicts tides within the window really well, but cannot be used to extrapolate'). Including inference in the analysis would be necessary if an entire year would have been modelled, but as only a relatively short window has been used to determine the rheology values, the errors in detecting the K1 amplitudes and phases from our 16 day tiltmeter records dominate the bias over the error due to inference (see reply to major comments below). We therefore focused our revisions on a more detailed uncertainty investigation of our harmonic analysis than including inference. We thank reviewer #2 for sharing his expertise in tidal dynamics and making us aware of using inference for analysing longer time series and leave the question on how to improve the actual tide model physics with an adjusted tide model output for future investigation.

Major comments:

1) Nomenclature : We extended the section on the sources of errors in tide models

2a) Inverse Barometric Effect (IBE) : The IBE is first calculated from barometric pressure records of nearby AWS and then added onto the raw tide model output. The result (IBE + tides + load tides) is then adjusted to match DInSAR measurements on the freely-floating part of the ice shelf. These adjusted tides are then validated with the independent GPS record Shirase. Afterwards, the adjusted tides are scaled using the alpha-map which is then validated with the GPS record Hillary. Note that both GPS records are solely for validation purposes. The alpha-map is a result of DInSAR alone.

2b) Dependency on time scales : We have identified that the most relevant control of the IBE is the window size over which to calculate a running mean of the barometric pressure. Our relatively long GPS record from the freely-floating part of the Southern McMurdo Ice Shelf was used to study the effects of changing the size of this window. First tides and load tides were removed from this GPS record to extract the IBE signal (called residual error). Second, the window length was varied linearly from 0 (instantaneous response) to 10 days (very delayed response) to calculate the IBE value in comparison to the residual error. Third, the RMSE is plotted as a function of the length of this window. Note that (i) the longer the length of the runing mean, the closer the IBE value is to zero; (ii) the shorter the length of the running mean, the smaller the RMSE with a best match if an instantaneous response is assumed:

[Figure]

*Illustration 1: Calculation of the best window size for the IBE: (Top panel) Comparison of the resiual GPS signal to two scaled barometric records (1 hPa = -1 cm). (Bottom left panel) Scatter plot of the pressure anomaly from a local barometer and the anomaly of the residual to calculate the IBE as a linear fit through the point cloud (points are color coded to point density). (Bottom right panel) The lenght of the averaging window to calculate the scatter plot on the left and the resulting RMSE to our measurements. Note that any deviation from an instantaneous response will increase the RMSE.*

3) Here might be the misunderstanding. The method used to optimize the tide model (tide + load + IBE) relies on adjusting it to best match multiple DInSAR measurements on the freely-floating part. It is independent from the GPS records and relies on SAR data alone (plus tide model, load model and barometer data as input). The method is therefore unable to detect individual tidal components and only adjusts the sum of the ones that are used in the underlying tide model. See related point 5.

4) Inference : The reason why we focused on the K1 component within the 16 day window in the first place was that t_tide didn't find other components with a sufficiently high signal-to-noise ratio for further analysis. Within this window, t_tide's estimate of K1 plus/minus its amplitude error covers the variability that is introduced by K1+P1 wihin these 16 days (see figure below, top panel). These uncertainties as well as the phase errors of t_tide's K1 components now feed into the paper when we determine the Young's modulus and viscosity values. We note that using inference would be necessary if t_tide's K1 components would have been extrapolated to times when the K1+P1 inference is larger than K1's amplitude error (f.e. April and October 2016). Similarly, t_tide's S2 component shows large uncertainties. The range of its S2 amplitude error is much greater than the annual variability in S2+K2. For these reasons, the tuning of the model rheology focuses solely on t_tide's K1 components (now within respective amplitude and phase errors).

[Figure]

5) Correlation :

We weren't aware of the fact that tide models differ systematically between diurnal and semi-diurnal constituents. It would be very much appreciated if reviewer #2 could elaborate on which ones have generally good diurnals and poor semi-diurnals (and vice versa) and if this error is more likely to be in the amplitude or in the phase of the components. We agree with reviewer #2's hypothesis that this might explain why different tide models fit better to our diurnal-dominated GPS records, and note that Ross_VMADCP_9cm and Ross_Inv_2002 seem to lead the phase of the Shirase GPS record (Figure A1), while CATS2008a_opt and TPXO7.2 are in phase. From our experience in the Ross Sea region (Southern McMurdo Ice Shelf, Darwin and Beardmore Glaciers) we have learned that CATS2008a_opt is generally good in terms of phases, but underestimates amplitudes particularly during spring tides.

6) Phase errors :

It is true that phase errors in the tide model also affect the error to DInSAR fields. Is there a way to retrieve phase (and amplitude) errors from f.e. the CATS2008a_opt tide model, similar to the error bounds that t_tide provides ? We think that phase errors, and thus the timing of the tidal forcing, mostly affect the rate of tidal change rather than the absolute amplitude. An inaccurate phase would therefore influence the determined viscosity value. To minimize this uncertainty, we have selected the tide model with the best phase (Ross_Inv_2002) for the flexure modelling part of the paper and use the model with the best amplitudes (TPXO7.2) to reconstruct vertical surface displacements at the times of satellite overpasses (Section 3.1, 2[nd] paragraph). We hypothesize that our adjusted tide-model output can be used in the future to improve the actual tide-model physics (in terms of amplitudes and phases of individual components) by assimilation (similar to the reviewer's explanation of deep ocean tides under the TOPEX/Poseidon mission)

7) Minchew et al. (2017) :

Initially we have chosen to only mention this study as a direct comparison between Minchew's approach and ours is currently not possible yet with our limited availability of SAR data. We agree that including some of the lessons they have learned is beneficial for the paper.

p.2, l34 to p.3, l2 : Baek and Shum (2011)
We changed the structure as requested and thank reviewer #2 for his explanation that even an accurate O1 will positively affect other components.

p.3, l.10-13 : Stammer et al., (2014)
We removed the very small error and the reference to Stammer et al. 2014

p.4, l.9 : Pawlowicz et al., (2002)
We included the reference to Foreman (1977) as requested

p.4, l.27 : perfect tide models
We have changed the sentence to weaken the statement about us demanding a 'perfect' tide prediction

p.4, l.28-29 : Wild et al., (2018)
We expand on the method to adjust tide model output as requested

p.6, l.7-8 : inference
We expand on the t_tide part of the analysis and thank reviewer #2 for his explanation of the problem and confirming that a prediction within a 16 days window is sufficient but should not be used to extrapolate (which is also our experience elsewhere, but using inference might be a solution to this problem).

p.6, l.25-28 : Hillary comparison
We agree that both are 'slightly above' but leave the wording as 'close to.., and slightly above' because +1.3cm and +4.4cm are still a bit different.

p.7, l.7-9 : Sources of errors in tide models
We elaborate on error sources of tide models near the grounding line (inaccurate bathymetry, wrong water column thickness, small-scale currents in the sub-ice shelf cavity, etc) and note that these uncertainties can be addressed by taking ice mechanics in the grounding zone into account.

p.7, l.24-26 : Discontinuities
We removed jargon.

p.8, l.29 to p.9, l.1 : K1 from a long record
We first use the adjusted Ross_Inv_2002 tide model (all components) to force the flexure models, we then extract the model solutions at the locations of the 7 tiltmeters. We then run t_tide on these solutions and compare their K1 components to the K1 components of the corresponding tiltmeters (now with amplitude and phase errors). As stated above, the K1+P1 inference is within these error bars. We therefore didn't use inference at this stage as we only look at a relatively short time window, when our 'K1 plus minus errors' is within the modulation of K1+P1 during these 16 days.

p.9, l.7-9: orbit alignment:
For the Darwin Glacier we acquired SAR data from only one track, for the SMIS we acquired SAR data from three different tracks. If the times of all satellite overpasses in a year are plotted against the prevailing tidal amplitude we can see that the observed tidal amplitude is only varying once throughout the year as stated in the text.

[Figure]

*Illustration 2: Prevailing tidal amplitudes at the times of satellite overpasses of the TerraSAR-X satellite with an exact repeat pass of 11 days. Solid lines are a result of a 2nd order polynomial fit and show that the observed tidal amplitude varies only once throughout the year*

p.9, l.20-21 : Phase
This is true, we state it now in the Methods section

p.9, l.26-28 : Viscoelasticity
We now defend this statement with a corresponding figure.

p.10, l.9-16 : Future work
Our long-term goal is to build on the work published in Marsh et al., 2014 who use inverse modeling of **elastic** tidal flexure to determine grounding-zone ice thickness from DInSAR. One idea is to invert a **viscoelastic** tidal flexure model for thickness, but this requires the use of tide models which aren't accurate enough yet to take advantage of the superiority of viscoelastic models. Another idea is to define criteria when an elastic model is sufficient to describe tidal flexure which would allow a purely elastic inversion of DInSAR. For this reason we would like to keep this paragraph in the paper but it was significantly shortened.

Technical Corrections:
1) Thank you for the explanation, we have corrected hyphens accordingly
2) We have tried to follow this suggestion as much as possible
3) We have changed 'heterogenity' to 'heterogeneity'
4) We have changed 'theta' to 'Delta A' in the text, tables and figures
Fig.1 caption : (a) changed as suggested (b) reworded the sentence
Fig.2 : changed as suggested
Fig.3 : changed as suggested
Fig.5 (now Fig.6) : changed as suggested
Fig.7 (now Fig.8) : changed as suggested
Fig.8 (now Fig.9) : changed as suggested and included label of Black Island
Fig.9 (now Fig.10) : changed as suggested
(new figure request) : the new figure has been included in the main text

Other related changes to the manuscript
All colorbars in the figures were modified to have a white background if they are displayed within individual panels.

---

## Author Comment (AC3) · 16 Jul 2019

[revised manuscript text omitted]

Present-day displacement measurements by interferometry are exacerbated by the requirement of phase unwrapping, which is the most crucial processing step in any InSAR method.  Discontinuities in the fringe pattern can cause jumps in the un-
25  wrapped phase and may therefore bias the continuous motion field.

Due to these complications, only very few studies have attempted to derive a tide model from DInSAR: Minchew et al. (2017) developed an unprecedented spatially and temporally dense SAR data acquisition campaign for the Rutford Ice Stream, Weddell Sea. Their novel Bayesian method to unequivocally separate a complete set of  tidal harmonics from nontidal ice-surface variability is unique, but beyond the data availability for the remainder of Antarctica. Baek and Shum (2011) succeeded
30  in using data from the ERS-1/2 tandem mission to map the dominant tidal costituent (O1) in Sulzberger Bay, Ross Sea, but data limitations prevented them from developing a full tidal model. In this case, too short a time span (71 days) eliminated a change of the observed tidal amplitudes as the repeat-pass cycle of the SAR satellites masked the sensor's sensitivity to tidal variability. However, even identifying only the dominant tidal consituent is valuable; as it indicates ways in which tide models need to be changed, and these changes will ultimately filter into other consituents. In the Ross Sea, the tidal oscillation is dominated by

diurnal harmonics (Padman et al., 2018). An accurate inversion of TerraSAR-X data with an exact integer number of repeat passes to a complete set of tidal constituents is therefore not possible from DInSAR measurements alone.

Tide models can be consulted to predict both the timing and magnitude of the dominant harmonics. Numerous tide models of various spatial scales (global vs regional) and complexity have been developed (see Stammer et al., 2014, for an overview). While forward models integrate the equations of fluid motion subjected to a gravitational forcing over time, inverse models assimilate measurements of vertical displacement from laser altimetry, tide gauges and GPS (Egbert and Erofeeva, 2002; Padman et al., 2003, 2008). Since the modelled physics is generally simple and gravitational forces are well known, tide model predictions are of high quality in areas where ice is freely floating on the ocean . In coastal areas, in turn, tide models are prone to inaccuracies due to errors in model bathymetry, grounding-line location and insufficient knowledge of the ice water drag coefficient (Padman et al., 2018). Another source of error arises from the conversion of ice-shelf draft to ice-shelf thickness and subsequent estimation of water-column thickness. This freeboard conversion assumes that ice near the coastline is in local hydrostatic equilibrium, whereas stresses from the grounded ice clearly prevent a freely-floating state. A bias of the hydrostatic solution towards thicker ice (Marsh et al., 2014), and therefore a thinning water-column thickness, may negatively affect the tidal prediction. In summary, the relatively coarse spatial resolution and underlying assumptions of contemporary tide models introduce inaccuracies especially in feature-rich coastal areas such as fjord-type outlet glaciers. Although average tide model accuracy has improved markedly in coastal areas over one decade, from about $\pm 10$ cm (Padman et al., 2002) to $\pm 6.5$ cm (Stammer et al., 2014), they are still a magnitude larger than the sub-centimetre accuracy of DInSAR (Rignot et al., 2011). For this reason, we consider DInSAR as the absolute truth and use these space-borne measurements to correct tide-model output.

In this manuscript, we show how the spatial benefits and high accuracy of DInSAR can be used to refine coarse resolution tide models to adequately resolve ocean tides along the feature-rich Antarctic coastline. First we introduce the necessary data set, describe the preprocessing and guide through the work flow. Second we test the algorithm for the Southern McMurdo Ice Shelf (SMIS), a small and almost stagnant ice shelf with a simple grounding-zone geometry, and expand the study to the Darwin Glacier, a relatively fast-flowing outlet glacier within a complex fjord-like embayment. We validate our results with dedicated field measurements taken within the Transantarctic Ice Deflection Experiment (TIDEx) in 2016. We then demonstrate how this exercise can also be applied to reveal errors in interferometric phase unwrapping and answer fundamental questions about the physical properties of ice in Antarctic glaciology.

**2    Methodology**

**2.1    Summary of SAR image processing**

To develop the method, we use 11-day repeat-pass TerraSAR-X data in StripMap imaging mode. The satellite acquires X-band radar data (wavelength 3.1 cm, frequency 9.6 GHz) with a ground resolution of slightly below 3x3 m and images covering an area of 30x50 km. We calculate vertical surface displacement due to ocean tides using the Gamma software package (Werner et al., 2000). InSAR and DInSAR image combinations are generally chosen so that a later image is always subtracted from an earlier image. For image triplets, the central SAR image serves as a common reference during the co-registration. We then

correct the resulting DInSAR interferograms for apparent vertical displacement due to horizontal surface motion (Rack et al., 2017) using the method presented in Wild et al. (2018).

**2.2 Tide models**

The predictions of five tide models are validated: the regional barotropic models (1) Circum-Antarctic Tidal Solution (CATS2008a_opt) developed by Padman et al. (2008), (2) Ross Sea Height-Based Tidal Inverse Model (Ross_Inv_2002) developed by Padman et al. (2003), (3) Ross Sea assimilation model (Ross_VMADCP_9cm), (4) the fully global barotropic assimilation model (TPXO7.2) from Oregon State University developed by Egbert and Erofeeva (2002), and (5) the (t_tide) prediction of GPS data from freely-floating areas following the harmonic analysis of Pawlowicz et al. (2002) which is based on Foreman (1977). The t_tide software is a widely-used toolbox for performing classical harmonic analysis of ocean tides. It can analyse any time series record and outputs the amplitude and phase of its dominant harmonics with error estimates, along with a tidal prediction that is free from non-tidal effects. The isostatic deformation of the Earth's lithosphere underneath the moving water masses is modelled using TPXO7.2 load tide model (Egbert and Erofeeva, 2002), which itself is based on 13 tidal constituents and added to all tide model predictions except t_tide. In addition to the tidal motion underneath the floating ice, much of the ice-surface variability can be attributed to the Inverse Barometric Effect (IBE, Padman et al., 2003). A +1 hPa anomaly of atmospheric pressure translates to an instantaneous -1 cm change on the ice-shelf surface. It is noteworthy to mention that we did not apply a running mean to the pressure records, as the application of any window length worsened the fit to available GPS data. To correct for the IBE outside the GPS period, we make use of atmospheric pressure records obtained by nearby automatic weather stations on Ross Island (Scott Base AWS) and the Ross Ice Shelf (Marilyn AWS). We validate these records with separate barometric measurements taken within the TIDEx campaign and find very good agreement.

In this paper, we use the terms 'traditional tide-modelling' or 'tide model' to refer to the sum of ocean-tide, load-tide model outputs and the IBE. Freely-floating areas of ice shelves and glaciers are expected to experience the full oscillation of this tide model. Traditional tide-modelling, however, neglects ice mechanics in grounding zones where tidal flexure significally affects the surface elevation signal in reality. Other signals that change sea-level height such as mean dynamic topography and storm surges are also excluded from this type of tide model.

**2.3 In-situ data**

We set-up a number of GPS receivers to measure ice-surface motion at millimetre accuracy and high temporal resolution. Although we used GPS data from the freely-floating parts to develop local tide models using t_tide, all GPS data was only used for validation purposes and did not feed into the algorithm. GPS measurements were differentially corrected using static base stations to increase their spatial accuracy. We also installed an array of seven tiltmeters to record surface flexure over 16 days across the grounding zone to confine the physical properties of Antarctic ice. The tiltmeters were complemented by a dense network of point measurements of ice thickness using the new autonomous phase-sensitive radar echo sounder (ApRES, Nicholls et al., 2015).

**2.4    Combining DInSAR and tide models**

To allow a correct interpretation of DInSAR images covering grounding zones, it is desirable that tide models replicate DInSAR observations in freely-floating areas. We first adjust the tide-model output to match the highly-accurate DInSAR measurements using a least sum of squares routine (Wild et al., 2018). By doing so, we consider just the tide model amplitude to contain errors.

5   Possible tide-model phase errors are then accounted for by adjusting the absolute amplitude and thus the rate of tidal change during the times of SAR data acquisition. Second we build on earlier work by Han and Lee (2014) and develop an empirical displacement map showing tide-deflection ratio throughout the satellite image ($\alpha$-map). By feeding the $\alpha$-map with the adjusted tide model output, the 'point forecast' is then spatially extended to predict the mean vertical displacement for every pixel at the times of SAR data acquisition. We then perform the double differences of the empirical model corresponding to the SAR image

10  combinations used to generate the DInSAR images. The original DInSAR satellite measurements are subsequently removed from the mean DInSAR images to calculate their misfits, $\mu$. We now compute the least-squares solution to the equation $Ax = b$ such that the 2-norm $|b - Ax|$ is minimized. Here, $A$ is the $m \times n$ DInSAR matrix of SAR image combinations with $m$ rows of SAR images and $n$ columns of coherent DInSAR interferograms; $b$ is a vector of $\alpha$-prediction misfits and $x$ the least-squares solution of this system of linear equations. The values of $x$ correspond to how much an $\alpha$-prediction deviates from the 'real'

15  vertical displacement at the times of SAR data acquisition. We therefore subtract these offsets, $\Delta A$, from the $\alpha$-prediction maps.

We now demonstrate the workflow in one spatial dimension with an example of the Southern McMurdo Ice Shelf (78°15' S, 167°7' E, SMIS). In this study area, we derived 9 DInSAR images from 12 TerraSAR-X scenes in 2014 (Rack et al., 2017). The low number of DInSAR images is a consequence of the SAR scenes being acquired on 3 different satellite tracks. The

20  resulting system of linear equations is therefore underdetermined as there are more offsets, $\Delta A$, than misfits, $\mu$, to constrain the least-squares solutions. We choose a pixel on the freely-floating end of a profile through the ice-shelf grounding zone to represent the unrestricted ice shelf movement and calculate the percentage vertical displacement of every other pixel from this location. Averaged over the 9 DInSAR interferograms this pixel retains 100% vertical displacement (red areas in Fig. 1), while grounded areas experience zero net uplift (purple areas). Individual pixels on the freely-floating part of the SMIS may show

25  $\alpha$-values slightly above 100%.

We now extract the $\alpha$-values along the profile from the $\alpha$-map. This $\alpha$-profile can be multiplied with the individual DInSAR measurements on its freely-floating end which results in empirically derived $\alpha$-predictions (Fig. 2 center). These mean predictions do not perfectly replicate the DInSAR measurements (Fig. 2 top). Their misfits, however, show a very systematic pattern (Fig. 2 bottom). It is desirable to find a combination of offsets that have the least deviation from the $\alpha$-predictions. We therefore

30  hypothesize that this rather systematic signal can be reconstructed using a least-squares strategy. We solve the underdetermined system simultaneously by finding the combination of offsets that result in a minimal sum of squares. The reconstructed offsets must then be removed from the $\alpha$-prediction for the times of SAR data acquisition (Fig. 3 top). The computed least-square offsets generally replicate the pattern of the misfits (Fig. 3 center) and result in smooth displacement profiles in the ice-shelf grounding zone (Fig. 3).

**3  Results**

In this section we apply the workflow in two spatial dimensions to the Darwin Glacier ($79°53'$ S, $159°00'$ E). In this study area, we derived a total of 45 DInSAR images from 12 SAR scenes being acquired on the same satellite track. SAR image combinations were generally chosen consecutively so that a later image is always subtracted from an earlier image. For image triples, the central image was taken as a common reference/master image. Additionally the data gap between SAR 8 and 9 was taken into account (no 8-9 combination as loss of coherence over this relatively long interval). The advantage of using every other remaining combination (Tab. 2) is that more double-differential measurements of tidal amplitude are available for the least-squares fitting algorithm than only using consecutive pairs alone. The system of linear equations is then overdetermined. A dedicated field campaign was conducted in the Darwin Glacier grounding zone in 2016 and *in-situ* data is available for numerical modelling and field validation purposes. In contrast to the simple geometry at the SMIS, the Darwin Glacier consists of a feature-rich embayment that is constrained by steep topography at its margins. Additionally a buttressing ice rise to the Ross Ice Shelf restricts outflow in the North.

**3.1  Reconstruction of displacement maps during satellite overpasses**

From the interferogram dataset we identify a corridor of only about 2 km width along the centerline where the glacier can be assumed to be freely floating (Fig. 1). This area is expected to experience the full oscillation predicted from tide models. We run five tide models to predict the tidal oscillation at the GPS station 'Shirase' over the time span of SAR data acquisitions. Here we use atmospheric pressure data from the automatic weather station 'Marilyn' which is located about 120 km away on the Ross Ice Shelf to correct for the inverse barometric effect. This record correlates well (Pearson's correlator 0.989) with a mean of seven barometers installed over 14 days across the Darwin Glacier during the TIDEx campaign. All tide-model predictions show a clear fortnightly occuring spring-neap tidal cycle which is superimposed by a dominant diurnal signal (Fig. 4). The approximately fortnightly occurring spring/neap tides are largely owed to the difference in wavelength between the K1 and O1 constituents.

We apply t_tide to our 16 day record of the 'Shirase' GPS to test the potential of short-term GPS surveys to improve current Antarctic tide models. The problem with using such a short window to determine a full set of tidal constituents results from the interplay of the lunar diurnal tide (K1, 23.93 h) with the solar diurnal tide (P1, 24.07 h) as they are close in frequency and P1 has an amplitude of $15 - 20$ % of K1. Without accounting for their inference, t_tide just extracts an apparent K1 from a 16 day record that is really K1+P1. As a consequence, the K1 tide from our harmonic analysis can vary by $30 - 40$ % over a 6-month period and its amplitude is only controlled by the exact time that the GPS data was acquired within the K1+P1 modulation cycle. Additionally, harmonic decomposition of GPS data is subject to inaccuracies itself with errors in both the extracted amplitudes and phases. These errors were found to be of the same magnitude as the K1+P1 inference. For this reason, we did not use inference to separate K1 and P1 (or similarly to separate the semi-diurnal S2 and K2 constituents), but perform a thorough analysis on the identified uncertainty range. While the analysis captures the dominant K1 constituent in the Ross Sea within a reasonable signal-to-noise ratio, fortnightly harmonics could not be retrieved adequately from this time series

alone. The t_tide prediction is therefore the least accurate tide model and requires the largest adjustment to match DInSAR (Tab. 1). Although all the corrected tide model outputs now replicate our DInSAR measurements, their rate of tidal change is affected by the adjustment. Offsets computed for the Ross_Inv_2002 tide model are generally below 10 cm, whereas other tide models require adjustments of up to 13.3 cm (Tab. 1). This agrees well with the findings of Han et al. (2013), who find that the Ross_Inv_2002 model is the optimum tide model for the Terra Nova Bay with a 4.1 cm RMSE against 11 days of tide gauge data. We therefore choose Ross_Inv_2002 for numerical modelling purposes to minimize any effects on a viscoelastic model, but use TPXO7.2 to reconstruct vertical displacement at the times of satellite overpasses as it fits best to our GPS measurements. We refer to the Appendix for a validation of individual tide model output with GPS data from 'Shirase' (Fig. A1).

After the adjustment, modelled tidal amplitudes range from -0.966 m to 0.781 m over the whole SAR period (Fig. 4). Mean absolute residual error to the 45 DInSAR measurements at the tide-model location 'Shirase' is just 7 mm (Tab. 2), which can be explained by interferogram noise. We attribute this accuracy to the exceptionally high phase coherence of the TerraSAR-X data set. The reconstruction algorithm results in 12 smooth vertical displacement maps for the times of SAR data acquisition (Fig. 6).

**3.2   Validation with GPS measurements**

We now validate these reconstructions with available field data. As both GPS records overlap with the acquisition of SAR image 11, we extract the vertical displacement along the glacier's centerline and plot the profiles against the two GPS point measurements. The GPS measurement at 'Hillary' is 0.169 m, which is close to the reconstruction of 0.156 m. The 'Shirase' GPS measurement is 0.566 m, which is slightly above the reconstruction of 0.522 m. We attribute the deviations of +1.3 and +4.4 cm, respectively, to a combination of interpolation artefacts, temporal smoothing of the GPS data and residual errors of the least-squares algorithm. The overall shape of the vertical displacement is well reproduced as observed with both GPS measurements (Fig. 7).

**3.3   Applications**

**3.3.1   Tide-model refinement**

A map of tide-deflection ratio ($\alpha$-map) can be combined with the tide model to predict an average time series of vertical displacement between the times of SAR image acquisition. With this approach, the coarse grid of traditional tide models is refined to resolve small-scale features of vertical tidal displacement throughout the embayment. The $\alpha$-value for the pixel containing the 'Hillary' GPS station is 46.06%. We use this value and linearly scale the adjusted tide-model output for the location of the 'Shirase' GPS to predict vertical tidal motion within the flexure zone. This scaling maintains the tide model's high correlation (Pearson's correlator 0.95) with the 'Shirase' record, but improves the RMSE between the TPXO7.2 output and the 'Hillary' record from 21.4 cm to 5.6 cm, which corresponds to an improvement of -74% to GPS data (Fig. 8). The

primary reason for this large improvement, however, is that the tide model now takes the damping of the tidal signal by ice mechanics in the grounding zone into account.

**3.3.2 Ice heterogeneity**

With the 12 reconstructed displacement maps at hand, it is now possible to perform any image combination. We mosaic the 45 double differences corresponding to DInSAR combinations (Tab. 2) to allow a more direct comparison between measured and modelled interferograms. SAR image combinations were chosen so that the loss of coherence between SAR 8 and 9 was taken into account and that a maximum number of consecutive, double-differential interferograms was available for the least-squares fitting routine. The synthetic interferograms replicate not only simple tidal fringes as measured with DInSAR, but also show complicated viscoelastic signals within the grounding zone (Fig. 5). For an overall assessment of model performance we calculate again the misfit between each modelled and observed interferogram for every pixel, but this time after using the adjusted tide-model output. The standard deviation of these misfits is shown in Fig. 9, with the majority of the glacier surface below noise level of interferograms ($\sigma < 1.0$ cm). We identify a narrow band with higher standard deviations ($\sigma \approx 2.0$ cm) from the inner shear margin of the Darwin Glacier extending along flow direction onto its ice shelf. Standard deviations are largest out in the shear zone of the fast-flowing Ross Ice Shelf and above steep rocky cliffs ($\sigma > 4.0$ cm) which is a result of poor phase coherence or layovers in SAR images in these areas .

**3.3.3 Detection of errors in phase unwrapping**

We now extend the earlier one dimensional analysis of the SMIS to a two dimensional re-analysis of the SMIS data set and calculate misfits of 9 DInSAR interferograms. Resulting standard deviations are generally smaller in this area ($\sigma < 0.3$ cm) and smoothly distributed throughout the map. We identify two regions of phase discontinuities between adjacent cells at the SMIS. Both extend from the center of an ice rise towards Black Island (cyan and green areas in Fig. 9) with $\sigma \approx 0.4$ cm and $\sigma \approx 0.5$ cm, respectively. We interpret these rapid increases as a proxy for errors in the DInSAR measurements, as the modelled least-square interferograms originate from a curvature-minimizing polynomial interpolation. We re-evaluate the remote-sensing part of the analysis and find discontinuities in DInSAR interferograms ID:1 and ID:8 that match the course of the two phase jumps in the standard deviation map. These discontinuities, in turn, occurred during phase unwrapping and can now be corrected.

**3.4 Finite-element modelling of viscoelasticity**

We hypothesize that any non-linear signal due to viscoelastic ice properties is significantly reduced or even completely lost during the averaging step to compute the $\alpha$-map. This signal can then be reconstructed by finding the offsets to match observations made with DInSAR. We therefore subtract the $\alpha$-prediction again from the 12 reconstructions to extract the theorised viscoelastic signal (Fig. 10). This signal is negligible at times during neap tide (SAR 4 and 9) but well pronounced for SAR images acquired during spring-tide periods (SAR 1,6,7,10 and 11).

In order to further explore this pattern, we now make use of the tiltmeter array and ApRES network of ice-thickness measurements at the Darwin Glacier (Fig. 12). We match the numerical solutions from two finite-element models to seven tiltmeter records, with the goal to derive information on the physical properties of Antarctic ice. Thereby, the Young's modulus, $E$, is a measure of ice stiffness and controls the width of the flexure region. The value for ice viscosity, $\nu$, influences the timing of the flexural response within the grounding zone (Wild et al., 2017). Two numerical models of ice-shelf flexure are employed. The elastic approximation (Holdsworth, 1969; Vaughan, 1995; Schmeltz et al., 2002) as formulated by Walker et al. (2013):

$$kw + \nabla^2(D\nabla^2 w) = q, \tag{1}$$

where $w(t)$ is the time-dependent vertical deflection of the neutral layer in a plate, $\nabla^2$ is the Laplace operator in 2-D space and $k = 5$ MPa m$^{-1}$ a spring constant of the foundation which is zero for the floating part. The applied tidal force $q(t)$ is defined by:

$$q = \rho_{sw}g[A(t) - w], \tag{2}$$

with $g = 9.81$ m s$^{-2}$ the gravitational acceleration and $A(t)$ the time-dependent tidal amplitude given by the adjusted Ross_Inv_2002 tide model. We choose this model, in contrast to the TPXO7.2 model, for finite-element simulations to minimize any potential effects of tide-model adjustment on viscoelasticity (Tab. 1). The stiffness of the ice shelf is given by (Love, 1906, p. 443):

$$D = \frac{EH^3}{12(1 - \lambda^2)}, \tag{3}$$

where $E$ is the Young's modulus for ice, $H(x,y)$ our ice thickness map derived from ApRES point measurements and $\lambda = 0.4$ the Poisson's ratio. We compare the elastic model with the viscoelastic approach developed by Walker et al. (2013):

$$\frac{\partial}{\partial t}\left[kw + \nabla^2\left(D\nabla^2 w\right)\right] + \frac{Ek}{2\nu(1 - \lambda^2)}w = \frac{\partial}{\partial t}q + \frac{E}{2\nu(1 - \lambda^2)}q, \tag{4}$$

where $\nu$ is ice viscosity. The following boundary conditions are applied for both models: the upstream boundary of the model domains on the grounded portion are anchored rigidly ($w = 0, \nabla^2 w = 0$), the downstream boundaries on the freely-floating ice shelf are set free. The location of the tide model computation is constrained to be equal to the tidal oscillation ($w = A(t), \nabla w = 0$) and the grounding line is represented by a fulcrum ($w = 0$). Both models are implemented in two spatial dimensions to capture effects of complex grounding-line configuration on ice-shelf flexure (Wild et al., 2018). We then solve the models using the commercial finite-element software COMSOL Multiphysics. As tiltmeters measure slope, $w'$, along their longitudinal axis, we derive the models' solutions for vertical displacement, $w$, with respect to the $x$ and $y$ directions. This allows us to retrieve surface slopes components in easting and northing direction and to rotate them into the individual orientations of the tiltmeter sensors. With our 16 day tiltmeter records it is only possible to capture their diurnal components with confidence. Semi-diurnal, fortnightly and monthly harmonics have been removed from the tiltmeter time series and we focus further analysis only on the K1 component within the 16 day window. Therefore, we now make extensive use of the t_tide program to automatically extract the modelled K1 harmonics from the modelled surface slopes and compare them against the K1 constituents from the tiltmeters. We thereby take amplitude and phase errors that originate from the harmonic analysis of

noisy tiltmeter records into account and find the best rheological parameters to match the elastic and viscoelastic models to these seven K1 components. Incorporating viscoelastic effects into the model simulations always improves the elastic fit to the tiltmeter data within the uncertainty range of K1 amplitude and phase (Tab. 3). We find that an average Young's modulus of $E = 1.0 \pm 0.56$ GPa and an ice viscosity value of $\nu = 10 \pm 3.65$ TPa s fits best to our measurements within uncertainty (Fig. 13). The viscoelastic model gives an average RMSE of 0.00118845 ° to the seven tiltmeters and improves on the elastic approximation with an average RMSE of 0.00147136 ° by $\approx -20\%$.

**4 Discussion**

**4.1 Seasonal bias in $\alpha$-map**

Due to the alignment of the satellite overpasses with the dominant diurnal tidal constituents in the Ross Sea, the observed stage of the tidal oscillation varies only slowly throughout the year. In the austral winter months, TerraSAR-X images have been acquired during stages of low tide, whereas satellite overpasses concur with stages of high tide during the austral summer months. The first 8 snapshots of our SAR data acquisitions for the Darwin Glacier show conditions at low tide and only the last 4 are acquired during high tide. Our $\alpha$-map, in turn, ignores this seasonality and may therefore have a low-tide bias. As a result, the contribution of a tide induced landward migration of the grounding line may be affected by the averaging process. The seasonal bias would then modify the scaling of the tide model within the flexure zone. This is supported by the finding that low-tide stages in the 'Hillary' GPS record are matched closely by the scaled tide model, but peaks during high-tide stages are still over estimated (Fig. 4)

**4.2 Viscoelasticity between snapshots**

Similarly, the linear scaling using an $\alpha$-map only modifies the predicted tidal amplitude, but neglects a viscoelastic time delay in the flexural response towards the grounding line. Wild et al. (2017) found that viscosity is most pronounced in the diurnal tidal components. Harmonic analysis of our GPS records reveals that the diurnal K1 and O1 constituents at 'Hillary' are lagging approximately 20 mins behind 'Shirase'. This signal is currently disregarded in the scaling work flow as ice is treated as a perfect elastic material that transfers tidal forcing instantaneously in the flexure zone. This assumption, however, allowed us to improve the accuracy of the predicted displacement by -74%. Currently, the viscoelastic signal can only be reconstructed for the times of SAR data acquisition. Including viscoelasticity between times of satellite overpasses offers a small, but systematic, opportunity for further refinement. In our study area, the rate of tidal change is up to 10 cm hr$^{-1}$ (Tab. 1) and the viscoelastic misfit corresponding to 20 minutes time delay is therefore up to about 3 cm.

When separating the viscoelastic contribution from the reconstructed maps of vertical displacement at times of satellite overpasses, we assume that an $\alpha$-prediction corresponds to an instantaneous elastic response. This is justified by viscoelasticity being most pronounced when rates of tidal change are maximal. By expressing the viscoelastic misfits in percent of prevailing

tidal amplitude during the times of satellite overpasses, the areas of pronounced viscoelastic effects can be visualised. They are most pronounced within the Darwin Glacier's shear zone (Fig. 11).

When predicting rates of tidal change using the adjusted Ross_Inv_2002 tide model, we identify a threshold of $\dot{A} \approx \pm 0.05$ m h$^{-1}$ (SAR times 1, 2, 7, 8, 10, 11 in Tab. 1) above which viscoelasticity is well represented in the reconstructed vertical displacement maps (panels 1, 6, 7, 8, 10 and 11 in Fig. 10). Image 6 is thereby an exception, as the used Ross_Inv_2002 tide model was largely adjusted (-0.082 m), which affects the viscoelastic model. We find that the error due to viscoelasticity on the floating part of the ice shelf increases with the absolute rate of tidal change (Fig. 11). SAR images acquired during periods of spring tides at the Darwin Glacier show also a significant viscoelastic contribution that diminishes during neap tide periods. These independent observations from satellite data alone support our suggested threshold of $\pm 0.05$ m h$^{-1}$ for the separation of elastic and viscoelastic signals, as derived from tiltmeter data on the Southern McMurdo Ice Shelf presented in an earlier study (Fig. 8 in Wild et al., 2017). The advantage of separating the elastic from the viscoelastic contribution to the tidal flexure pattern is the large potential for improving current inverse modelling techniques to determine grounding-zone ice thickness from DInSAR measurements alone. Hereby, an elastic model is currently employed to optimize grounding-zone ice thickness to match the surface flexure from DInSAR. The applicability of an elastic model varies from location to location as effective viscosity is dependent on ice temperature and shear stress (Marsh et al., 2014). Our method to separate the two contributions to the flexure pattern may therefore help to remove the viscoelastic contamination and allow purely elastic inverse modelling. Such an analysis, however, goes beyond the scope of this manuscript and will be published elsewhere.

**4.3 Large-scale ice anisotropy**

Fast-moving glacial environments like the Darwin Glacier are subject to large deformation by flow convergence and divergence, ice compression and extension, lateral shearing at the margins accompanied by fracture under tension and rapid thinning by basal melt. With cumulative deformation, a crystallographic fabric evolves that reflects the glacier's flow history (Alley, 1988), and with it strain-dependent mechanical anisotropy of ice. The standard deviation map, Fig. 9, shows a narrow band of larger misfits extending from the Darwin Glacier's inner shear margin out towards the freely-floating ice shelf. As preferred crystallographic orientation develops with strain, effective viscosity decreases of about a factor of ten compared to initially isotropic polycrystalline ice (Hudleston, 2015). Our analysis of tiltmeter data reveals a five-fold reduced viscosity at the very dynamic Darwin Glacier compared to an earlier study at the almost stagnant Southern McMurdo Ice Shelf (Wild et al., 2017). We hypothesize that this microscopic process explains the macroscopic response observed here, and accounts for the measured glacial heterogeneity within the embayment. Large scale observations of ice anisotropy, in turn, are currently the missing key to improve parametrisations to account for polar ice anisotropy in ice-sheet flow modelling (Gagliardini et al., 2009). Other processes have been proposed which lead to ice softening in areas with high strain rates. Thermomechanical modelling suggests that shear heating and consequent thermal softening reduces lateral drag in ice-stream margins (Perol and Rice, 2015). Fracture modelling implies that damage reduces ice viscosity along confined crevassed zones with consequences on ice-shelf scale (Albrecht and Levermann, 2014). Full-Stokes viscoelastic modelling shows that Glen's non-linear flow law and tidal stresses in the ice-shelf flexure zone are sufficient to explain large-scale temporal variations in ice dynamics (Rosier and

Gudmundsson, 2018). These processes, or a combination of them, might certainly be at play but they do not explain why a band of higher standard deviations can be observed in the shear zone of the Darwin Glacier which is absent in the flexure zone of the SMIS (Fig. 9). We therefore attribute this difference to ice-fabric reorientation in the shear margin.

**4.4 Refining tidal constituents using DInSAR**

5    The idea of using SAR interferometry to derive a full set of tidal harmonics was first laid out in a study of tides in the Weddell Sea (Rignot et al., 2000). The authors discussed that DInSAR images cannot be transformed into individual displacement fields because of the nonuniqueness of the inversion. A large number of independent DInSAR images is required to overcome this problem and to resolve the phase of tidal constituents that are close to the repeat-pass of the SAR satellite. For example, multiples of the lunar diurnal constituent K1 (23.93 h) are relatively close to the exact integer repeat-pass of TerraSAR-X

10   (11 days) meaning that the observed amplitude of the K1 constituent is only varying once throughout the year. Consequently SAR images need to be acquired at least over the duration of one year to provide some redundancy for the inversion step of DInSAR images to tidal constituents. However, with an exact 12 h period, the stage of the semidiurnal solar tide, S2, will always be the same at each satellite pass making TerraSAR-X and similar satellites with repeat-passes of integer days blind to the S2 constituent. For example Minchew et al. (2017) needed a unique spatially and temporally dense SAR acquisition campaign

15   as well as *a-priori* knowledge of the temporal basis functions from GPS data to empirically determine tidal constituents on Rutford Ice Stream. The four COSMO-SkyMed satellites in orbit, however, produce repeat-passes of 1, 3, 4 and 8 days and are blind to the S2 constituent as well even when using $> 1000$ available DInSAR images. Although other dominant tidal constituents like M2 (12.4 h) and O1 (25.82 h) were inferred successfully, the method presented here can achieve a higher accuracy fo the total tide with less DInSAR images. From another perspective, the inclusion of an auxiliary tide model eases

20   the requirement of a very large number of DInSAR images.

**5   Conclusions and Outlook**

Accurate prediction of ocean tides in coastal areas is crucial as the majority of Antarctica's ice is discharged through large outlet glaciers. We presented a data fusion method between DInSAR and traditional Antarctic tide models to predict spatial variability of tidal motion near the grounding line. The primary value of using DInSAR in conjunction with tide models lies

25   in the spatio-temporal benefits of resolving complex grounding zone deformation. Their symbiosis not only improves current accuracies of the predicted tidal amplitudes in coastal regions generally, but also avoids issues related to the timing of the tidal wave and the sun-synchronous satellite orbit when attempting to derive tide-models from SAR data alone. In our study area, the method presented in this paper improves traditional tide modelling in average by -22% from 11.8 cm to 9.3 cm RMSE against 16 days of GPS data. The GPS station 'Shirase' on the freely-floating part of the Darwin Glacier has proven invaluable

30   to determine which tide model has to be used to best reconstruct the vertical displacement during satellite overpasses. For the Darwin Glacier, the TPXO7.2 tide model predicts best the tidal oscillation. With using DInSAR measurements to adjust the TPXO7.2 tidal prediction, its RMSE could be improved by -39% from 10.8 cm to 6.7 cm.

Our GPS record from 'Shirase' is too short to improve already available Antarctic tide models. A longer record is required to adequately resolve a full set of tidal constituents. We produced an empirical displacement map from DInSAR for tidal deflection ($\alpha$-map). Comparison of a GPS record within the tidal flexure zone 'Hillary' with predicted vertical displacement from feeding the $\alpha$-map with the adjusted TPXO7.2 tide model shows a -74% improvement over using the tide-model output alone. This independent validation supports the finding that our method for making use of DInSAR is very useful for refining tide models in Antarctic grounding zones.

Numerical modelling of ice dynamics in Antarctic grounding zones commonly assumes that ice is isotropic and homogeneous i.e. of the same density and rheological properties throughout. Our analysis reveals that this assumption is valid for the Southern McMurdo Ice Shelf, an almost stagnant area with a simple grounding line configuration, but invalid for the Darwin Glacier, a fast-flowing outlet glacier with complex shear margins causing non-negligible ice heterogeneity within the embayment.

Further work is required in order to improve tide models in a larger variety of grounding zones by including effects of grounding-line migration and variability of horizontal ice flow.

*Code availability.* The code is freely available to the scientific community. Collaboration is anticipated and desired.

*Data availability.* TerraSAR-X data presented in this paper are subject to license agreements. GPS/tiltmeter and ApRES data are available upon request.

*Code and data availability.* No data sets, nor software, are part of this study.

*Sample availability.* No samples were collected for this study.

**Appendix A: GPS evaluation of tide models**

The quality of the used tide model to correctly reconstruct tidal displacement at the times of SAR data acquisitions, is also crucial to accurately predict spatial variability in tidal motion for all times. Here, we assume that a freely-floating area on the ice shelf experiences the full oscillation as predicted from a tide model. In this area, however, tide-model output deviates from our DInSAR measurements. This indicates either that the area under investigation is prevented from a freely-floating state by lateral stresses within the embayment, or that the tide-model prediction is inaccurate for this area. We circumvent this ambiguity by making use of the high vertical accuracy of DInSAR and correct the tide-model prediction to match our satellite measurements. This raises the question of whether the adjustment improves or worsens the match to a 'real' tidal motion ? We

therefore independently evaluate the pre- and post adjustment tide-model predictions and calculate their RMSE to 16 days of GPS data from the freely-floating area (Tab. A1). The adjustment improves all traditional tide model predictions by up to -39% for TPXO7.2, and only worsens the RMSE for the t_tide output by +11%, indicating that a harmonic analysis of GPS data can not be improved by using DInSAR for correction purposes. We choose TPXO7.2 for further processing as it displays the

5    overall smallest RMSE (6.7 cm) and replicates the small-scale variability observed during the neap-tide period in the second half of our GPS record.

*Author contributions.* All authors conceived the study and conducted fieldwork. CW developed the algorithm, performed the data analysis and wrote a first version of the paper. OM and WR processed the SAR images for the Darwin Glacier and Southern McMurdo Ice Shelf, respectively. All authors finalized and approved the manuscript.

10   *Competing interests.* The authors declare that the research was conducted in the absence of any commercial or financial relationships that could be construed as a potential conflict of interest.

*Disclaimer.*

*Acknowledgements.* We thank Antarctica New Zealand for logistical support and the Scott Base staff for their dedication to the Transantarctic Ice Deflection Experiment. The project was supported by the Royal Geographic Society (Marsden Fast Start, PI O. Marsh) and by the Past

15   Antarctic Science and Future Implications Program (PACaFI). D. Price, M. Ryan, D. Floricioiu and E. Scheffler contributed to fieldwork. TerraSAR-X data were provided through DLR project HYD1421. Landsat-8 images courtesy of the US Geological Survey. AWS data from Scott Base were provided by NIWA, and for Marilyn station by AMRC, SSEC, UW-Madison. The authors enjoyed fruitful discussions with H. Purdie and M. Sellier. CW also thanks R. Mueller for sharing oceanographic expertise. The thorough comments of one anonymous reviewer and L. Padman's valuable insight into tidal dynamics largely improved the paper. We also thank A. Robinson for editing.

[revised manuscript text omitted]

---

## Author Comment (AC4) · 16 Jul 2019

see attached pdf

Please also note the supplement to this comment:
https://www.the-cryosphere-discuss.net/tc-2018-269/tc-2018-269-AC4-supplement.pdf
* * *

---

## Author Comment (AC5) · 16 Jul 2019

**Differential InSAR for tide modelling in Antarctic ice-shelf grounding zones**

Christian T. Wild[1], Oliver J. Marsh[1,2], and Wolfgang Rack[1]

[1]Gateway Antarctica, University of Canterbury, Private Bag 4800, Christchurch 8140, New Zealand
[2]British Antarctic Survey, High Cross, Madingley Road, Cambridge, CB3 0ET, United Kingdom

**Correspondence:** Christian T. Wild (christian.wild@canterbury.ac.nz)

**Abstract.** Differential interferometric synthetic aperture radar (DInSAR) is an essential tool for detecting ice-sheet motion near Antarctica's oceanic margin. These space-borne measurements have been used extensively in the past to map the location and retreat of ice-shelf grounding lines as an indicator for the onset of marine ice-sheet instability and to calculate the mass balance of [c1]ice sheets and individual catchments. The main difficulty in interpreting DInSAR is that images originate from a combination of several SAR images and do not indicate instantaneous ice deflection at the time of satellite data acquisition. Here, we combine the sub-centimetre accuracy and spatial benefits of DInSAR with the temporal benefits of tide models to infer the spatiotemporal dynamics of ice-ocean interaction during the times of satellite overpasses. We demonstrate the potential of this synergy with TerraSAR-X data from the almost stagnant Southern McMurdo Ice Shelf. We then validate our algorithm with GPS data from the fast-flowing Darwin Glacier, draining the Antarctic Plateau through the Transantarctic Mountains into the Ross Sea. We are able to match DInSAR [c2]derived vertical displacements to [c3]7 mm; generally improve traditional [c4]tide model output by up to -39% from 10.8 cm to 6.7 cm RMSE against GPS data from areas where ice is in local hydrostatic equilibrium with the ocean; and up to -74% from 21.4 cm to 5.6 cm RMSE against GPS data in feature-rich coastal areas where [c5]tide models have not been applicable before. Numerical modelling then reveals a Young's modulus of [c6]$E = 1.0 \pm 0.56$ GPa and an ice viscosity of [c7]$\nu = 10 \pm 3.65$ TPa s when finite-element simulations of tidal flexure are matched to 16 days of tiltmeter data; supporting the [c8]hypothesis that strain dependent anisotropy may significantly decrease effective viscosity compared to isotropic polycrystalline ice on large spatial scales. Applications of our method range from (i) refining coarsly-gridded tide models to resolve small-scale features at the spatial resolution and vertical accuracy of SAR imagery, to (ii) separating elastic and viscoelastic contributions in the satellite derived flexure measurement and (iii) gaining information about large-scale ice [c9]heterogeneity in Antarctic ice-shelf grounding zones, the missing key to improve current ice-sheet flow models. The reconstruction of the individual components forming DInSAR images has the potential to become a
* * *
[c1]

[c2] *Text added.*

[c3]

[c4]

[c5]

[c6] *Text added.*

[c7] *Text added.*

[c8]

[c9]

standard remote-sensing method in polar tide modelling. Unlocking the algorithm's full potential to answer multi-disciplinary research questions is desired and demands collaboration within the scientific community.

*Copyright statement.* TEXT

**1 Introduction**

5 The periodic rise and fall of the ocean's surface is caused by the gravitational interplay of the Earth-Moon-Sun system and our planet's rotation. Knowledge of ocean tides is fundamental to fully understand oceanic processes, sedimentation rates and behaviour of marine ecosystems. In Antarctica, the tidal oscillation also controls the motion of ice sheets near the coastline and ocean mixing in the sub ice-shelf cavity which modifies heat transport to the ice-ocean interface (Padman et al., 2018).

SAR satellites repeatedly illuminate Earth's surface and record the backscattered radar wave. While the SAR signal's am-
10 plitude depends on reflection intensity and is mainly characterized by the surface, the recorded phase holds information about the distance travelled by the signal (Massom and Lubin, 2006). Two-pass interferometry (InSAR) can be used to determine surface motion with sub-centimetre accuracy over vast remote areas, [c1]and InSAR has been applied to measure surface velocity of floating ice (e.g. Tong et al., 2018) and to observe tidal strain of landfast sea ice (Han and Lee, 2018). In grounding zone areas, where an [c2]ice sheet comes in contact with the ocean for the first time and forms floating glaciers and [c3]ice shelves,
15 InSAR has become the state-of-the-art practice to measure the flux divergence of ice-flow velocity (Mouginot et al., 2014; Han and Lee, 2015) and thus the mass balance of many [c4]ice shelves around Antarctica (Rignot et al., 2013). InSAR can also be used to identify vertical deflection due to ocean tides. Horizontal [c5] and vertical motion cannot be distinguished [c6]in single interferograms but the unsteady tidal contribution can be extracted by [c7]DInSAR using triple or quadruple combination[c8]s of [c9] SAR images. This [c10]is based on the assumption that horizontal flow is [c11]time invariant, and that its phase contribution
20 [c12]therefore cancels out. The double-differential measurement of vertical displacement only is known as Differential InSAR (DInSAR). While DInSAR has often been applied to detect the grounding line movement around Antarctica (Konrad et al., 2018) the signal can also be used to measure spatial variability of ocean tides at very high ground resolution. This second application is complicated by the fact that DInSAR interferograms show a combination of multiple stages of the tidal oscillation.
* * *
[c1]
[c2]
[c3]
[c4]
[c5]
[c6]
[c7]
[c8] *Text added.*
[c9]
[c10]
[c11]
[c12]

Tidal migration of the grounding line as well as viscoelastic time delays in ice displacement, tidally-induced velocity variations and geometric effects on the surface flexure also complicate the correct interpretation of DInSAR interferograms to date (Rack et al., 2017; Wild et al., 2018).

Present-day displacement measurements [c1]by interferometry are exacerbated by the requirement of phase unwrapping, which is the most crucial processing step in any InSAR method. [c2] [c3]Discontinuities in the [c4]fringe pattern can cause jumps in the unwrapped phase and may therefore bias the continuous motion field.

Due to these complications, only very few studies have attempted to derive a tide model from DInSAR: Minchew et al. (2017) developed an unprecedented spatially and temporally dense SAR data acquisition campaign for the Rutford Ice Stream, Weddell Sea. Their novel Bayesian method to unequivocally separate a complete set of [c5] tidal harmonics from nontidal ice-surface variability is unique, but beyond the data availability for the remainder of Antarctica. Baek and Shum (2011) [c6]succeeded in using data from the ERS-1/2 tandem mission to map the dominant tidal costituent (O1) in Sulzberger Bay, Ross Sea, but data limitations prevented them from developing a full tidal model. In this case, too short a time span (71 days) eliminated a change of the observed tidal amplitudes as the repeat-pass cycle of the SAR satellites [c7]masked the sensor's sensitivity to tidal variability. [c8]However, even identifying only the dominant tidal consituent is valuable; as it indicates ways in which tide models need to be changed, and these changes will ultimately filter into other consituents. In the Ross Sea, the tidal oscillation is dominated by diurnal harmonics (Padman et al., 2018). An accurate inversion of TerraSAR-X data with an exact integer number of [c9]repeat passes to a complete set of tidal constituents is therefore not possible from DInSAR measurements alone.

[revised manuscript text omitted]

10   atmospheric pressure translates to [c4]an instantaneous -1 cm [c5]change on the ice-shelf surface. [c6]It is noteworthy to mention that we did not apply a running mean to the pressure records, as the application of any window length worsened the fit to available GPS data. To correct for the IBE [c7]outside the GPS period, we make use of atmospheric pressure records obtained by nearby automatic weather stations on Ross Island (Scott Base AWS) and the Ross Ice Shelf (Marilyn AWS). We validate these records with separate barometric measurements taken within the TIDEx campaign and find very good agreement.

15    [c1]In this paper, we use the terms 'traditional tide-modelling' or 'tide model' to refer to the sum of ocean-tide, load-tide model outputs and the IBE. Freely-floating areas of ice shelves and glaciers are expected to experience the full oscillation of this tide model. Traditional tide-modelling, however, neglects ice mechanics in grounding zones where tidal flexure significally affects the surface elevation signal in reality. Other signals that change sea-level height such as mean dynamic topography and storm surges are also excluded from this type of tide model.

20   ## 2.3   In-situ data

We set-up a number of GPS receivers to measure ice-surface motion at millimetre accuracy and high temporal resolution. [c2]Although we used GPS data from the freely-floating parts [c3]to develop local tide models using t_tide, [c4]all GPS data was only used for validation purposes and did not feed into the algorithm. [c5] [c6] GPS measurements were differentially corrected using static base stations [c7]to increase their spatial accuracy. We also [c8]installed an array of seven tiltmeters [c9]to record surface
* * *
[c2] *Text added.*

[c3] *Text added.*

[c4]

[c5]

[c6] *Text added.*

[c7] *Text added.*

[c1] *Text added.*

[c2]

[c3]

[c4] *Text added.*

[c5]

[c6]

[c7] *Text added.*

[c8]

[c9]

flexure over 16 days across the grounding zone to confine the physical properties of Antarctic ice. [c10]The tiltmeters were complemented by a dense network of point measurements of ice thickness using the new autonomous phase-sensitive radar echo sounder (ApRES, Nicholls et al., 2015).

**2.4 Combining DInSAR and tide models**

5   [c1]To allow a correct interpretation of DInSAR images covering grounding zones, it is desirable that tide models replicate DInSAR observations [c2]in freely-floating areas. We first adjust the tide-model output to match the highly-accurate DInSAR measurements using a least sum of squares routine (Wild et al., 2018). [c3]By doing so, we consider just the tide model amplitude to contain errors. Possible tide-model phase errors are then accounted for by adjusting the absolute amplitude and thus the rate of tidal change during the times of SAR data acquisition. Second we build on earlier work by Han and Lee (2014)

10   and develop an empirical displacement map showing tide-deflection ratio throughout the satellite image ($\alpha$-map). By feeding the $\alpha$-map with the adjusted tide model output, the [c4]'point forecast' is then spatially extended to predict the mean vertical displacement for every pixel at the times of SAR data acquisition. We then perform the [c5]double differences of the empirical model corresponding to the SAR image combinations used to generate the DInSAR images. The original DInSAR satellite measurements are subsequently removed from the mean DInSAR images to calculate their misfits, $\mu$. We now compute the

15   least-squares solution to the equation $Ax = b$ such that the 2-norm $|b - Ax|$ is minimized. Here, $A$ is the $m \times n$ DInSAR matrix of SAR image combinations with $m$ rows of SAR images and $n$ columns of coherent DInSAR interferograms; $b$ is a vector of $\alpha$-prediction misfits and $x$ the least-squares solution of this [c6]system of linear equations. The values of $x$ correspond to how much an $\alpha$-prediction deviates from the 'real' vertical displacement at the times of SAR data acquisition. We therefore subtract these offsets, [c7]$\Delta A$, from the $\alpha$-prediction maps.

20   We [c8]now demonstrate the workflow in one spatial dimension with an example of the Southern McMurdo Ice Shelf (78°15' S, 167°7' E, SMIS). In this study area, we derived 9 DInSAR images from 12 TerraSAR-X scenes in 2014 (Rack et al., 2017). [c9]The low number of DInSAR images is a consequence of the SAR scenes being acquired on 3 different satellite tracks. The resulting system of linear equations is therefore underdetermined as there are more offsets, $\Delta A$, than misfits, $\mu$, to constrain the least-squares solutions. We choose a pixel on the freely-floating end of a profile through the ice-shelf grounding zone to

25   represent the unrestricted ice shelf movement and calculate the percentage vertical displacement of every other pixel from this location. Averaged over the 9 DInSAR interferograms this pixel retains 100% vertical displacement (red areas in Fig. 1), while
* * *
[c10]

[c1]

[c2]

[c3] *Text added.*

[c4]

[c5]

[c6]

[c7]

[c8] *Text added.*

[c9] *Text added.*

grounded areas experience zero net uplift (purple areas). Individual pixels on the freely-floating part of the SMIS may show $\alpha$-values slightly above 100%.

We now extract the $\alpha$-values along the profile from the $\alpha$-map. This $\alpha$-profile can be multiplied with the individual DInSAR measurements on its freely-floating end which results in empirically derived $\alpha$-predictions (Fig. 2 center). These mean predictions do not perfectly replicate the DInSAR measurements (Fig. 2 top). Their misfits, however, show a very systematic pattern (Fig. 2 bottom). It is desirable to find a combination of offsets that have the least deviation from the $\alpha$-predictions. We therefore hypothesize that this rather systematic signal can be reconstructed using a least-squares strategy. [c1] We solve the [c2]underdetermined system simultaneously by finding the combination of offsets that result in a minimal sum of squares. The reconstructed offsets must then be removed from the $\alpha$-prediction for the times of SAR data acquisition (Fig. 3 top). The computed least-square offsets generally replicate the pattern of the misfits (Fig. 3 center) and result in smooth displacement profiles in the ice-shelf grounding zone (Fig. 3).

**3   Results**

In this section we apply the workflow in two spatial dimensions to the Darwin Glacier (79°53' S, 159°00' E). [c3]In this study area, we derived a total of 45 DInSAR images from 12 SAR scenes being acquired on the same satellite track. SAR image combinations were generally chosen consecutively so that a later image is always subtracted from an earlier image. For image triples, the central image was taken as a common reference/master image. Additionally the data gap between SAR 8 and 9 was taken into account (no 8-9 combination as loss of coherence over this relatively long interval). The advantage of using every other remaining combination (Tab. 2) [c4]is that more double-differential measurements of tidal amplitude are available for the least-squares fitting algorithm than only using consecutive pairs alone. The system of linear equations is then overdetermined. A dedicated field campaign was conducted in [c5]the Darwin Glacier grounding zone in 2016 and *in-situ* data is available [c6]for numerical modelling and field validation purposes. In contrast to the simple geometry at the SMIS, the Darwin Glacier consists of a feature-rich embayment that is constrained by steep topography at its margins. Additionally a buttressing ice rise to the Ross Ice Shelf restricts outflow in the North.

**3.1   Reconstruction of displacement maps during satellite overpasses**

From the interferogram dataset we identify a corridor of only about 2 km width along the centerline where the glacier can be assumed to be freely floating (Fig. 1). This area is expected to experience the full oscillation predicted from tide models. We run five tide models to predict the tidal oscillation at the GPS station 'Shirase' over the [c7]time span of SAR data acquisitions.
* * *
[c1] Here, the linear system is under-determined with 9 DInSAR equations and 12 unknown SAR offsets.

[c2] *Text added.*

[c3] *Text added.*

[c4] *Text added.*

[c5] its

[c6] *Text added.*

[c7] time-span

Here we use atmospheric pressure data from the automatic weather station 'Marilyn' which is located about 120 km away on the Ross Ice Shelf to correct for the inverse barometric effect. This record correlates well (Pearson's correlator 0.989) with a mean of seven barometers installed over 14 days across the Darwin Glacier during the TIDEx campaign. All tide-model predictions show a clear fortnightly occuring spring-neap tidal cycle which is superimposed by a dominant diurnal signal (Fig. 4). The approximately fortnightly occurring spring/neap tides are largely owed to the difference in wavelength between the K1 and O1 constituents. [c8]

We apply t_tide to our 16 day record of the 'Shirase' GPS to test the potential of short-term GPS surveys to improve current Antarctic [c1]tide models. [c2]The problem with using such a short window to determine a full set of tidal constituents results from the interplay of the lunar diurnal tide (K1, 23.93 h) with the solar diurnal tide (P1, 24.07 h) as they are close in frequency and P1 has an amplitude of $15 - 20$ % of K1. Without accounting for their inference, t_tide just extracts an apparent K1 from a 16 day record that is really K1+P1. As a consequence, the K1 tide from our harmonic analysis can vary by $30 - 40$ % over a 6-month period and its amplitude is only controlled by the exact time that the GPS data was acquired within the K1+P1 modulation cycle. Additionally, harmonic decomposition of GPS data is subject to inaccuracies itself with errors in both the extracted amplitudes and phases. These errors were found to be of the same magnitude as the K1+P1 inference. For this reason, we did not use inference to separate K1 and P1 (or similarly to separate the semi-diurnal S2 and K2 constituents), but perform a thorough analysis on the identified uncertainty range. While the analysis captures the dominant [c3]K1 constituent [c4]in the Ross Sea within a reasonable signal-to-noise ratio, fortnightly harmonics could not be retrieved adequately from this time series alone. The t_tide prediction is therefore the least accurate tide model and requires the largest adjustment to match DInSAR (Tab. 1). Although all the corrected tide model outputs now replicate our DInSAR measurements, their rate of tidal change is affected by the adjustment. Offsets computed for the Ross_Inv_2002 tide model are generally below 10 cm, whereas other tide models require adjustments of up to 13.3 cm (Tab. 1). This agrees well with the findings of Han et al. (2013), who find that the Ross_Inv_2002 model is the optimum tide model for the Terra Nova Bay with a 4.1 cm RMSE against 11 days of tide gauge data. We therefore choose Ross_Inv_2002 for numerical modelling purposes to minimize any effects on a viscoelastic model, but use TPXO7.2 to reconstruct vertical displacement at the times of satellite overpasses as it fits best [c5]to our GPS measurements. We refer to the Appendix for a validation of individual tide model output with GPS data from 'Shirase' (Fig. A1).

After the adjustment, modelled tidal amplitudes range from -0.966 m to 0.781 m over the whole SAR period (Fig. 4). Mean [c6]absolute residual error to the 45 DInSAR measurements at the tide-model location 'Shirase' is just [c7]7 mm (Tab. 2), [c8]which can be explained by interferogram noise. We attribute this accuracy to the exceptionally high phase coherence of the TerraSAR-
* * *
[c8] The K1 tide has a period of 23.93 hours and is the dominant tidal constituent in the Ross Sea (Padman et al., 2018).

[c1] tide-models

[c2] *Text added.*

[c3] diurnal

[c4] *Text added.*

[c5] *Text added.*

[c6] *Text added.*

[c7] 0.84 mm

[c8] which is within

X data set. The reconstruction algorithm results in 12 smooth vertical displacement maps for the times of SAR data acquisition (Fig. 6).

**3.2 Validation with GPS measurements**

We now validate these reconstructions with available field data. As both GPS records overlap with the acquisition of SAR image 11, we extract the vertical displacement along the glacier's centerline and plot the profiles against the two GPS point measurements. The GPS measurement at 'Hillary' is 0.169 m, which is close to the reconstruction of 0.156 m. The 'Shirase' GPS measurement is 0.566 m, which is slightly above the reconstruction of 0.522 m. We attribute the deviations of +1.3 and +4.4 cm, respectively, to a combination of interpolation artefacts, temporal smoothing of the GPS data and residual errors of the least-squares algorithm. The overall shape of the vertical displacement is well reproduced as observed with both GPS measurements (Fig. 7).

**3.3 Applications**

**3.3.1 Tide-model refinement**

A map of tide-deflection ratio ($\alpha$-map) can be combined with the tide model to predict an average [c1]time series of vertical displacement between the times of SAR image acquisition. With this approach, the coarse grid of [c2]traditional tide models is refined to resolve small-scale features of vertical tidal displacement throughout the embayment. The $\alpha$-value for the pixel containing the 'Hillary' GPS station is 46.06%. We use this value and linearly scale the [c3]adjusted tide-model output for the location of the 'Shirase' GPS to predict vertical tidal motion within the flexure zone. This scaling maintains the tide model's high correlation (Pearson's correlator 0.95) with the 'Shirase' record, but improves the RMSE between the TPXO7.2 output and the 'Hillary' record from 21.4 cm to 5.6 cm, which corresponds to an improvement of -74% [c4]to GPS data (Fig. 8). [c5]The primary reason for this large improvement, however, is that the tide model now takes the damping of the tidal signal by ice mechanics in the grounding zone into account.

**3.3.2 Ice [c6]heterogeneity**

With the [c7]12 reconstructed displacement maps at hand, it is now possible to perform any image combination. We mosaic the 45 double differences corresponding to DInSAR combinations (Tab. 2) [c8]to allow a more direct comparison between measured and modelled interferograms. [c9]SAR image combinations were chosen so that the loss of coherence between SAR 8 and 9
* * *
[c1]

[c2] *Text added.*

[c3] *Text added.*

[c4] *Text added.*

[c5] *Text added.*

[c6]

[c7]

[c8] *Text added.*

[c9] *Text added.*

was taken into account and that a maximum number of consecutive, double-differential interferograms was available for the least-squares fitting routine. [c10]The synthetic interferograms replicate not only simple tidal fringes as measured with DInSAR, but also show complicated viscoelastic signals within the grounding zone (Fig. 5). [c11]For an overall assessment of model performance we [c12] calculate again the misfit between each modelled and observed interferogram for every pixel, but this time after using the adjusted tide-model output. The standard deviation of these misfits is shown in Fig. 9, with the majority of the glacier surface below noise level of interferograms ($\sigma < 1.0$ cm). We identify a narrow band with higher standard deviations ($\sigma \approx 2.0$ cm) from the inner shear margin of the Darwin Glacier extending along [c13]flow direction onto its ice shelf. Standard deviations are largest out [c14]in the shear zone of the fast-flowing Ross Ice Shelf [c15]and above steep rocky cliffs ($\sigma > 4.0$ cm) [c16]which is a result of poor [c17]phase coherence or layovers in SAR images in [c18]these areas .

**3.3.3 Detection of errors in phase unwrapping**

We now [c1]extend the earlier one dimensional analysis of the SMIS to a two dimensional re-analysis of the SMIS data set and calculate misfits of 9 DInSAR interferograms. Resulting standard deviations are generally smaller in this area ($\sigma < 0.3$ cm) and smoothly distributed throughout the map. We identify two regions of [c2]phase discontinuities between adjacent cells at the SMIS. Both extend from the center of an ice rise towards [c3]Black Island (cyan and green areas in Fig. 9) with $\sigma \approx 0.4$ cm and $\sigma \approx 0.5$ cm, respectively. We interpret these rapid increases as a proxy for errors in the DInSAR measurements, as the modelled least-square interferograms originate from a curvature-minimizing polynomial interpolation. We re-evaluate the remote-sensing part of the analysis and find discontinuities in DInSAR interferograms ID:1 and ID:8 that match the course of the two [c4]phase jumps in the standard deviation map. These discontinuities, in turn, [c5]occurred during phase unwrapping [c6] [c7]and can now be corrected.

**3.4 Finite-element modelling of viscoelasticity**

We hypothesize that any non-linear signal due to viscoelastic ice properties is significantly reduced or even completely lost during the averaging step to compute the $\alpha$-map. This signal can then be reconstructed by finding the offsets to match obser-
* * *
[c10] *Text added.*

[c11] *Text added.*

[c12]

[c13]

[c14]

[c15] *Text added.*

[c16]

[c17]

[c18]

[c1]

[c2]

[c3]

[c4] *Text added.*

[c5]

[c6]

[c7] *Text added.*

vations made with DInSAR. We therefore subtract the $\alpha$-prediction again from the 12 reconstructions to extract the theorised viscoelastic signal (Fig. 10). This signal is negligible at times during neap tide (SAR 4 and 9) but well pronounced for SAR images acquired during spring-tide periods (SAR 1,6,7,10 and 11).

In order to further explore this pattern, we now make use of the tiltmeter array and ApRES network of ice-thickness mea-
5    surements at the Darwin Glacier (Fig. 12). We match the numerical solutions from two finite-element models to seven tiltmeter records, with the goal to derive information on the physical properties of Antarctic ice. Thereby, the Young's modulus, $E$, is a measure of ice stiffness and controls the width of the flexure region. The value for ice viscosity, $\nu$, influences the timing of the flexural response within the grounding zone (Wild et al., 2017). Two numerical models of ice-shelf flexure are employed. The elastic approximation (Holdsworth, 1969; Vaughan, 1995; Schmeltz et al., 2002) as formulated by Walker et al. (2013):

$$\quad kw + \nabla^2(D\nabla^2 w) = q, \tag{1}$$

where $w(t)$ is the [c1]time-dependent vertical deflection of the neutral layer in a plate, $\nabla^2$ is the Laplace operator in 2-D space and $k = 5$ MPa m$^{-1}$ a spring constant of the foundation which is zero for the floating part. The applied tidal force $q(t)$ is defined by:

$$q = \rho_{sw} g[A(t) - w], \tag{2}$$

15    with $g = 9.81$ m s$^{-2}$ the gravitational acceleration and $A(t)$ the [c2]time-dependent tidal amplitude given by the adjusted Ross_Inv_2002 tide model. We choose this model, in contrast to the TPXO7.2 model, for finite-element simulations to minimize any potential effects of tide-model adjustment on viscoelasticity (Tab. 1). The stiffness of the ice shelf is given by (Love, 1906, p. 443):

$$D = \frac{EH^3}{12(1 - \lambda^2)}, \tag{3}$$

20    where $E$ is the Young's modulus for ice, $H(x,y)$ our ice thickness map derived from ApRES point measurements and $\lambda = 0.4$ the Poisson's ratio. We compare the elastic model with the viscoelastic approach developed by Walker et al. (2013):

$$\frac{\partial}{\partial t}\left[kw + \nabla^2\left(D\nabla^2 w\right)\right] + \frac{Ek}{2\nu(1 - \lambda^2)}w = \frac{\partial}{\partial t}q + \frac{E}{2\nu(1 - \lambda^2)}q, \tag{4}$$

where $\nu$ is ice viscosity. The following boundary conditions are applied for both models: the upstream boundary of the model domains on the grounded portion are anchored rigidly ($w = 0, \nabla^2 w = 0$), the downstream boundaries on the freely-floating
25    ice shelf are set free. The location of the tide model computation is constrained to be equal to the tidal oscillation ($w = A(t), \nabla w = 0$) and the grounding line is represented by a fulcrum ($w = 0$). Both models are implemented in two spatial dimensions to capture effects of complex grounding-line configuration on ice-shelf flexure (Wild et al., 2018). We then solve the models using the commercial finite-element software COMSOL Multiphysics. As tiltmeters measure slope, $w'$, along their longitudinal axis, we derive the models' solutions for vertical displacement, $w$, with respect to the $x$ and $y$ directions.
* * *
[c1] time-dependant
[c2] time-dependant

This allows us to retrieve surface slopes components in easting and northing direction and to rotate them into the individual orientations of the tiltmeter sensors. With our 16 day tiltmeter records it is only possible to capture their diurnal [c3] components with confidence. [c4]Semi-diurnal, fortnightly and monthly harmonics have been removed from the tiltmeter time series and we focus further analysis only on the K1 component [c5]within the 16 day window. Therefore, we now make extensive use of the

5 t_tide program to automatically extract the modelled K1 harmonics from the modelled surface slopes and compare them against the K1 constituents from the tiltmeters. [c6]We thereby take amplitude and phase errors that originate from the harmonic analysis of noisy tiltmeter records into account and find the best rheological parameters to match the elastic and viscoelastic models to these seven K1 components. Incorporating viscoelastic effects into the model simulations always improves the elastic fit to the tiltmeter data within the uncertainty range of K1 amplitude and phase (Tab. 3). [c7]We find that an average Young's modulus

10 of $E = 1.0 \pm 0.56$ GPa and an ice viscosity value of $\nu = 10 \pm 3.65$ TPa s fits best to our measurements within uncertainty (Fig. 13). The viscoelastic model gives an average RMSE of 0.00118845 ° to the seven tiltmeters and improves on the elastic approximation with an average RMSE of 0.00147136 ° by $\approx -20\%$.

**4   Discussion**

**4.1   Seasonal bias in $\alpha$-map**

15 Due to the alignment of the satellite overpasses with the dominant diurnal tidal constituents in the Ross Sea, the observed stage of the tidal oscillation varies only slowly throughout the year. In the austral winter months, [c1]TerraSAR-X images [c2]have been acquired during stages of low tide, whereas satellite overpasses concur with stages of high tide during the austral summer months. The first 8 snapshots of our [c3]SAR data acquisitions for the Darwin Glacier show conditions at low tide and only the last 4 are acquired during high tide. Our $\alpha$-map, in turn, ignores this seasonality and may therefore have a low-tide bias. As a

20 result, the contribution of a tide induced landward migration of the grounding line may be affected by the averaging process. The seasonal bias would then modify the scaling of the [c4]tide model within the flexure zone. This [c5] is supported by the finding that low-tide stages in the 'Hillary' GPS record are matched closely by the scaled tide model, but peaks during high-tide stages are still over estimated (Fig. 4)
* * *
[c3]
[c4]
[c5] *Text added.*
[c6] *Text added.*
[c7]
[c1]
[c2]
[c3]
[c4]
[c5]

**4.2 Viscoelasticity between snapshots**

Similarly, the linear scaling using an $\alpha$-map only modifies the predicted tidal amplitude, but neglects a viscoelastic time delay in the flexural response towards the grounding line. Wild et al. (2017) found that viscosity is most pronounced in the diurnal tidal components. Harmonic analysis of our GPS records reveals that the diurnal K1 and O1 constituents at 'Hillary' are lagging approximately 20 mins behind 'Shirase'. This signal is currently disregarded in the scaling work flow as ice is treated as a perfect elastic material that transfers tidal [c6]forcing instantaneously in the flexure zone. This assumption, however, allowed us to improve the accuracy of the [c7]predicted displacement by -74%. Currently, the viscoelastic signal can only be reconstructed for the times of SAR data acquisition. Including viscoelasticity between times of satellite overpasses [c8]offers a small, but systematic, opportunity for further refinement. [c9]In our study area, the rate of tidal change is up to 10 cm hr$^{-1}$ (Tab. 1) [c10]and the viscoelastic misfit corresponding to 20 minutes time delay is therefore up to about 3 cm.

When separating the viscoelastic contribution from the reconstructed maps of vertical displacement at times of satellite overpasses, we assume that an $\alpha$-prediction corresponds to an instantaneous elastic response. This is justified by viscoelasticity being most pronounced when rates of tidal change are maximal. [c1]By expressing the viscoelastic misfits in percent of prevailing tidal amplitude during the times of satellite overpasses, the areas of pronounced viscoelastic effects can be visualised. They are most pronounced within the Darwin Glacier's shear zone (Fig. 11). [c2]

When predicting rates of tidal change using the adjusted Ross_Inv_2002 tide model, we identify a threshold of $\dot{A} \approx \pm 0.05$ m h$^{-1}$ (SAR times 1, 2, 7, 8, 10, 11 in Tab. 1) above which viscoelasticity is well represented in the reconstructed vertical displacement maps (panels 1, 6, 7, 8, 10 and 11 in Fig. 10). Image 6 is thereby an exception, as the used Ross_Inv_2002 tide model was [c3]largely adjusted (-0.082 m), which affects the viscoelastic model. [c4]We find that the error due to viscoelasticity on the floating part of the ice shelf increases with the absolute rate of tidal change (Fig. 11). [c5]SAR images acquired during periods of spring tides at the Darwin Glacier show a significant viscoelastic contribution that diminishes during neap tide periods. These independent observations [c7]from satellite data alone support our suggested threshold of $\pm 0.05$ m h$^{-1}$ for the separation of elastic and viscoelastic signals, as derived from tiltmeter data on the Southern McMurdo Ice Shelf presented in an earlier study (Fig. 8 in Wild et al., 2017). The advantage of separating the elastic from the viscoelastic contribution to the tidal flexure pattern is the large potential for improving current inverse modelling techniques to determine grounding-zone ice thickness from DInSAR measurements alone. Hereby, an elastic model is currently employed to optimize grounding-zone ice thickness
* * *
[c6]

[c7]

[c8]

[c9] *Text added.*

[c10] *Text added.*

[c1] *Text added.*

[c2]

[c3]

[c4] *Text added.*

[c5] *Text added.*

[c7] *Text added.*

to match the surface flexure from DInSAR. [c8] The applicability of an elastic model varies from location to location as effective viscosity is dependent on ice temperature and shear stress (Marsh et al., 2014). Our method to separate the two contributions to the flexure pattern may therefore help to remove the viscoelastic contamination and allow purely elastic inverse modelling. [c9] [c10]Such an analysis, however, goes beyond the scope of this manuscript and will be published elsewhere.

**4.3  Large-scale ice anisotropy**

Fast-moving glacial environments like the Darwin Glacier are subject to large deformation by flow convergence and divergence, ice compression and extension, lateral shearing at the margins accompanied by fracture under tension and rapid thinning [c1]by basal melt. With cumulative deformation, a crystallographic fabric evolves that reflects the glacier's flow history (Alley, 1988), and with it strain-dependent mechanical anisotropy of ice. The standard deviation map, Fig. 9, shows a narrow band of larger misfits extending from the Darwin Glacier's inner shear margin out towards the freely-floating ice shelf. As preferred crystallographic orientation develops with strain, effective viscosity decreases of about a factor of ten compared to initially isotropic polycrystalline ice (Hudleston, 2015). Our analysis of tiltmeter data reveals a five-fold reduced viscosity at the very dynamic Darwin Glacier compared to an earlier study at the almost stagnant Southern McMurdo Ice Shelf (Wild et al., 2017). We [c2]hypothesize that this microscopic process explains the macroscopic response observed here, and accounts for the measured glacial [c3]heterogeneity within the embayment. Large scale observations of ice anisotropy, in turn, are currently the missing key to improve parametrisations to account for polar ice anisotropy in ice-sheet flow modelling (Gagliardini et al., 2009). [c4]Other processes have been proposed which lead to ice softening in areas with high strain rates. Thermomechanical modelling suggests that shear heating and consequent thermal softening reduces lateral drag in ice-stream margins (Perol and Rice, 2015). [c5]Fracture modelling implies that damage reduces ice viscosity along confined crevassed zones with consequences on ice-shelf scale (Albrecht and Levermann, 2014). [c6]Full-Stokes viscoelastic modelling shows that Glen's non-linear flow law and tidal stresses in the ice-shelf flexure zone are sufficient to explain large-scale temporal variations in ice dynamics (Rosier and Gudmundsson, 2018). [c7]These processes, or a combination of them, might certainly be at play but they do not explain why a band of higher standard deviations can be observed in the shear zone of the Darwin Glacier which is absent in the flexure zone of the SMIS (Fig. 9). [c8]We therefore attribute this difference to ice-fabric reorientation in the shear margin.
* * *
[c8] ~~This is because an elastic model for tidal flexure is only forced by the 'apparent' tidal amplitude (Eq.), which can be measured directly from the interferogram on the freely-floating area. A viscoelastic model additionally incorporates the time derivative of the tidal forcing (Eq.) and hence captures the rate of tidal change. This information, in turn, can not be deduced directly from the interferogram which makes the usage of auxiliary tide models inevitable. Tide models, however, have shown to be prone to large inaccuracies around Antarctica making a successful inversion of viscoelastic flexure models highly elusive.~~

[c9]

[c10]

[c1]

[c2]

[c3]

[c4] *Text added.*

[c5] *Text added.*

[c6] *Text added.*

[c7] *Text added.*

[c8] *Text added.*

**4.4 Refining tidal constituents using DInSAR**

[c1]The idea of using SAR interferometry to derive a full set of tidal harmonics was first laid out in a study of tides in the Weddell Sea (Rignot et al., 2000). [c2]The authors discussed that DInSAR images cannot be transformed into individual displacement fields because of the nonuniqueness of the inversion. A large number of independent DInSAR images is required to overcome this problem and to resolve the phase of tidal constituents that are close to the repeat-pass of the SAR satellite. For example, multiples of the lunar diurnal constituent K1 (23.93 h) are relatively close to the exact integer repeat-pass of TerraSAR-X (11 days) meaning that the observed amplitude of the K1 constituent is only varying once throughout the year. Consequently SAR images need to be acquired at least over the duration of one year to provide some redundancy for the inversion step of DInSAR images to tidal constituents. However, with an exact 12 h period, the stage of the semidiurnal solar tide, S2, will always be the same at each satellite pass making TerraSAR-X and similar satellites with repeat-passes of integer days blind to the S2 constituent. [c3]For example Minchew et al. (2017) [c4]needed a unique spatially and temporally dense SAR acquisition campaign as well as *a-priori* knowledge of the temporal basis functions from GPS data to empirically determine tidal constituents on Rutford Ice Stream. The four COSMO-SkyMed satellites in orbit, however, produce repeat-passes of 1, 3, 4 and 8 days and are blind to the S2 constituent as well even when using > 1000 available DInSAR images. Although other dominant tidal constituents like M2 (12.4 h) and O1 (25.82 h) were inferred successfully, the method presented here can achieve a higher accuracy fo the total tide with less DInSAR images. From another perspective, the inclusion of an auxiliary tide model eases the requirement of a very large number of DInSAR images.

**5 Conclusions and Outlook**

[c5]Accurate prediction of ocean tides in coastal areas is crucial as the majority of Antarctica's ice is discharged through large outlet glaciers. [c6]We presented a data fusion method between DInSAR and traditional Antarctic tide models to predict spatial variability of tidal motion near the grounding line. The primary value of using DInSAR in conjunction with tide models lies in the spatio-temporal benefits of resolving complex grounding zone deformation. Their symbiosis not only improves current accuracies of the predicted tidal amplitudes in coastal regions generally, but also avoids issues related to the timing of the tidal wave and the sun-synchronous satellite orbit when attempting to derive tide-models from SAR data alone. [c7]In our study area, the method presented in this paper improves traditional tide modelling in average by -22% from 11.8 cm to 9.3 cm RMSE against 16 days of GPS data. The GPS station 'Shirase' on the freely-floating part of the Darwin Glacier has proven invaluable
* * *
[c1] *Text added.*

[c2] *Text added.*

[c3] *Text added.*

[c4] *Text added.*

[c5] *Text added.*

[c6]

[c7] *Text added.*

to determine which [c8]tide model has to be used to best reconstruct the vertical displacement during satellite overpasses. For the Darwin Glacier, the TPXO7.2 tide model predicts best the tidal oscillation. With using DInSAR measurements to adjust the TPXO7.2 tidal prediction, its RMSE could be improved by -39% from 10.8 cm to 6.7 cm. [c9]

Our GPS record from 'Shirase' is too short to [c1]improve already available Antarctic tide models. A longer record is required to adequately resolve a full set of tidal constituents. [c2] [c3]We produced an empirical displacement map from DInSAR for tidal deflection ($\alpha$-map). Comparison of [c4]a GPS record within the tidal flexure zone 'Hillary' with predicted vertical displacement from feeding the $\alpha$-map with the adjusted TPXO7.2 tide model shows a -74% improvement over using the tide-model output alone. This independent validation supports the finding that [c5]our method for making use of DInSAR is very useful for refining tide models in Antarctic [c6]grounding zones. [c7]

Numerical modelling of ice dynamics in Antarctic grounding zones commonly assumes that ice is isotropic and homogeneous i.e. of [c8]the same density and rheological properties throughout. Our analysis reveals that this assumption is valid for the Southern McMurdo Ice Shelf, an almost stagnant area with a simple grounding line [c9]configuration, but invalid for the Darwin Glacier, a fast-flowing outlet glacier with complex shear margins causing non-negligible ice [c10]heterogeneity within the embayment.

Further work is required [c11]in order to improve tide models in a larger variety of grounding zones by including effects of grounding-line migration and variability of horizontal ice flow.

*Code availability.* The code is freely available to the scientific community. Collaboration is anticipated and desired.

*Data availability.* TerraSAR-X data presented in this paper are subject to license agreements. GPS/tiltmeter and ApRES data are available upon request.
* * *
[c8]

[c9]

[c1]

[c2]

[c3] *Text added.*

[c4]

[c5] *Text added.*

[c6]

[c7]

[c8] *Text added.*

[c9] *Text added.*

[c10]

[c11]

*Code and data availability.* No data sets, nor software, are part of this study.

*Sample availability.* No samples were collected for this study.

**Appendix A: GPS evaluation of tide models**

The quality of the used [c1]tide model to correctly reconstruct tidal displacement at the times of SAR data acquisitions, is also crucial to accurately predict spatial variability in tidal motion for all times. Here, we assume that a freely-floating area on the [c2]ice shelf experiences the full oscillation as predicted from a tide model. In this area, however, tide-model output deviates from our DInSAR measurements. This indicates either that the area under investigation is prevented from a freely-floating state by lateral stresses within the embayment, or that the tide-model prediction is inaccurate for this area. We circumvent this ambiguity by making use of the high vertical accuracy of DInSAR and correct the tide-model prediction to match our satellite measurements. This raises the question of whether the adjustment improves or worsens the match to a 'real' tidal motion ? We therefore independently evaluate the pre- and post adjustment tide-model predictions and calculate their RMSE to 16 days of GPS data from the freely-floating area (Tab. A1). The adjustment improves all traditional tide model predictions by up to -39% for TPXO7.2, and only worsens the RMSE for the t_tide output by +11%, indicating that a harmonic analysis of GPS data can not be improved by using DInSAR for correction purposes. We choose TPXO7.2 for further processing as it displays the overall smallest RMSE (6.7 cm) and replicates the small-scale variability observed during the neap-tide period in the second half of our GPS record.

*Author contributions.* All authors conceived the study and conducted fieldwork. CW developed the algorithm, performed the data analysis and wrote a first version of the paper. OM and WR processed the SAR images for the Darwin Glacier and Southern McMurdo Ice Shelf, respectively. All authors finalized and approved the manuscript.

*Competing interests.* The authors declare that the research was conducted in the absence of any commercial or financial relationships that could be construed as a potential conflict of interest.

*Disclaimer.*
* * *
[c1] tide-model

[c2] ice-shelf

*Acknowledgements.* We thank Antarctica New Zealand for logistical support and the Scott Base staff for their dedication to the Transantarctic Ice Deflection Experiment. The project was supported by the Royal Geographic Society (Marsden Fast Start, PI O. Marsh) and by the Past Antarctic Science and Future Implications Program (PACaFI). D. Price, M. Ryan, D. Floricioiu and E. Scheffler contributed to fieldwork. TerraSAR-X data were provided through DLR project HYD1421. Landsat-8 images courtesy of the US Geological Survey. AWS data from Scott Base were provided by NIWA, and for Marilyn station by AMRC, SSEC, UW-Madison. The authors enjoyed fruitful discussions with H. Purdie and M. Sellier. CW also thanks R. Mueller for sharing oceanographic expertise. The thorough comments of one anonymous reviewer and L. Padman's valuable insight into tidal dynamics largely improved the paper. We also thank A. Robinson for editing.

[revised manuscript text omitted]

---

## Referee Report (RR1)

```
%% Copernicus Publications Manuscript Preparation Template for LaTeX
Submissions
%% ---------------------------------
%% This template should be used for copernicus.cls
%% The class file and some style files are bundled in the Copernicus
Latex Package, which can be downloaded from the different journal
webpages.
%% For further assistance please contact Copernicus Publications at:
production@copernicus.org
%%
https://publications.copernicus.org/for_authors/manuscript_preparation.ht
ml

%% Please use the following documentclass and journal abbreviations for
discussion papers and final revised papers.

%% 2-column papers and discussion papers
\documentclass[tc, manuscript]{copernicus}

\graphicspath{{./Figures/}}
\usepackage[finalnew]{trackchanges}
%finalold  - Reject all edits.
%finalnew  - Accept all edits.
%footnotes - Display edits as footnotes.
%margins   - Display edits as margin notes.
%inline    - Display edits inline.

\usepackage{booktabs}

\begin{document}

\title{Differential InSAR for tide modelling in Antarctic ice-shelf
grounding zones}

% \Author[affil]{given_name}{surname}

\Author[1]{Christian T.}{Wild}
\Author[1,2]{Oliver J.}{Marsh}
\Author[1]{Wolfgang}{Rack}

\affil[1]{Gateway Antarctica, University of Canterbury, Private Bag 4800,
Christchurch 8140, New Zealand}
\affil[2]{British Antarctic Survey, High Cross, Madingley Road,
Cambridge, CB3 0ET, United Kingdom}

%% The [] brackets identify the author with the corresponding
affiliation. 1, 2, 3, etc. should be inserted.

\runningtitle{Differential InSAR and tide modelling}

\runningauthor{Wild et al.}
```

\correspondence{Christian T. Wild (christian.wild@canterbury.ac.nz)}

\received{}
\pubdiscuss{} %% only important for two-stage journals
\revised{}
\accepted{}
\published{}

%% These dates will be inserted by Copernicus Publications during the
typesetting process.

\firstpage{1}

\maketitle

[revised manuscript text omitted]

\section{Methodology}
\label{Meths}

\subsection{Summary of SAR image processing}
To develop the method, we use 11-day repeat-pass TerraSAR-X data in StripMap imaging mode. The satellite acquires X-band radar data (wavelength 3.1 cm, frequency 9.6 GHz) with a ground resolution of slightly below 3x3~m and images covering an area of 30x50 km. We calculate vertical surface displacement due to ocean tides using the Gamma software package \citep{werner2000gamma}. \add{InSAR and DInSAR image combinations are generally chosen so that a later image is always subtracted from an earlier image. For image triplets, the central SAR image serves as a common reference during the co-registration.} We then correct the resulting DInSAR interferograms for apparent vertical

Commented [L12]: "in turn" isn't correct here.

Commented [L13]: poorly written

Commented [L14]: 1) what are the different dimensions? one along-track and the other cross-track?

2) are these image sizes just the way the along-track swath data are packaged, or what you can get for a target or opportunity, or ???

displacement due to horizontal surface motion \citep{rack2017analysis}
using the method presented  by \cite{wild2018unraveling}.

\subsection{Tide models}
The predictions of five tide models are validated: the regional
barotropic models (1) Circum-Antarctic Tidal Solution (CATS2008)
developed by \cite{padman2008improving}, (2) Ross Sea Height-Based Tidal
Inverse Model (Ross\_Inv\_2002) developed by \cite{padman2003tides}, (3)
Ross Sea assimilation model (Ross\_VMADCP\_9cm), (4) the fully global
barotropic assimilation model (TPXO7.2) from Oregon State University
developed by \cite{egbert2002efficient}, and (5) the (t\_tide) prediction
of GPS data from freely-floating areas following the harmonic analysis of
\cite{pawlowicz2002classical} \add{which is based on} FROTRAN codes
developed by \cite{foreman1977manual}. The t\_tide software is a widely-
used toolbox for performing classical harmonic analysis of ocean tides.
It can be sued to analyse any time series record and outputs the
amplitude and phase of its dominant harmonics \add{with error estimates},
along with a tidal prediction that is free from non-tidal effects. The
isostatic deformation of the Earth's lithosphere underneath the moving
water masses is modelled using TPXO7.2 load tide model
\citep{egbert2002efficient}, which itself is based on 13 tidal
constituents and added to all tide model predictions except t\_tide. In
addition to the tidal motion underneath the floating ice, much of the
ice-surface variability can be attributed to the Inverse Barometric
Effect \cite[IBE,][]{padman2003tides}. A +1~hPa anomaly of atmospheric
pressure translates to \change{a}{an instantaneous} -1~cm
\change{drop}{change} on the ice-shelf surface. \add{~~It is noteworthy to
mention~~Note that we did not apply a running mean to the pressure records,
as the application of any window length worsened the fit to available GPS
data.} To correct for the IBE \add{outside the GPS period}, we make use
of atmospheric pressure records obtained by nearby automatic weather
stations on Ross Island (Scott Base AWS) and the Ross Ice Shelf (Marilyn
AWS). We validate these records with separate barometric measurements
taken within the TIDEx campaign and find very good agreement.

\add{In this paper, we use the terms 'traditional tide modelling' or
'tide model' to refer to the sum of ocean-tide, load-tide model outputs
and the IBE. Freely-floating areas of ice shelves and glaciers are
expected to experience the full oscillation of this tide model.
Traditional tide modelling, however, neglects ice mechanics in grounding
zones where tidal flexure significantly affects the surface elevation
signal in reality. Other signals that change sea-level height such as
mean dynamic topography and storm surges are also excluded from this type
of tide model.}

\subsection{In-situ data}
We set up a number of GPS receivers to measure ice-surface motion at
millimetre accuracy and high temporal resolution. \change{Here we
use}{Although we used} GPS data from the freely-floating parts \change{of
the ice surface and}{to} develop local tide models using t\_tide,
\add{all GPS data was only used for validation purposes and did not feed
into the algorithm}. \remove{GPS measurements from within the tidal
flexure regions are only used as validation data sets.} \remove{All} GPS
measurements were differentially corrected using static base stations

\add{to increase their spatial accuracy}. We also
\change{install}{installed} an array of seven tiltmeters
\change{recording}{to record} surface flexure over 16 days across the
grounding zone to confine the physical properties of Antarctic ice.
\change{This is}{The tiltmeters were} complemented by a dense network of
point measurements of ice thickness using a the new autonomous phase-
sensitive radar echo sounder \cite[ApRES,][]{nicholls2015ground}.

\subsection{Combining DInSAR and tide models}
\change{A tide model must perfectly predict the}{To allow a correct
interpretation of DInSAR images covering grounding zones, it is desirable
that tide models replicate} DInSAR observations \change{in an area that
can be expected to experience the full tidal forcing.}{in freely-floating
areas.} We first adjust the tide-model output to match the highly-
accurate DInSAR measurements using a least sum of squares routine
\citep{wild2018unraveling}. \add{By doing so, we consider just the tide
model amplitude to contain errors. Possible tide-model phase errors are
then accounted for by adjusting the absolute amplitude and thus the rate
of tidal change during the times of SAR data acquisition.} Second we
build on earlier work by \cite{han2014tide} and to develop an empirical
displacement map showing tide-deflection ratio throughout the satellite
image ($\alpha$-map). By feeding the $\alpha$-map with the adjusted tide
model output, the \change{'point-forecast'}{'point forecast'} is then
spatially extended to predict the mean vertical displacement for every
pixel at the times of SAR data acquisition. We then perform the
\change{double-differences}{double differences} of the empirical model
corresponding to the SAR image combinations used to generate the DInSAR
images. The original DInSAR satellite measurements are subsequently
removed from the mean DInSAR images to calculate their misfits, $\mu$. We
now compute the least-squares solution to the equation $Ax=b$ such that
the 2-norm $|b-Ax|$ is minimized. Here, $A$ is the $m\times n$ DInSAR
matrix of SAR image combinations with $m$ rows of SAR images and $n$
columns of coherent DInSAR interferograms; $b$ is a vector of $\alpha$-
prediction misfits and $x$ the least-squares solution of this
\change{over-determined system}{system of linear equations}. The values
of $x$ correspond to how much an $\alpha$-prediction deviates from the
'real' vertical displacement at the times of SAR data acquisition. We
therefore subtract these offsets, \change{$\theta$}{$\Delta A$}, from the
$\alpha$-prediction maps.

We \add{now} demonstrate the workflow in one spatial dimension with an
example of the Southern McMurdo Ice Shelf ($78^{\circ}15\textrm{' S}$,
$167^{\circ}7\textrm{' E}$, SMIS). In this study area, we derived 9
DInSAR images from 12 TerraSAR-X scenes in 2014 \citep{rack2017analysis}.
\add{The low number of DInSAR images is a consequence of the SAR scenes
being acquired on 3 different satellite tracks. The resulting system of
linear equations is therefore underdetermined as there are more offsets,
$\Delta A$, than misfits, $\mu$, to constrain the least-squares
solutions.} We choose a pixel on the freely-floating end of a profile
through the ice-shelf grounding zone to represent the unrestricted ice
shelf movement and calculate the percentage vertical displacement of
every other pixel from this location. Averaged over the 9 DInSAR
interferograms this pixel retains 100\% vertical displacement (red areas
in Fig. \ref{fig:alpha_maps}), while grounded areas experience zero net

uplift (purple areas). Individual pixels on the freely-floating part of the SMIS may show $\alpha$-values slightly above 100\%.

We now extract the $\alpha$-values along the profile from the $\alpha$-map. This $\alpha$-profile can be multiplied with the individual DInSAR measurements on its freely-floating end which results in empirically derived $\alpha$-predictions (Fig. \ref{fig:ls_process_DInSAR_SMIS} center). These mean predictions do not perfectly replicate the DInSAR measurements (Fig. \ref{fig:ls_process_DInSAR_SMIS} top). Their misfits, however, show a very systematic pattern (Fig. \ref{fig:ls_process_DInSAR_SMIS} bottom). It is desirable to find a combination of offsets that have the least deviation from the $\alpha$-predictions. We therefore hypothesize that this rather systematic signal can be reconstructed using a least-squares strategy. \remove{Here, the linear system is under-determined with 9 DInSAR equations and 12 unknown SAR offsets.} We solve the \add{underdetermined} system simultaneously by finding the combination of offsets that result in a minimal sum of squares. The reconstructed offsets must then be removed from the $\alpha$-prediction for the times of SAR data acquisition (Fig. \ref{fig:ls_process_SAR_SMIS} top). The computed least-square offsets generally replicate the pattern of the misfits (Fig. \ref{fig:ls_process_SAR_SMIS} center) and result in smooth displacement profiles in the ice-shelf grounding zone (Fig. \ref{fig:ls_process_SAR_SMIS}).

\section{Results}
\label{Res}
In this section we apply the workflow in two spatial dimensions to the Darwin Glacier ($79^{\circ}53\textrm{' S}, 159^{\circ}00\textrm{' E}$). \add{In this study area, we derived a total of 45 DInSAR images from 12 SAR scenes being acquired on the same satellite track. SAR image combinations were generally chosen consecutively so that a later image is always subtracted from an earlier image. For image triples, the central image was taken as a common reference/master image. Additionally the data gap between SAR 8 and 9 was taken into account (no 8-9 combination as loss of coherence over this relatively long interval). The advantage of using every other remaining combination} (Tab. \ref{tab:DInSAR_table}) \add{is that more double-differential measurements of tidal amplitude are available for the least-squares fitting algorithm than only using consecutive pairs alone. The system of linear equations is then overdetermined.} A dedicated field campaign was conducted in \change{its}{the Darwin Glacier} grounding zone in 2016 and \textit{in-situ} data is available \add{for numerical modelling and field validation purposes}. In contrast to the simple geometry at the SMIS, the Darwin Glacier consists of a feature-rich embayment that is constrained by steep topography at its margins. Additionally a buttressing ice rise to the Ross Ice Shelf restricts outflow in the North.

\subsection{Reconstruction of displacement maps during satellite overpasses}
From the interferogram dataset we identify a corridor of only about 2~km width along the centerline where the glacier can be assumed to be freely floating (Fig. \ref{fig:alpha_maps}). This area is expected to experience the full oscillation predicted from tide models. We run five tide models

to predict the tidal oscillation at the GPS station 'Shirase' over the \change{time-span}{time span} of SAR data acquisitions. Here we use atmospheric pressure data from the automatic weather station 'Marilyn' which is located about 120~km away on the Ross Ice Shelf to correct for the inverse barometric effect. This record correlates well (Pearson's correlator 0.989) with a mean of seven barometers installed over 14 days across the Darwin Glacier during the TIDEx campaign. All tide-model predictions show a clear fortnightly occuring spring-neap tidal cycle which is superimposed by a dominant diurnal signal (Fig. \ref{fig:forcing_Darwin}). The approximately fortnightly  spring/neap tidal cycle is primarily determined byes  the difference in  frequency between the dominant K1 and O1 diurnal constituents. \remove{The K1 tide has a period of 23.93~hours and is the dominant tidal constituent in the Ross Sea (Padman et al., 2018).}

We apply t\_tide to our 16~day record of the 'Shirase' GPS to test the potential of short-term GPS surveys to improve current Antarctic \change{tide-models}{tide models}. \add{The problem with using such a short window to determine a full set of tidal constituents results from the interplay of the lunar diurnal tide (K1, 23.93~h) with the solar diurnal tide (P1, 24.07~h) as they are close in frequency and P1 has an amplitude of $15-20$~\% of K1. Without accounting for their inference, t\_tide  just extracts an apparent K1 from a 16~day record that is really K1+P1. As a consequence, the K1 tide from our harmonic analysis can vary by $30-40$~\% over a 6-month period and its amplitude is only controlled by the exact time that the GPS data was acquired within the K1+P1 modulation cycle. Additionally, harmonic decomposition of GPS data is subject to inaccuracies itself with errors in both the extracted amplitudes and phases. These errors were found to be of the same magnitude as the K1+P1 inference. For this reason, we did not use inference to separate K1 and P1 (or similarly to separate the semi-diurnal S2 and K2 constituents), but perform a thorough analysis on the identified uncertainty range.} While the analysis captures the dominant \change{diurnal}{K1} constituent \add{in the Ross Sea within a reasonable signal-to-noise ratio}, fortnightly harmonics could not be retrieved adequately from this time series alone. The t\_tide prediction is therefore the least accurate tide model and requires the largest adjustment to match DInSAR (Tab. \ref{tab:SAR_table}). Although all the corrected tide model outputs now replicate our DInSAR measurements, their rate of tidal change is affected by the adjustment. Offsets computed for the Ross\_Inv\_2002 tide model are generally below 10~cm, whereas other tide models require adjustments of up to 13.3~cm (Tab. \ref{tab:SAR_table}). This agrees well with the findings of \cite{han2013accuracy}, who find that the Ross\_Inv\_2002 model is the optimum tide model for the Terra Nova Bay with a 4.1~cm RMSE against 11 days of tide gauge data. We therefore choose Ross\_Inv\_2002 for numerical modelling purposes to minimize any effects on a viscoelastic model, but use TPXO7.2 to reconstruct vertical displacement at the times of satellite overpasses as it fits best \add{to} our GPS measurements. We refer to the Appendix for a validation of individual tide model output with GPS data from 'Shirase' (Fig. \ref{fig:GPS_validation}).

After the adjustment, modelled tidal amplitudes range from -0.966~m to 0.781~m over the whole SAR period (Fig. \ref{fig:forcing_Darwin}). Mean

\add{absolute} residual error to the 45 DInSAR measurements at the tide-model location 'Shirase' is just \change{0.84~mm}{7~mm} (Tab. \ref{tab:DInSAR_table}), \change{which is within}{which can be explained by} interferogram noise. We attribute this accuracy to the exceptionally high phase coherence of the TerraSAR-X data set. The reconstruction algorithm results in 12 smooth vertical displacement maps for the times of SAR data acquisition (Fig. \ref{fig:coeff_maps_Darwin}).

\subsection{Validation with GPS measurements}
We now validate these reconstructions with available field data. As both GPS records overlap with the acquisition of SAR image 11, we extract the vertical displacement along the glacier's centerline and plot the profiles against the two GPS point measurements. The GPS measurement at 'Hillary' is 0.169~m, which is close to the reconstruction of 0.156~m. The 'Shirase' GPS measurement is 0.566~m, which is slightly above the reconstruction of 0.522~m. We attribute the deviations of +1.3 and +4.4~cm, respectively, to a combination of interpolation artefacts, temporal smoothing of the GPS data and residual errors of the least-squares algorithm. The overall shape of the vertical displacement is well reproduced as observed with both GPS measurements (Fig. \ref{fig:LS_flexure_curves_Darwin}).

\subsection{Applications}
\subsubsection{Tide-model refinement}
A map of tide-deflection ratio ($\alpha$-map) can be combined with the tide model to predict an average \change{time-series}{time series} of vertical displacement between the times of SAR image acquisition. With this approach, the coarse grid of \add{traditional} tide models is refined to resolve small-scale features of vertical tidal displacement throughout the embayment. The $\alpha$-value for the pixel containing the 'Hillary' GPS station is 46.06\%. We use this value and linearly scale the \add{adjusted} tide-model output for the location of the 'Shirase' GPS to predict vertical tidal motion within the flexure zone. This scaling maintains the tide model's high correlation (Pearson's correlator 0.95) with the 'Shirase' record, but improves the RMSE between the TPXO7.2 output and the 'Hillary' record from 21.4~cm to 5.6~cm, which corresponds to an improvement of -74\% \add{to GPS data} (Fig. \ref{fig:Hillary_validation}). \add{The primary reason for this large improvement, however, is that the tide model now takes the damping of the tidal signal by ice mechanics in the grounding zone into account.}

\subsubsection{Ice \change{heterogenity}{heterogeneity}}
With the \change{twelve}{12} reconstructed displacement maps at hand, it is now possible to perform any image combination. We mosaic the 45 double differences corresponding to DInSAR combinations (Tab. \ref{tab:DInSAR_table}) \add{to allow a more direct comparison between measured and modelled interferograms}. \add{SAR image combinations were chosen so that the loss of coherence between SAR 8 and 9 was taken into account and that a maximum number of consecutive, double-differential interferograms was available for the least-squares fitting routine.} \add{The synthetic interferograms replicate not only simple tidal fringes as measured with DInSAR, but also show complicated viscoelastic signals within the grounding zone} (Fig. \ref{fig:Darwin_DInSARs}). \add{For an overall assessment of model performance we} \remove{and then} calculate

again the misfit between each modelled and observed interferogram for every pixel, but this time after using the adjusted tide-model output. The standard deviation of these misfits is shown in Fig. \ref{fig:LS_STD_maps}, with the majority of the glacier surface below noise level of interferograms ($\sigma < 1.0$~cm). We identify a narrow band with higher standard deviations ($\sigma \approx 2.0$~cm) from the inner shear margin of the Darwin Glacier extending along \change{flow-direction}{flow direction} onto its ice shelf. Standard deviations are largest out \change{on the}{in the shear zone of the fast-flowing} Ross Ice Shelf \add{and above steep rocky cliffs} ($\sigma > 4.0$~cm) \change{and a}{which is a} result of poor \change{coherence between}{phase coherence or layovers in} SAR images in \change{this area}{these areas} .

\subsubsection{Detection of errors in phase unwrapping}
We now \change{apply the algorithm to the}{extend the earlier one dimensional analysis of the SMIS to a two dimensional re-analysis of the} SMIS data set and calculate misfits of 9 DInSAR interferograms. Resulting standard deviations are generally smaller in this area ($\sigma < 0.3$~cm) and smoothly distributed throughout the map. We identify two regions of \change{jumps}{phase discontinuities} between adjacent cells at the SMIS. Both extend from the center of an ice rise towards \change{the dry land}{Black Island} (cyan and green areas in Fig. \ref{fig:LS_STD_maps}) with $\sigma \approx 0.4$~cm and $\sigma \approx 0.5$~cm, respectively. We interpret these rapid increases as a proxy for errors in the DInSAR measurements, as the modelled least-square interferograms originate from a curvature-minimizing polynomial interpolation. We re-evaluate the remote-sensing part of the analysis and find discontinuities in DInSAR interferograms ID:1 and ID:8 that match the course of the two \add{phase} jumps in the standard deviation map. These discontinuities, in turn, \change{are a result of using a minimum cost-flow algorithm on a triangular network for unwrapping}{occurred during phase unwrapping} \remove{interferometric phase differences to relative surface displacement} \add{and can now be corrected}.

\subsection{Finite-element modelling of viscoelasticity}
We hypothesize that any non-linear signal due to viscoelastic ice properties is significantly reduced or even completely lost during the averaging step to compute the $\alpha$-map. This signal can then be reconstructed by finding the offsets to match observations made with DInSAR. We therefore subtract the $\alpha$-prediction again from the 12 reconstructions to extract the theorised viscoelastic signal (Fig. \ref{fig:visco_maps_Darwin}). This signal is negligible at times during neap tide (SAR 4 and 9) but well pronounced for SAR images acquired during spring-tide periods (SAR 1,6,7,10 and 11).

In order to further explore this pattern, we now make use of the tiltmeter array and ApRES network of ice-thickness measurements at the Darwin Glacier (Fig. \ref{fig:Darwin_field_overview}). We match the numerical solutions from two finite-element models to seven tiltmeter records, with the goal to derive information on the physical properties of Antarctic ice. Thereby, the Young's modulus, $E$, is a measure of ice stiffness and controls the width of the flexure region. The value for ice viscosity, $\nu$, influences the timing of the flexural response within

Commented [L36]: not clear what counts as the "inner" shear margin.

Commented [L37]: I don't know the term "layovers" in this context.

Commented [L38]: I don't know what "increases" refers to. Sounds like time-domain, but I guess you mean "spatial gradients to the locations of maxima" or something like that.

Commented [L39]: So, we don't really have much information about this "curvature-minimizing polynomial interpolation". Can I repeat your study by following your "explanation" ?

Commented [L40]: what's the "course of the two phase jumps" ?

Commented [L41]: Note that my understanding of the ice rheology stuff is limited.

Commented [L42]: Have we seen Fig. 11 yet? Looks like you went straight from Fig. 10 to Fig. 12

the grounding zone \citep{wild2017viscosity}. Two numerical models of ice-shelf flexure are employed. The elastic approximation \citep{holdsworth1969flexure, vaughan1995tidal,schmeltz2002tidal} as formulated by \cite{walker2013ice}:
\begin{equation}\label{elastic_model}
 kw + \nabla^{2} (D \nabla^{2} w) = q,
\end{equation}
where $w(t)$ is the \change{time-dependant}{time-dependent} vertical deflection of the neutral layer in a plate, $\nabla^{2}$ is the Laplace operator in 2-D space and $k=5$~MPa~m$^{-1}$ a spring constant of the foundation which is zero for the floating part. The applied tidal force $q(t)$ is defined by:
\begin{equation}
 q = \rho_{sw} g [A(t) - w],
\end{equation}
with $g=9.81$~m~s$^{-2}$ the gravitational acceleration and $A(t)$ the \change{time-dependant}{time-dependent} tidal amplitude given by the adjusted Ross\_Inv\_2002 tide model. We choose this model, in contrast to the TPXO7.2 model, for finite-element simulations to minimize any potential effects of tide-model adjustment on viscoelasticity (Tab. \ref{tab:SAR_table}). The stiffness of the ice shelf is given by \cite[p. 443]{love1906treatise}:
\begin{equation}\label{03}
 D = \frac{E H^{3}}{12(1-\lambda^{2})},
\end{equation}
where $E$ is the Young's modulus for ice, $H(x,y)$ our ice thickness map derived from ApRES point measurements and $\lambda=0.4$ the Poisson's ratio. We compare the elastic model with the viscoelastic approach developed by \cite{walker2013ice}:
\begin{equation}\label{visco_model}
  \frac{\partial}{\partial t} \left[ kw + \nabla^{2} \left( D \nabla^{2} w \right) \right] + \frac{Ek}{2 \nu(1-\lambda^{2})} w
  = \frac{\partial}{\partial t} q + \frac{E}{2 \nu (1-\lambda^{2})} q,
\end{equation}
where $\nu$ is ice viscosity. The following boundary conditions are applied for both models: the upstream boundary of the model domains on the grounded portion are anchored rigidly ($w=0, \nabla^{2} w=0$), the downstream boundaries on the freely-floating ice shelf are set free. The location of the tide model computation is constrained to be equal to the tidal oscillation ($w=A(t), \nabla w = 0$) and the grounding line is represented by a fulcrum ($w=0$). Both models are implemented in two spatial dimensions to capture effects of complex grounding-line configuration on ice-shelf flexure \citep{wild2018unraveling}. We then solve the models using the commercial finite-element software COMSOL Multiphysics. As tiltmeters measure slope, $w'$,  along their longitudinal axis, we derive the models' solutions for vertical displacement, $w$, with respect to the $x$ and $y$ directions. This allows us to retrieve surface slopes components in easting and northing direction and to rotate them into the individual orientations of the tiltmeter sensors. With our 16~day tiltmeter records it is only possible to capture their diurnal \remove{and semi-diurnal} components with confidence. \change{Fortnightly}{Semi-diurnal, fortnightly} and monthly harmonics have been removed from the tiltmeter time series and we focus further analysis only on the K1 component \add{within the 16~day window}.

Therefore, we now make extensive use of the t\_tide program to automatically extract the modelled K1 harmonics from the modelled surface slopes and compare them against the K1 constituents from the tiltmeters. \add{We thereby take amplitude and phase errors that originate from the harmonic analysis of noisy tiltmeter records into account and find the best rheological parameters to match the elastic and viscoelastic models to these seven K1 components. Incorporating viscoelastic effects into the model simulations always improves the elastic fit to the tiltmeter data within the uncertainty range of K1 amplitude and phase} (Tab. \ref{tab:K1_table}). \change{A Young's modulus of $E=1.0$~GPa and an ice viscosity value of $\nu=10$~TPa~s fits best our measurements}{We find that an average Young's modulus of $E=1.0\pm0.56$~GPa and an ice viscosity value of $\nu=10\pm3.65$~TPa~s fits best to our measurements within uncertainty} (Fig. \ref{fig:K1_components}). The viscoelastic model gives an average RMSE of 0.00118845~$^{\circ}$ to the seven tiltmeters and improves on the elastic approximation with an average RMSE of 0.00147136~$^{\circ}$ by $\approx -20$\%.

\section{Discussion}
\label{Dis}
\subsection{Seasonal bias in $\alpha$-map}
Due to the alignment of the satellite overpasses with the dominant diurnal tidal constituents in the Ross Sea, the observed stage of the tidal oscillation varies only slowly throughout the year. In the austral winter months, \change{SAR}{TerraSAR-X} images \change{are}{have been} acquired during stages of low tide, whereas satellite overpasses concur with stages of high tide during the austral summer months. The first 8 snapshots of our \change{TerraSAR-X data set}{SAR data acquisitions} for the Darwin Glacier show conditions at low tide and only the last 4 are acquired during high tide. Our $\alpha$-map, in turn, ignores this seasonality and may therefore have a low-tide bias. As a result, the contribution of a tide induced landward migration of the grounding line may be affected by the averaging process. The seasonal bias would then modify the scaling of the \change{tide-model}{tide model} within the flexure zone. This \remove{theory} is supported by the finding that low-tide stages in the 'Hillary' GPS record are matched closely by the scaled tide model, but peaks during high-tide stages are still over--estimated (Fig. \ref{fig:forcing_Darwin})

\subsection{Viscoelasticity between snapshots}
Similarly, the linear scaling using an $\alpha$-map only modifies the predicted tidal amplitude, but neglects a viscoelastic time delay in the flexural response towards the grounding line. \cite{wild2017viscosity} found that viscosity is most pronounced in the diurnal tidal components. Harmonic analysis of our GPS records reveals that the diurnal K1 and O1 constituents at 'Hillary' are lagging approximately 20 mins behind 'Shirase'. This signal is currently disregarded in the scaling work flow as ice is treated as a perfect elastic material that transfers tidal \change{motion}{forcing} instantaneously in the flexure zone. This assumption, however, allowed us to improve the accuracy of the \change{tide-model prediction}{predicted displacement} by -74\%. Currently, the viscoelastic signal can only be reconstructed for the times of SAR data acquisition. Including viscoelasticity between times of satellite overpasses \change{may therefore be only}{offers} a small, but

systematic, opportunity for further refinement. \add{In our study area, the rate of tidal change is up to $10$~cm~hr$^{-1}$} (Tab. \ref{tab:SAR_table}) \add{and the viscoelastic misfit corresponding to 20~minutes time delay is therefore up to about 3~cm.}

When separating the viscoelastic contribution from the reconstructed maps of vertical displacement at times of satellite overpasses, we assume that an $\alpha$-prediction corresponds to an instantaneous elastic response. This is justified by viscoelasticity being most pronounced when rates of tidal change are maximal. \add{By expressing the viscoelastic misfits in percent of prevailing tidal amplitude during the times of satellite overpasses, the areas of pronounced viscoelastic effects can be visualised. They are most pronounced within the Darwin Glacier's shear zone} (Fig. \ref{fig:visco_errors_Darwin}). \remove{SAR images acquired during periods of spring tides at the Darwin Glacier show also a significant viscoelastic contribution that diminishes during neap tide periods.}

When predicting rates of tidal change using the adjusted Ross\_Inv\_2002 tide model, we identify a threshold of $\dot{A}\approx \pm0.05$~m~h$^{-1}$ (SAR times 1, 2, 7, 8, 10, 11 in Tab. \ref{tab:SAR_table}) above which viscoelasticity is well represented in the reconstructed vertical displacement maps (panels 1, 6, 7, 8, 10 and 11 in Fig. \ref{fig:visco_maps_Darwin}). Image 6 is thereby an exception, as the used Ross\_Inv\_2002 tide model was \change{adjusted largely}{largely adjusted} (-0.082 m), which affects the viscoelastic model. \add{We find that the error due to viscoelasticity on the floating part of the ice shelf increases with the absolute rate of tidal change} (Fig. \ref{fig:visco_errors_Darwin}). \add{SAR images acquired during periods of spring tides at the Darwin Glacier show \add{also} a significant viscoelastic contribution that diminishes during neap tide periods.} These independent observations \add{from satellite data alone} support our suggested threshold of $\pm0.05$~m~h$^{-1}$ for the separation of elastic and viscoelastic signals, as derived from tiltmeter data on the Southern McMurdo Ice Shelf presented in an earlier study \citep[Fig. 8 in][]{wild2017viscosity}. The advantage of separating the elastic from the viscoelastic contribution to the tidal flexure pattern is the large potential for improving current inverse modelling techniques to determine grounding-zone ice thickness from DInSAR measurements alone. Hereby, an elastic model is currently employed to optimize grounding-zone ice thickness to match the surface flexure from DInSAR. \remove{This is because an elastic model for tidal flexure is only forced by the 'apparent' tidal amplitude (Eq.), which can be measured directly from the interferogram on the freely-floating area. A viscoelastic model additionally incorporates the time derivative of the tidal forcing (Eq.) and hence captures the rate of tidal change. This information, in turn, can not be deduced directly from the interferogram which makes the usage of auxiliary tide models inevitable. Tide models, however, have shown to be prone to large inaccuracies around Antarctica making a successful inversion of viscoelastic flexure models highly elusive.} The applicability of an elastic model varies from location to location as effective viscosity is dependent on ice temperature and shear stress \citep{marsh2014grounding}. Our method to separate the two contributions to the flexure pattern may therefore help to remove the viscoelastic

contamination and allow purely elastic inverse modelling.
\remove{Furthermore, the threshold of $\pm0.05$~m~h$^{-1}$ is invaluable
to determine which satellite data acquisitions should be used for this
calculation.} \change{This}{Such an} analysis, however, goes beyond the
scope of this manuscript and will be published elsewhere.

\subsection{Large-scale ice anisotropy}
Fast-moving glacial environments like the Darwin Glacier are subject to
large deformation by flow convergence and divergence, ice compression and
extension, lateral shearing at the margins accompanied by fracture under
tension and rapid thinning \change{at the ice-ocean interface}{by basal
melt}. With cumulative deformation, a crystallographic fabric evolves
that reflects the glacier's flow history \citep{alley1988fabrics}, and
with it strain-dependent mechanical anisotropy of ice. The standard
deviation map, Fig. \ref{fig:LS_STD_maps}, shows a narrow band of larger
misfits extending from the Darwin Glacier's inner shear margin out
towards the freely-floating ice shelf. As preferred crystallographic
orientation develops with strain, effective viscosity decreases of about
a factor of ten compared to initially isotropic polycrystalline ice
\citep{hudleston2015structures}. Our analysis of tiltmeter data reveals a
five-fold reduced viscosity at the very dynamic Darwin Glacier compared
to an earlier study at the almost stagnant Southern McMurdo Ice Shelf
\citep{wild2017viscosity}. We \change{theorise}{hypothesize} that this
microscopic process explains the macroscopic response observed here, and
accounts for the measured glacial \change{heterogenity}{heterogeneity}
within the embayment. Large scale observations of ice anisotropy, in
turn, are currently the missing key to improve parametrisations to
account for polar ice anisotropy in ice-sheet flow modelling
\citep{gagliardini2009review}. \add{Other processes have been proposed
which lead to ice softening in areas with high strain rates.
Thermomechanical modelling suggests that shear heating and consequent
thermal softening reduces lateral drag in ice-stream margins}
\citep{perol2015shear}. \add{Fracture modelling implies that damage
reduces ice viscosity along confined crevassed zones with consequences on
ice-shelf scale} \citep{albrecht2014fracture}. \add{Full-Stokes
viscoelastic modelling shows that Glen's non-linear flow law and tidal
stresses in the ice-shelf flexure zone are sufficient to explain large-
scale temporal variations in ice dynamics} \citep{rosier2018tidal}.
\add{These processes, or a combination of them, might certainly be at
play but they do not explain why a band of higher standard deviations can
be observed in the shear zone of the Darwin Glacier which is absent in
the flexure zone of the SMIS} (Fig. \ref{fig:LS_STD_maps}). \add{We
therefore attribute this difference to ice-fabric reorientation in the
shear margin.}

\subsection{Refining tidal constituents using DInSAR}
\add{The idea of using SAR interferometry to derive a full set of tidal
harmonics was first laid out in a study of tides in the Weddell Sea}
\citep{rignot2000observation}. \add{The authors discussed that DInSAR
images cannot be transformed into individual displacement fields because
of the nonuniqueness of the inversion. A large number of independent
DInSAR images is required to overcome this problem and to resolve the
phase of tidal constituents that are close to the repeat-pass of the SAR
satellite. For example, multiples of the lunar diurnal constituent K1

**Commented [L48]:** Again, not really clear what "inner" means Darwin Glacier.

**Commented [L49]:** Maybe a paragraph break here

(23.93~h) are relatively close to the exact integer repeat-pass of
TerraSAR-X (11~days)_, meaning that the observed amplitude of the K1
constituent is only varying once throughout the year. Consequently_, SAR
images need to be acquired at least over the duration of one year to
provide some redundancy for the inversion step of DInSAR images to tidal
constituents. However, with an exact 12~h period, the stage of the
semidiurnal solar tide, S2, will always be the same at each satellite
pass_, making TerraSAR-X and similar satellites with repeat ~passes of
integer days blind to the S2 constituent.} \add{For example,}
\cite{minchew2017tidally} \add{needed a unique spatially and temporally
dense SAR acquisition campaign as well as \textit{a-priori} knowledge of
the temporal basis functions from GPS data to empirically determine tidal
constituents on Rutford Ice Stream. The four COSMO-SkyMed satellites in
orbit, however_, produce repeat-passes of 1, 3, 4 and 8 days and are blind
to the S2 constituent as well even when using $>1000$ available DInSAR
images. Although other dominant tidal constituents like M2 (12.4~h) and
O1 (25.82~h) were inferred successfully, the method presented here can
achieve a higher accuracy of~fo the total tide with fewer~less DInSAR
images. From another perspective, _the inclusion of an auxiliary tide
model eases the requirement of a very large number of DInSAR images.}

\conclusions[Conclusions and Outlook]  %% \conclusions[modified heading
if necessary]
\label{Conc}
\add{Accurate prediction of ocean tides in coastal areas around
Antarctica _is crucial as the majority of Antarctica's ice is discharged
through large outlet glaciers.} \change{Here we present the first data
fusion of DInSAR with traditional Antarctic tide-modelling to predict
spatial variability of tidal motion. The principal value of using DInSAR
and tide models in tandem lies in the spatio-temporal benefits of
resolving small features over large regions.}{We presented a data fusion
method between DInSAR and traditional Antarctic tide models to predict
spatial variability of tidal motion near the grounding line. The primary
value of using DInSAR in conjunction with tide models lies in the spatio-
temporal benefits of resolving complex grounding zone deformation.} Their
symbiosis not only improves current accuracies of the predicted tidal
amplitudes in coastal regions generally, but also avoids issues related
to the timing of the tidal wave _and the sun-synchronous satellite orbit
when attempting to derive tide-models from SAR data alone. \add{In our
study area,} the method presented in this paper improves traditional tide
modelling in average by -22\% from 11.8~cm to 9.3~cm RMSE against 16 days
of GPS data. The GPS station 'Shirase' on the freely-floating part of the
Darwin Glacier has proven invaluable to determine which \change{tide-
model}{tide model} has to be used to best reconstruct the vertical
displacement during satellite overpasses. For the Darwin Glacier, the
TPXO7.2 tide model predicts best the tidal oscillation. With using DInSAR
measurements to adjust the TPXO7.2 tidal prediction, its RMSE could be
improved by -39\% from 10.8~cm to 6.7~cm. \remove{which exceeds the
average tide model improvement of -35\% within the last decade (Stammer
et al., 2014).}

Our GPS record from 'Shirase' is too short to \change{develop a local
tide model that improves}{improve} already available Antarctic tide
models. A longer record is required to adequately resolve a full set of

Commented [L50]: A casual reader might get confused here: how is COSMO-SkyMed related to TerraSAR-X (Not at all, right?)  SO, first time Minchew comes up, explain this, and why you can't use the same satellite for your SMIS and Darwin Glacier studies but have to use TerraSAR-X instead.

Commented [L51]: This is wrong, and probably easiest to delete it.

Commented [L52]: I don't think this is right. It does *not* avoid issues with timing (phase); it simply works to minimize problems that arise from phase errors. The earlier comment about the "thought experiment" about a major harmonic with an extreme phase error seems relevant here.

tidal constituents. \remove{The GPS record from 'Hillary' could not be used for this purpose as it was recorded within the tidal flexure zone.} \add{We produced an empirical displacement map from DInSAR for tidal deflection ($\alpha$-map).} Comparison of \change{its measurements}{a GPS record within the tidal flexure zone ('Hillary')} with predicted vertical displacement from feeding the $\alpha$-map with the adjusted TPXO7.2 tide model shows a -74\% improvement over using the tide-model output alone. This independent validation supports the finding that \add{our method for making use of} DInSAR is very useful for refining tide models in Antarctic \change{grounding-zones}{grounding zones}. \remove{Accurate prediction of ocean tides in coastal areas is crucial as the majority of Antarctica's ice is discharged through large outlet glaciers.}

Numerical modelling of ice dynamics in Antarctic grounding zones commonly assumes that ice is isotropic and homogeneous i.e. of \add{the} same density and rheological properties throughout. Our analysis reveals that this assumption is valid for the Southern McMurdo Ice Shelf, an almost stagnant area with a simple grounding line \add{configuration}, but invalid for the Darwin Glacier, a fast-flowing outlet glacier with complex shear margins causing non-negligible ice \change{heterogenity}{heterogeneity} within the embayment.

Further work is required \change{(1) to incorporate viscoelasticity to continue refining predictions of tidal motion between times of satellite overpasses, (2) to develop an automated method to monitor grounding-line migration due to ocean tides and (3) to perform inverse modelling of tidal elastic flexure to indirectly measure ice thickness from SAR data.}{in order to improve tide models in a larger variety of grounding zones by including effects of grounding-line migration and variability of horizontal ice flow.}

%% The following commands are for the statements about the availability of data sets and/or software code corresponding to the manuscript.
%% It is strongly recommended to make use of these sections in case data sets and/or software code have been part of your research the article is based on.

\codeavailability{The code is freely available to the scientific community. Collaboration is anticipated and desired.
} %% use this section when having only software code available

\dataavailability{TerraSAR-X data presented in this paper are subject to license agreements. GPS/tiltmeter and ApRES data are available upon request.} %% use this section when having only data sets available

\codedataavailability{No data sets, nor software, are part of this study.} %% use this section when having data sets and software code available

\sampleavailability{No samples were collected for this study.} %% use this section when having geoscientific samples available

**Commented [L53]:** a negative improvement is not an improvement!

```
\appendix
\section{GPS evaluation of tide models}    %% Appendix A
The quality of the used \change{tide-model}{tide model} to correctly
reconstruct tidal displacement at the times of SAR data acquisitions, is
also crucial to accurately predict spatial variability in tidal motion
for all times. Here, we assume that a freely-floating area on the
\change{ice-shelf}{ice shelf} experiences the full oscillation as
predicted from a tide model. In this area, however, tide-model output
deviates from our DInSAR measurements. This indicates either that the
area under investigation is prevented from a freely-floating state by
lateral stresses within the embayment, or that the tide-model prediction
is inaccurate for this area. We circumvent this ambiguity by making use
of the high vertical accuracy of DInSAR and correct the tide-model
prediction to match our satellite measurements. This raises the question
of whether the adjustment improves or worsens the match to a 'real' tidal
motion ? We therefore independently evaluate the pre- and post adjustment
tide-model predictions and calculate their RMSE to 16 days of GPS data
from the freely-floating area (Tab. \ref{tab:GPS_table}). The adjustment
improves all traditional tide model predictions by up to -39\% for
TPXO7.2, and only worsens the RMSE for the t\_tide output by +11\%,
indicating that a harmonic analysis of GPS data can not be improved by
using DInSAR for correction purposes. We choose TPXO7.2 for further
processing as it displays the overall smallest RMSE (6.7~cm) and
replicates the small-scale variability observed during the neap-tide
period in the second half of our GPS record.
%\subsection{}      %% Appendix A1, A2, etc.

\noappendix        %% use this to mark the end of the appendix section

%% Regarding figures and tables in appendices, the following two options
are possible depending on your general handling of figures and tables in
the manuscript environment:

%% Option 1: If you sorted all figures and tables into the sections of
the text, please also sort the appendix figures and appendix tables into
the respective appendix sections.
%% They will be correctly named automatically.

%% Option 2: If you put all figures after the reference list, please
insert appendix tables and figures after the normal tables and figures.
%% To rename them correctly to A1, A2, etc., please add the following
commands in front of them:

%% Please add \clearpage between each table and/or figure. Further
guidelines on figures and tables can be found below.

\authorcontribution{All authors conceived the study and conducted
fieldwork. CW developed the algorithm, performed the data analysis and
```

wrote a first version of the paper. OM and WR processed the SAR images
for the Darwin Glacier and Southern McMurdo Ice Shelf, respectively. All
authors finalized and approved the manuscript.
} %% it is strongly recommended to make use of this section

\competinginterests{The authors declare that the research was conducted
in the absence of any commercial or financial relationships that could be
construed as a potential conflict of interest.
} %% this section is mandatory even if you declare that no competing
interests are present

\disclaimer{} %% optional section

\begin{acknowledgements}
We thank Antarctica New Zealand for logistical support and the Scott Base
staff for their dedication to the Transantarctic Ice Deflection
Experiment. The project was supported by the Royal Geographic Society
(Marsden Fast Start, PI O. Marsh) and by the Past Antarctic Science and
Future Implications Program (PACaFI). D. Price, M. Ryan, D. Floricioiu
and E. Scheffler contributed to fieldwork. TerraSAR-X data were provided
through DLR project HYD1421. Landsat-8 images courtesy of the US
Geological Survey. AWS data from Scott Base were provided by NIWA, and
for Marilyn station by AMRC, SSEC, UW-Madison. The authors enjoyed
fruitful discussions with H. Purdie and M. Sellier. CW also thanks R.
Mueller for sharing oceanographic expertise. The thorough comments of one
anonymous reviewer and L. Padman's valuable insight into tidal dynamics
largely improved the paper. We also thank A. Robinson for editing.
\end{acknowledgements}

%% REFERENCES

%% The reference list is compiled as follows:

%\begin{thebibliography}{}

%\bibitem[AUTHOR(YEAR)]{LABEL1}
%REFERENCE 1

%\bibitem[AUTHOR(YEAR)]{LABEL2}
%REFERENCE 2

%\end{thebibliography}

%% Since the Copernicus LaTeX package includes the BibTeX style file
copernicus.bst,
%% authors experienced with BibTeX only have to include the following two
lines:
%%
\bibliographystyle{copernicus}
\bibliography{./JabRef/WC_database}

```
%%
%% URLs and DOIs can be entered in your BibTeX file as:
%%
%% URL = {http://www.xyz.org/~jones/idx_g.htm}
%% DOI = {10.5194/xyz}

%% LITERATURE CITATIONS
%%
%% command                      & example result
%% \citet{jones90}|             & Jones et al. (1990)
%% \citep{jones90}|             & (Jones et al., 1990)
%% \citep{jones90,jones93}|     & (Jones et al., 1990, 1993)
%% \citep[p.~32]{jones90}|      & (Jones et al., 1990, p.~32)
%% \citep[e.g.,][]{jones90}|    & (e.g., Jones et al., 1990)
%% \citep[e.g.,][p.~32]{jones90}| & (e.g., Jones et al., 1990, p.~32)
%% \citeauthor{jones90}|        & Jones et al.
%% \citeyear{jones90}|          & 1990

\clearpage

\begin{figure}[ht!]
\centering{\includegraphics[width=\textwidth]{alpha_maps_v2.png}}
\caption{$\alpha$-maps of percentage vertical displacement due to ocean
tides. Red colors highlight areas that can be assumed to be
\change{freely-floating}{freely floating}. The white crosses show the
tide-model locations that also serve as a common reference point across
the images. The solid black line is the location of the profiles shown in
Figures \ref{fig:ls_process_DInSAR_SMIS} and
\ref{fig:ls_process_SAR_SMIS} on the Southern McMurdo IceShelf (left).
The dashed black line shows the location of the profiles along the Darwin
Glacier's centerline shown in Fig. \ref{fig:LS_flexure_curves_Darwin}
(right) and and 'Hillary' in the tidal-flexure
zone. White contours delineate areas of constant vertical displacement.
The map background is contrast-stretched Landsat 8 panchromatic imagery.
The geographic projection is Antarctic Polar Stereographic with easting
and northing coordinates shown in kilometers.}
\label{fig:alpha_maps}
\end{figure}
\clearpage

\begin{figure}[ht!]
\centering{\includegraphics[width=86mm]{ls_process_DInSAR_v2.png}}
\caption{Vertical displacements along a profile through the grounding
zone of the Southern McMurdo Ice Shelf, as (top) measured with 9 DInSAR
interferograms, (center) predicted from an empirical displacement model
($\alpha$-map) and (bottom) their difference.}
\label{fig:ls_process_DInSAR_SMIS}
\end{figure}
\clearpage

\begin{figure}[ht!]
\centering{\includegraphics[width=86mm]{ls_process_SAR_v2.png}}
```

```latex
\caption{Reconstruction of vertical displacement along the profile during
the 12 times of satellite overpasses on the Southern McMurdo Ice Shelf.
(Top) a combination of an empirical displacement model with adjusted CATS
tide-model output, (center) their least-square adjustment and (bottom)
the final vertical displacement profiles during the times of SAR data
acquisition.}
\label{fig:ls_process_SAR_SMIS}
\end{figure}
\clearpage

\begin{figure}[ht!]
\centering{\includegraphics[width=\textwidth]{forcing_v2.png}}
\caption{The tidal oscillation at the Darwin Glacier as predicted by four
tide models and a harmonic analysis of GPS data from the freely-floating
area. The tide-model outputs are adjusted to match DInSAR observations
using a least-squares fitting technique published in
\cite{wild2018unraveling}. Black vertical lines coincide with times of
SAR data acquisitions. Values for the prevailing tidal amplitudes and
their adjustment at these times are given in Table \ref{tab:SAR_table}.
Gray shaded areas delineate the duration of the TIDEx campaign, when GPS
data was acquired for validation (Figs. \ref{fig:Hillary_validation} and
\ref{fig:GPS_validation}).}
\label{fig:forcing_Darwin}
\end{figure}
\clearpage

\begin{table*}[ht!]
\centering
\caption{SAR imagery used for the Darwin Glacier, least-squares
adjustment (\change{$\theta$}{$\Delta A$} in m) for 5 tide models, tidal
amplitude ($A$ in m) as predicted with the TPXO7.2 tide model and rate of
tidal change ($\dot{A}$ in m~h$^{-1}$) as predicted with the
Ross\_Inv\_2002 tide model.}
\label{tab:SAR_table}
\begin{tabular}{lc|rrrrr|rr}\toprule
SAR: & Date 13:57 (UTC) & $\Delta A_{\text{CATS}}$ & $\Delta
A_{\text{RossInv}}$ & $\Delta A_{\text{Ross9cm}}$ & $\Delta
A_{\text{TPXO7.2}}$ & $\Delta A_{\text{t\_tide}}$ & $A_{\text{TPXO7.2}}$
& $\dot{A}_{\text{RossInv}}$ \\\midrule
1               & 25/05/16        & 0.109         & 0.098             &
0.133           & 0.101             & -0.004 & -0.341  & -0.059      \\
2               & 05/06/16        & -0.061        & -0.066            & -
0.097           & -0.039            & -0.030 & -0.666  & -0.057      \\
3               & 16/06/16        & 0.029         & 0.035             & -
0.050           & 0.034             & 0.013  & -0.409  & -0.007      \\
4               & 27/06/16        & 0.032         & -0.012            & -
0.049           & -0.009            & -0.111 & 0.002   & -0.022      \\
5               & 08/07/16        & 0.054         & -0.022            &
0.107           & 0.008             & 0.122  & -0.271  & -0.037       \\
6               & 19/07/16        & -0.086        & -0.082            &
0.035           & -0.088            & 0.206  & -0.661  & -0.005      \\
7               & 30/07/16        & -0.091        & 0.015             & -
0.026           & -0.045            & 0.074  & -0.687  & 0.080       \\
```

```latex
8              & 10/08/16      & -0.078         & 0.001          & -
0.035           & -0.040         & -0.080 & -0.276  & 0.072      \\
9              & 26/10/16      & -0.073         & -0.023         & -
0.053           & -0.046         & -0.244 & -0.132  & 0.011      \\
10             & 06/11/16      & -0.044         & -0.009         & -
0.088           & -0.023         & -0.172 & 0.087   & 0.096      \\
11             & 17/11/16      & 0.099          & 0.025          & -
0.002           & 0.084          & 0.059  & 0.522   & 0.052      \\
12             & 28/11/16      & 0.109          & 0.041          &
0.124            & 0.062           & 0.168  & 0.398   & -0.029
\\\midrule
\multicolumn{2}{r|}{mean absolute $\Delta A$} & 0.072 & 0.036 & 0.067 &
0.048 & 0.107 & - & - \\\bottomrule
\end{tabular}
\end{table*}

\clearpage

\begin{table*}[ht!]
\centering
\caption{DInSAR images of the Darwin Glacier. The SAR combination from 12
available SAR images, the tidal amplitude ($A$ in m) as measured at the
Shirase location as well as predicted with the TPXO7.2 tide model.}
\label{tab:DInSAR_table}
%\begin{adjustbox}{totalheight=\textheight-2\baselineskip}
\begin{minipage}[b]{0.45\linewidth}\centering
 \begin{tabular}{lr|rr|r}\toprule
ID & SAR combination & $A_{\text{TerraSAR-X}}$ & $A_{\text{TPXO7.2}}$ &
$\Delta$    \\\midrule
1  & (1-2)-(2-3)     & 0.581     & 0.582     & -0.001 \\
2  & (1-2)-(3-4)     & 0.740     & 0.735     & 0.004  \\
3  & (1-2)-(4-5)     & 0.057     & 0.052     & 0.005  \\
4  & (1-2)-(5-6)     & -0.061    & -0.065    & 0.004  \\
5  & (1-2)-(6-7)     & 0.298     & 0.299     & -0.001 \\
6  & (1-2)-(7-8)     & 0.734     & 0.736     & -0.002 \\
7  & (1-2)-(9-10)    & 0.552     & 0.544     & 0.008  \\
8  & (1-2)-(10-11)   & 0.736     & 0.760     & -0.024 \\
9  & (1-2)-(11-12)   & 0.207     & 0.201     & 0.006  \\
10 & (2-3)-(3-4)     & 0.154     & 0.154     & 0.001  \\
11 & (2-3)-(4-5)     & -0.529    & -0.530    & 0.001  \\
12 & (2-3)-(5-6)     & -0.646    & -0.647    & 0.001  \\
13 & (2-3)-(6-7)     & -0.288    & -0.283    & -0.005 \\
14 & (2-3)-(7-8)     & 0.137     & 0.154     & -0.017 \\
15 & (2-3)-(9-10)    & -0.034    & -0.038    & 0.004  \\
16 & (2-3)-(10-11)   & 0.190     & 0.178     & 0.012  \\
17 & (2-3)-(10-11)   & -0.376    & -0.381    & 0.004  \\
18 & (3-4)-(4-5)     & -0.688    & -0.683    & -0.004 \\
19 & (3-4)-(5-6)     & -0.805    & -0.801    & -0.005 \\
20 & (3-4)-(6-7)     & -0.446    & -0.436    & -0.009 \\
21 & (3-4)-(7-8)     & -0.014    & 0.001     & -0.015 \\
22 & (3-4)-(9-10)    & -0.192    & -0.192    & -0.000 \\
23 & (3-4)-(10-11)   & 0.055     & 0.025     & 0.030  \\
\vdots & \vdots & \vdots & \vdots & \vdots \\\bottomrule
\end{tabular}
```

```latex
\end{minipage}
\hspace{0.5cm}
\begin{minipage}[b]{0.45\linewidth}
%\end{minipage} \hfill
%\begin{minipage}{0.5\textwidth}
\begin{tabular}{lr|rr|r}\toprule
ID & SAR combination & $A_{\text{TerraSAR-X}}$ & $A_{\text{TPXO7.2}}$ &
$\Delta$   \\\midrule
\vdots & \vdots & \vdots & \vdots & \vdots \\
24 & (3-4)-(11-12)   & -0.536   & -0.534   & -0.002 \\
25 & (4-5)-(5-6)     & -0.112   & -0.117   & 0.005  \\
26 & (4-5)-(6-7)     & 0.246    & 0.247    & -0.001 \\
27 & (4-5)-(7-8)     & 0.679    & 0.684    & -0.005 \\
28 & (4-5)-(9-10)    & 0.496    & 0.492    & 0.004  \\
29 & (4-5)-(10-11)   & 0.690    & 0.708    & -0.018 \\
30 & (4-5)-(11-12)   & 0.155    & 0.149    & 0.006  \\
31 & (5-6)-(6-7)     & 0.363    & 0.364    & -0.001 \\
32 & (5-6)-(7-8)     & 0.811    & 0.801    & 0.010  \\
33 & (5-6)-(9-10)    & 0.620    & 0.609    & 0.011  \\
34 & (5-6)-(10-11)   & 0.803    & 0.825    & -0.023 \\
35 & (5-6)-(11-12)   & 0.274    & 0.266    & 0.008  \\
36 & (6-7)-(7-8)     & 0.430    & 0.437    & -0.007 \\
37 & (6-7)-(9-10)    & 0.237    & 0.244    & -0.007 \\
38 & (6-7)-(10-11)   & 0.466    & 0.461    & 0.005  \\
39 & (6-7)-(11-12)   & -0.107   & -0.098   & -0.009 \\
40 & (7-8)-(9-10)    & -0.202   & -0.192   & -0.009 \\
41 & (7-8)-(10-11)   & 0.025    & 0.024    & 0.000  \\
42 & (7-8)-(11-12)   & -0.541   & -0.535   & -0.006 \\
43 & (9-10)-(10-11)  & 0.228    & 0.217    & 0.011  \\
44 & (9-10)-(11-12)  & -0.343   & -0.342   & -0.001 \\
45 & (10-11)-(11-12) & -0.565   & -0.559   & -0.006 \\\hline
\multicolumn{4}{r|}{mean absolute error}& 0.007 \\\bottomrule
\end{tabular}
\end{minipage}
%\end{minipage}
%\end{adjustbox}
\end{table*}

\clearpage

\begin{figure}[ht!]
\centering{\includegraphics[width=0.75\textwidth]{Darwin_DInSARs.png}}
\caption{Selection of three measured and modelled images from 45
available DInSAR combinations. The top panels show conditions at a
relatively large double-differential tidal amplitude (ID 37), the center
panels display a pronounced visoelastic signal in the Darwin
\change{Glacier's}{Glaciers} grounding zone (ID 39) and the bottom panels
show a complex flexural pattern (ID 15).}
\label{fig:Darwin_DInSARs}
\end{figure}
\clearpage

\begin{figure}[ht!]
```

[revised manuscript text omitted]

\begin{table*}[ht!]
\centering
\caption{Amplitude and phase of the K1 tidal constituents from harmonic
analysis of tiltmeter measurements and values of the rheological
parameters to minimize the average RMSE. Amplitudes are given in degrees,
phases are the phase lag of the K1 constituent with respect to the
equilibrium tide on Greenwich longitude.}
\label{tab:K1_table}
\begin{tabular}{lr|lcl|lcr}\toprule
                           &  & &K1 amplitude $\pm$ error ($^{\circ}$)&
&          & K1 phase $\pm$ error ($^{\circ}$) &           \\\midrule
&tiltmeter 1                         & -0.001 & 0.0033  & +0.001 & -
24.34    & 206.22        & +24.34     \\
&tiltmeter 2                         & -0.001 & 0.0044  & +0.001 & -
8.02     & 215.35        & +8.02      \\
&tiltmeter 3                         & -0.001 & 0.0044  & +0.001 & -
7.46     & 218.64        & +7.46      \\
&tiltmeter 4                         & -0.002 & 0.0065  & +0.002 & -
13.63    & 219.34        & +13.63     \\
&tiltmeter 5                         & -0.001 & 0.0055  & +0.001 & -
12.36    & 198.09        & +12.36     \\
&tiltmeter 6                         & -0.001 & 0.0014  & +0.001 & -
14.95    & 207.34        & +14.95     \\
&tiltmeter 7                         & -0.001 & 0.0025  & +0.001 & -
18.06    & 181.97        & +18.06
\\\midrule
elastic & \begin{tabular}{@{}r@{}} best $E$ (GPa) \\  average RMSE
($^{\circ}$) \end{tabular} & \begin{tabular}{@{}l@{}}1.5 \\ 0.00098
\end{tabular} & \begin{tabular}{@{}c@{}}1.0 \\ 0.00147 \end{tabular} &
\begin{tabular}{@{}r@{}}0.5 \\ 0.00198 \end{tabular} &
\begin{tabular}{@{}l@{}}0.5 \\ 0.00127 \end{tabular} &
\begin{tabular}{@{}c@{}}1.0 \\ 0.00147 \end{tabular} &
\begin{tabular}{@{}r@{}}2.0 \\ 0.00182 \end{tabular}\\\midrule
&best $E$ (GPa)               & 1.5      & 1.0 & 1.0
& 0.5      & 1.0                & 2.0           \\
viscoelastic &best $\nu$ (TPa s)           & 19.9     & 10.0 & 10.0
& 12.6     & 10.0                & 7.9         \\
&average RMSE ($^{\circ}$)     & 0.00077       & 0.00119 & 0.00170
& 0.00122 & 0.00119                 & 0.00128 \\\bottomrule
\end{tabular}
\end{table*}
```

```
\appendixfigures  %% needs to be added in front of appendix figures
\begin{figure}[ht!]
\centering{\includegraphics[width=172mm]{GPS_validation_v2.png}}
\caption{Validation of the tidal predictions of 5 tide models with a GPS
record from the freely-floating 'Shirase' station. The tide-model outputs
are adjusted to match DInSAR observations using a least-squares fitting
technique published in \cite{wild2018unraveling}. Root-mean-square-errors
before and after this adjustment are presented in Tab.
\ref{tab:GPS_table}. The length of the records corresponds to the gray-
shaded area in Fig. \ref{fig:forcing_Darwin}.}
\label{fig:GPS_validation}
\end{figure}

\appendixtables   %% needs to be added in front of appendix tables
\begin{table}[ht!]
\centering
\caption{Root-mean-square-errors in m between tide-model output and GPS
data from 'Shirase' before and after the adjustment to match DInSAR.}
\label{tab:GPS_table}
\begin{tabular}{lrr}\toprule
Tide-model:           & RMSE before:      & RMSE after:  \\\midrule
CATS2008a\_opt          & 0.117 & 0.087                \\
Ross\_Inv\_2002         & 0.112 & 0.091                \\
Ross\_VMADCP\_9cm       & 0.135 & 0.127                \\
TPXO7.2                 & 0.108 & 0.067                \\\midrule
mean & 0.118 & 0.093 \\\midrule
t\_tide                 & 0.127 & 0.141 \\\bottomrule
\end{tabular}
\end{table}

%% FIGURES

%% When figures and tables are placed at the end of the MS (article in
one-column style), please add \clearpage
%% between bibliography and first table and/or figure as well as between
each table and/or figure.

%% ONE-COLUMN FIGURES

%%f

%
%%% TWO-COLUMN FIGURES
%
%%f

%
%
%%% TABLES
%%%
```

```
%%% The different columns must be seperated with a & command and should
%%% end with \\ to identify the column brake.
%
%%% ONE-COLUMN TABLE
%
%%t
%\begin{table}[t]
%\caption{TEXT}
%\begin{tabular}{column = lcr}
%\tophline
%
%\middlehline
%
%\bottomhline
%\end{tabular}
%\belowtable{} % Table Footnotes
%\end{table}
%
%%% TWO-COLUMN TABLE
%
%%t
%\begin{table*}[t]
%\caption{TEXT}
%\begin{tabular}{column = lcr}
%\tophline
%
%\middlehline
%
%\bottomhline
%\end{tabular}
%\belowtable{} % Table Footnotes
%\end{table*}
%
%%% LANDSCAPE TABLE
%
%%t
%\begin{sidewaystable*}[t]
%\caption{TEXT}
%\begin{tabular}{column = lcr}
%\tophline
%
%\middlehline
%
%\bottomhline
%\end{tabular}
%\belowtable{} % Table Footnotes
%\end{sidewaystable*}
%
%
%%% MATHEMATICAL EXPRESSIONS
%
%%% All papers typeset by Copernicus Publications follow the math
typesetting regulations
```

```
%%% given by the IUPAC Green Book (IUPAC: Quantities, Units and Symbols
in Physical Chemistry,
%%% 2nd Edn., Blackwell Science, available at:
http://old.iupac.org/publications/books/gbook/green_book_2ed.pdf, 1993).
%%%
%%% Physical quantities/variables are typeset in italic font (t for time,
T for Temperature)
%%% Indices which are not defined are typeset in italic font (x, y, z, a,
b, c)
%%% Items/objects which are defined are typeset in roman font (Car A, Car
B)
%%% Descriptions/specifications which are defined by itself are typeset
in roman font (abs, rel, ref, tot, net, ice)
%%% Abbreviations from 2 letters are typeset in roman font (RH, LAI)
%%% Vectors are identified in bold italic font using \vec{x}
%%% Matrices are identified in bold roman font
%%% Multiplication signs are typeset using the LaTeX commands \times (for
vector products, grids, and exponential notations) or \cdot
%%% The character * should not be applied as mutliplication sign
%
%
%%% EQUATIONS
%
%%% Single-row equation
%
%\begin{equation}
%
%\end{equation}
%
%%% Multiline equation
%
%\begin{align}
%& 3 + 5 = 8\\
%& 3 + 5 = 8\\
%& 3 + 5 = 8
%\end{align}
%
%
%%% MATRICES
%
%\begin{matrix}
%x & y & z\\
%x & y & z\\
%x & y & z\\
%\end{matrix}
%
%
%%% ALGORITHM
%
%\begin{algorithm}
%\caption{...}
%\label{a1}
%\begin{algorithmic}
%...
```

```
%\end{algorithmic}
%\end{algorithm}
%
%
%%% CHEMICAL FORMULAS AND REACTIONS
%
%%% For formulas embedded in the text, please use \chem{}
%
%%% The reaction environment creates labels including the letter R, i.e.
(R1), (R2), etc.
%
%\begin{reaction}
%%% \rightarrow should be used for normal (one-way) chemical reactions
%%% \rightleftharpoons should be used for equilibria
%%% \leftrightarrow should be used for resonance structures
%\end{reaction}
%
%
%%% PHYSICAL UNITS
%%%
%%% Please use \unit{} and apply the exponential notation

\end{document}
```

---

## Author Response (AR2)

**Reply to minor revisions of 'Differential InSAR for tide modelling in Antarctic ice-shelf grounding zones'**

Christian Wild, Oliver Marsh, Wolfgang Rack

Comment L1:
The 7mm are the mean absolute error between 45 DInSAR images and their corresponding 45 reconstructions from mosaicking the maps of vertical ice displacement at the times of satellite data acquisition together (Table 2). We reworded this sentence.

Comment L2:
Removed the (i),(ii),(iii) from the list as suggested by the reviewer

Comment L3:
The amplitude of the backscattered wave depends on the physical (aspect, surface roughness) and electrical (permittivity) properties of the target surface. Their contributions vary depending on the wavelength of the sensor and which type of reflector it is (surface or volume scatterer). As this paper dives into phase analysis, we only mention controls on the received amplitude without further details.

Comment L4:
Correct, InSAR shows horizontal AND vertical movement. We have removed 'lateral'.

Comment L5:
We have replaced the Tong 2018 cite with the recent Mouginot paper on Antarctic wide mapping of ice velocity from InSAR

Comment L6:
We have reworded and define DInSAR now a second time after the abstract before it is used in a sentence.

Comment L7:
We have included both suggested references

Comment L8:
We have removed the paragraph as suggested.

Comment L9:
We explain one sentence earlier why Minchew's data set is so special. However, we have included an additional term that data is 'sparse' for the remainder of Antarctica.

Comment L10:
The 71 day limit is the duration of the ERS tandem mission and not only the data set that Baek and Shum had available. We have reworded and added another sentence.

Comment L11:
We have removed this sentence

Comment L12:
Removed 'in turn' from the sentence.

Comment L13:
We have removed this poorly written part.

Comment L14:
The 30km refers to the swath width (i.e. look direction) while the 50km is equivalent to 8 seconds of stripmap data in azimuth direction. We've added this information.

Comment L15:
Changed as suggested by the reviewer

Comment L16:
Changed as suggested by the reviewer

Comment L17:
Changed to T_TIDE as suggested by the reviewer. Here and throughout the manuscript

Comment L18:
This is in regard to the sensor and not to the time series itself. We have reworded.

Comment L19:
We have replaced the statement about non-tidal effects.

Comment L20:
We have assigned 'a' and 'b' suffixes

Comment L21:
Changed here and in the next sentence  to 'traditional tide modelling'

Comment L22:
See Scheffler's Master thesis who finds an accuracy between 3-5mm.. We have included the reference.

Comment L23:
Changed as suggested

Comment L24:
We have changed to 'constrain' as finite-element models are generally 'constrained' by field data like ice thickness etc

Comment L25:
We removed 'new'

Comment L26:
We agree that phase errors of individual constituents will only be smallish. This affects the rate of change of the harmonic more than the harmonic's amplitude at a certain point in time. By adjusting the total tide, this rate of change is corrected and therefore a good first step in the right direction. We haven't considered  fitting in complex space yet. We included the statement about phase errors being

'smallish'.

Comment L27:
Maybe the confusion comes from 'A' here being a matrix and not the tidal amplitude ? We have changed 'A' to 'C' . The \Delta A further down, however, is the difference between the tidal amplitude and the empirical model.

Comment L28:
I hope it is more clear now. This is the definition of \Delta A

Comment L29:
To perform interferometry, SAR images must originate from the same satellite track. At the Darwin, we had 12 SAR images from the same track which allowed the calculation of 45 DInSAR images. At the SMIS, we also have 12 SAR images but from 3 different tracks. Therefore we could only calculate 9 DInSAR images. So if the number of tracks goes up and the number of SAR images stays the same, the number of resulting DInSAR images will go down.

We agree that the primary control on 'low number' is SAR mission duration and repeat interval, but a secondary control is the viewing geometry of the sensor. If it is always the same track, tides can be studied well (this study). If the ice is illuminated from at least 2 different tracks, the 3D motion vector can be fully resolved. So it depends on what you want to measure with DInSAR. We have added a statement about this limiting the calculation of DInSAR combinations.

Comment L30:
This is true - flexure in the grounding zone can overshoot. Our modelling shows that it is always overshooting in an elastic case for all tidal amplitudes (Fig. 1 left). Viscoelastic models show that overshooting happens only at the min/max tide and the ice in the flexure zone is lagging behind the freely-.floating part. As ice is viscoelastic on tidal frequencies, I think it will depend on the exact timing of the measurement (IceSAT) if an overshoot can be observed or not.

[Figure]

Fig.1:  (left) idealised elastic versus viscoelastic model experiments, (right) transects along 45 DinSAR images of the Darwin glacier, orange is measured, black is modelled

At the Darwin glacier, we don't observe any overshooting in the grounding zone and even spot individual flexure curves which show a very well pronounced viscoelastic pattern (Fig. 1 right). These however are averaged out in the calculation of an alpha map. We added a short sentence.

Comment L31:
We think that callouts are much more practical than panel IDs and keep them if it is allowed by the journal style.

Comment L32:
The gap was there because of a quota in data acquisitions and the desire to acquire data both in the summer and winter. We've added a sentence.

Comment L33:
We have added North arrows to both maps

Comment L34:
We agree, it is much more logic to have the figures ordered this way. Changed as suggested.

Comment L35:
No this sentence refers to the Figure showing 3 examples of DInSAR versus their individual reconstructions.

Comment L36:
We removed 'inner' from the paper and replaced with 'near Diamond Hill'

Comment L37:
Layover is a consequence of slant-range distortion similar to shadowing and foreshortening. No changes to the paper.

Comment L38:
We have changed 'increases' to 'spatial gradients' as suggested

Comment L39:
It is this Python function (`scipy.interpolate.RectBivariateSpline`). We have added additional information to our interpolation routine.

Comment L40:
We have removed 'course of the'

Comment L41:
We note that the reviewer is an expert in tide modelling and hope we could spark his interest in ice rheology with this paper.

Comment L42:
There was a hick up with the ordering of the figures. We rearranged.

Comment L43:
Changed as suggested

Comment L44:
We have rounded the floats to their fifth decimal.

Comment L45:
We have added a few sentences and elaborate on the low tide bias.

Comment L46:
Changed as suggested earlier

Comment L47:
Changed as suggested

Comment L48:
We have removed 'inner' and replaced 'near Diamond Hill'
Comment L49:
Changed as suggested

Comment L50:
Oli: why didn't we use COSMO-SkyMed ? But we could

Comment L51:
Changed as suggested

Comment L52:
The reviewer is right, problems are minimized. We reworded and changed 'avoids' to 'eases'.

Comment L53:
We have removed the '-' from all percentage improvements in the manuscript.

**Differential InSAR for tide modelling in Antarctic ice-shelf grounding zones**

Christian T. Wild[1], Oliver J. Marsh[1,2], and Wolfgang Rack[1]

[1]Gateway Antarctica, University of Canterbury, Private Bag 4800, Christchurch 8140, New Zealand
[2]British Antarctic Survey, High Cross, Madingley Road, Cambridge, CB3 0ET, United Kingdom

**Correspondence:** Christian T. Wild (christian.wild@canterbury.ac.nz)

**Abstract.** Differential interferometric synthetic aperture radar (DInSAR) is an essential tool for detecting ice-sheet motion near Antarctica's oceanic margin. These space-borne measurements have been used extensively in the past to map the location and retreat of ice-shelf grounding lines as an indicator for the onset of marine ice-sheet instability and to calculate the mass balance of ice sheets and individual catchments. The main difficulty in interpreting DInSAR is that images originate from a combination of several SAR images and do not indicate instantaneous ice deflection at the time of satellite data acquisition. Here, we combine the sub-centimetre accuracy and spatial benefits of DInSAR with the temporal benefits of tide models to infer the spatiotemporal dynamics of ice-ocean interaction during the times of satellite overpasses. We demonstrate the potential of this synergy with TerraSAR-X data from the almost stagnant Southern McMurdo Ice Shelf. We then validate our algorithm with GPS data from the fast-flowing Darwin Glacier, draining the Antarctic Plateau through the Transantarctic Mountains into the Ross Sea. We are able to [c1]reconstruct DInSAR derived vertical displacements to 7 mm [c2]MAE; generally improve traditional tide model output by up to [c3]39% from 10.8 cm to 6.7 cm RMSE against GPS data from areas where ice is in local hydrostatic equilibrium with the ocean; and up to [c4]74% from 21.4 cm to 5.6 cm RMSE against GPS data in feature-rich coastal areas where tide models have not been applicable before. Numerical modelling then reveals a Young's modulus of $E = 1.0 \pm 0.56$ GPa and an ice viscosity of $\nu = 10 \pm 3.65$ TPa s when finite-element simulations of tidal flexure are matched to 16 days of tiltmeter data; supporting the hypothesis that strain dependent anisotropy may significantly decrease effective viscosity compared to isotropic polycrystalline ice on large spatial scales. Applications of our method [c5]include: [c6]refining coarsely-gridded tide models to resolve small-scale features at the spatial resolution and vertical accuracy of SAR imagery[c7]; [c8]separating elastic and viscoelastic contributions in the satellite derived flexure measurement[c9]; and [c10]gaining information about large-scale ice heterogeneity in Antarctic ice-shelf grounding zones, the missing key to improve current ice-sheet flow
* * *
[c1]

[c2] *Text added.*

[c3]

[c4]

[c5]

[c6]

[c7]

[c8]

[c9] *Text added.*

[c10]

models. The reconstruction of the individual components forming DInSAR images has the potential to become a standard remote-sensing method in polar tide modelling. Unlocking the algorithm's full potential to answer multi-disciplinary research questions is desired and demands collaboration within the scientific community.

*Copyright statement.* TEXT

**1 Introduction**

The periodic rise and fall of the ocean's surface is caused by the gravitational interplay of the Earth-Moon-Sun system and our planet's rotation. Knowledge of ocean tides is fundamental to fully understand oceanic processes, sedimentation rates and behaviour of marine ecosystems. In Antarctica, the tidal oscillation also controls the motion of ice sheets near the coastline and ocean mixing in the sub ice-shelf cavity which modifies heat transport to the ice-ocean interface (Padman et al., 2018).

10 SAR satellites repeatedly illuminate Earth's surface and record the backscattered radar wave. While the SAR signal's amplitude depends on reflection intensity and is mainly characterized by [c1]physical and electrical properties of the surface, the recorded phase holds information about the distance travelled by the signal (Massom and Lubin, 2006). Two-pass interferometry (InSAR) can be used to determine [c2] surface motion with sub-centimetre accuracy over vast remote areas. [c3]Therefore, InSAR has been applied to measure [c4]Antarctic-wide ice velocity (Mouginot et al., 2019) and to observe tidal strain of landfast

15 sea ice (Han and Lee, 2018). In grounding zone areas, where an ice sheet comes in contact with the ocean for the first time and forms floating glaciers and ice shelves, InSAR has become the state-of-the-art practice to measure the flux divergence of ice-flow velocity (Mouginot et al., 2014; Han and Lee, 2015) and thus the mass balance of many ice shelves around Antarctica (Rignot et al., 2013). InSAR can also be used to identify vertical deflection due to ocean tides. Horizontal and vertical motion cannot be distinguished in single interferograms but the unsteady tidal contribution can be extracted by [c5]using triple

20 or quadruple combinations of SAR images. This is based on the assumption that horizontal flow is time invariant, and that its phase contribution therefore cancels out. The double-differential measurement of vertical displacement only is known as Differential InSAR (DInSAR). While DInSAR has often been applied to detect the grounding line movement around Antarctica (Konrad et al., 2018) the signal can also be used to measure spatial variability of ocean tides at very high ground resolution (Minchew et al., 2017; Baek and Shum, 2011). This second application is complicated by the fact that DInSAR interferograms

25 show a combination of multiple stages of the tidal oscillation. Tidal migration of the grounding line as well as viscoelastic time delays in ice displacement, tidally-induced velocity variations and geometric effects on the surface flexure also complicate the correct interpretation of DInSAR interferograms to date (Rack et al., 2017; Wild et al., 2018).
* * *
[c1] *Text added.*
[c2]
[c3]
[c4]
[c5]

Present-day displacement measurements by interferometry are exacerbated by the requirement of phase unwrapping, which is the most crucial processing step in any InSAR method. Discontinuities in the fringe pattern can cause jumps in the unwrapped phase and may therefore bias the continuous motion field. Due to these complications, only very few studies have attempted to derive a tide model from DInSAR[c6]. Minchew et al. (2017) developed an unprecedented spatially and temporally dense SAR

5  data acquisition campaign for the Rutford Ice Stream, Weddell Sea. Their novel Bayesian method to unequivocally separate a complete set of [c7] tidal harmonics from nontidal ice-surface variability is unique, but beyond the [c8]sparse data availability for the remainder of Antarctica. Baek and Shum (2011) succeeded in using data from the ERS-1/2 tandem mission to map the dominant tidal [c9]constituent (O1) in Sulzberger Bay, Ross Sea, but data limitations prevented them from developing a full tidal model. In this case, too short a time span (71 days) [c10]of the ERS tandem mission eliminated a [c11]sufficient change of the

10  observed tidal amplitudes as [c12]aliasing problems between the repeat-pass cycle of the SAR satellites [c13]and the tidal oscillation masked the sensor's sensitivity to tidal variability. [c14]As a result, many tidal constituents had their variability too poorly sampled to derive a full set of tidal harmonics. [c15]However, even identifying only dominant single tidal constituents is valuable as it indicates ways in which tide models need to be changed, and these changes will ultimately improve modelling of other constituents. In the Ross Sea, the tidal oscillation is dominated by diurnal harmonics (Padman et al., 2018). [c16]

15  Tide models can be consulted to predict both the timing and magnitude of the dominant harmonics. Numerous tide models of various spatial scales (global vs regional) and complexity have been developed (see Stammer et al., 2014, for an overview). While forward models integrate the equations of fluid motion subjected to a gravitational forcing over time, inverse models assimilate measurements of vertical displacement from laser altimetry, tide gauges and GPS (Egbert and Erofeeva, 2002; Padman et al., 2003a, 2008). Since the modelled physics is generally simple and gravitational forces are well known, tide

20  model predictions are of high quality in areas where ice is freely floating on the ocean[c1]. In coastal areas, [c2]tide models are prone to inaccuracies due to errors in model bathymetry, grounding-line location and insufficient knowledge of the ice water drag coefficient (Padman et al., 2018). Another source of error arises from the conversion of ice-shelf draft to ice-shelf thickness and subsequent estimation of water-column thickness. This freeboard conversion assumes that ice near the coastline is in local hydrostatic equilibrium, whereas stresses from the grounded ice clearly prevent a freely-floating state. A bias of the hydrostatic
* * *
[c6] :

[c7]

[c8] *Text added.*

[c9]

[c10] *Text added.*

[c11] *Text added.*

[c12] *Text added.*

[c13] *Text added.*

[c14] *Text added.*

[c15]

[c16]

[c1] *Text added.*

[c2]

solution towards thicker ice (Marsh et al., 2014), and therefore a thinning water-column thickness, may negatively affect the tidal prediction. In summary, the relatively coarse spatial resolution and underlying assumptions of contemporary tide models introduce inaccuracies especially in feature-rich coastal areas such as fjord-type outlet glaciers. Although average tide model accuracy has improved markedly in coastal areas over one decade, from about $\pm 10$ cm (Padman et al., 2002) to $\pm 6.5$ cm

5 (Stammer et al., 2014), they are still a[c3]n order of magnitude larger than the sub-centimetre accuracy of DInSAR (Rignot et al., 2011). For this reason, we consider DInSAR as the absolute truth and use these space-borne measurements to correct tide-model output.

In this manuscript, we show how the spatial benefits and high accuracy of DInSAR can be used to refine coarse resolution tide models to adequately resolve ocean tides along the feature-rich Antarctic coastline. First we introduce the necessary data

10 set[c1]and describe the preprocessing. [c2] Second we test the algorithm for the Southern McMurdo Ice Shelf (SMIS), a small and almost stagnant ice shelf with a simple grounding-zone geometry, and expand the study to the Darwin Glacier, a relatively fast-flowing outlet glacier within a complex fjord-like embayment. We validate our results with dedicated field measurements taken within the Transantarctic Ice Deflection Experiment (TIDEx) in 2016. We then demonstrate how this exercise can also be applied to reveal errors in interferometric phase unwrapping and answer fundamental questions about the physical properties

15 of ice in Antarctic glaciology.

**2  Methodology**

**2.1  Summary of SAR image processing**

To develop the method, we use 11-day repeat-pass TerraSAR-X data in StripMap imaging mode. The satellite acquires X-band radar data (wavelength 3.1 cm, frequency 9.6 GHz) with a ground resolution of slightly below 3x3 m and images covering

20 an area of 30x50 km [c3]in look and azimuth direction. We calculate vertical surface displacement due to ocean tides using the Gamma software package (Werner et al., 2000). InSAR and DInSAR image combinations are generally chosen so that a later image is always subtracted from an earlier image. For image triplets, the central SAR image serves as a common reference during the co-registration. We then correct the resulting DInSAR interferograms for apparent vertical displacement due to horizontal surface motion (Rack et al., 2017) using the method presented [c4]by Wild et al. (2018).

25 ### 2.2  Tide models

The predictions of five tide models are validated: the regional barotropic models (1) Circum-Antarctic Tidal Solution ([c5]CATS2008) developed by Padman et al. (2008), (2) Ross Sea Height-Based Tidal Inverse Model (Ross_Inv_2002) developed by Padman
* * *
[c3] *Text added.*

[c1] ;

[c2] and guide through the work flow.

[c3] *Text added.*

[c4] in

[c5] CATS2008a_opt

et al. (2003a), (3) Ross Sea assimilation model (Ross_VMADCP_9cm), (4) the fully global barotropic assimilation model (TPXO7.2) from Oregon State University developed by Egbert and Erofeeva (2002), and (5) the ([c6]T_TIDE) prediction of GPS data from freely-floating areas following the harmonic analysis of Pawlowicz et al. (2002) which is based on [c7]FORTRAN codes developed by Foreman (1977). The [c8]T_TIDE software is a widely-used toolbox for performing classical harmonic analysis of ocean tides. It can [c9]be used to analyse [c10] time series records [c11]from GPS, tiltmeter and other sensors which are evenly spaced and adequately resolved. [c12]T_TIDE outputs the amplitude and phase of its dominant harmonics with error estimates, along with a tidal prediction that [c13]can extend beyond the duration of the input record. The isostatic deformation of the Earth's lithosphere underneath the moving water masses is modelled using TPXO7.2 load tide model (Egbert and Erofeeva, 2002), which itself is based on 13 tidal constituents and added to all tide model predictions except [c14]T_TIDE. In addition to the tidal motion underneath the floating ice, much of the ice-surface variability can be attributed to the Inverse Barometric Effect (IBE, Padman et al., 2003b). A +1 hPa anomaly of atmospheric pressure translates to an instantaneous -1 cm change on the ice-shelf surface. [c15]Note that we did not apply a running mean to the pressure records, as the application of any window length worsened the fit to available GPS data. To correct for the IBE outside the GPS period, we make use of atmospheric pressure records obtained by nearby automatic weather stations on Ross Island (Scott Base AWS) and the Ross Ice Shelf (Marilyn AWS). We validate these records with separate barometric measurements taken within the TIDEx campaign and find very good agreement.

In this paper, we use the terms 'traditional tide [c1]modelling' or 'tide model' to refer to the sum of ocean-tide, load-tide model outputs and the IBE. Freely-floating areas of ice shelves and glaciers are expected to experience the full oscillation of this tide model. Traditional tide [c2]modelling, however, neglects ice mechanics in grounding zones where tidal flexure [c3]significantly affects the surface elevation signal in reality. Other signals that change sea-level height such as mean dynamic topography and storm surges are also excluded from this type of tide model.

**2.3 In-situ data**

We set [c4]up a number of GPS receivers to measure ice-surface motion at millimetre accuracy and high temporal resolution (Scheffler, 2017). Although we used GPS data from the freely-floating parts to develop local tide models using [c5]T_TIDE, all GPS data was only used for validation purposes and did not feed into the algorithm. GPS measurements were differentially
* * *
[c6] t_tide
[c7] *Text added.*
[c8] t_tide
[c9] *Text added.*
[c10] any
[c11] *Text added.*
[c12] and
[c13] is free from non-tidal effects
[c14] t_tide
[c15] It is noteworthy to mention
[c1] _
[c2] _
[c3] significally
[c4] _
[c5] t_tide

corrected using static base stations to increase their spatial accuracy. We also installed an array of seven tiltmeters to record surface flexure over 16 days across the grounding zone to [c6]constrain the physical properties of Antarctic ice. The tiltmeters were complemented by a dense network of point measurements of ice thickness using [c7]an autonomous phase-sensitive radar echo sounder (ApRES, Nicholls et al., 2015).

**2.4 Combining DInSAR and tide models**

To allow a correct interpretation of DInSAR images covering grounding zones, it is desirable that tide models replicate DInSAR observations in freely-floating areas. We first adjust the tide-model output to match the highly-accurate DInSAR measurements using a least sum of squares routine (Wild et al., 2018). By doing so, we consider just the tide model amplitude to contain errors. Possible tide-model phase errors are [c1]only smallish and accounted for by adjusting the absolute amplitude and thus the rate of tidal change during the times of SAR data acquisition. Second we build on earlier work by Han and Lee (2014) [c2]to develop an empirical displacement map showing tide-deflection ratio throughout the satellite image ($\alpha$-map). By feeding the $\alpha$-map with the adjusted tide model output, the 'point forecast' is then spatially extended to predict the mean vertical displacement for every pixel at the times of SAR data acquisition. We then perform the double differences of the empirical model corresponding to the SAR image combinations used to generate the DInSAR images. The original DInSAR satellite measurements are subsequently removed from the mean DInSAR images to calculate their misfits, $\mu$. We now compute the least-squares solution to the equation $Cx = b$ such that the 2-norm $|b - Cx|$ is minimized. Here, $C$ is the $m \times n$ DInSAR matrix of SAR image combinations with $m$ rows of SAR images and $n$ columns of coherent DInSAR interferograms; $b$ is a vector of $\alpha$-prediction misfits and $x$ the least-squares solution of this system of linear equations. The values of $x$ correspond to how much an $\alpha$-prediction deviates from the 'real' vertical displacement at the times of SAR data acquisition. We therefore subtract these offsets, $\Delta A$, from the $\alpha$-prediction maps.

We now demonstrate the workflow in one spatial dimension with an example of the Southern McMurdo Ice Shelf (78°15' S, 167°7' E, SMIS). In this study area, we derived 9 DInSAR images from 12 TerraSAR-X scenes in 2014 (Rack et al., 2017). The low number of DInSAR images is a consequence of the SAR scenes being acquired on 3 different satellite tracks[c3], which limits the number of coherent SAR scenes for DInSAR. The resulting system of linear equations is therefore underdetermined as there are more offsets, $\Delta A$, than misfits, $\mu$, to constrain the least-squares solutions. We choose a pixel on the freely-floating end of a profile through the ice-shelf grounding zone to represent the unrestricted ice shelf movement and calculate the percentage vertical displacement of every other pixel from this location. Averaged over the 9 DInSAR interferograms this pixel retains 100% vertical displacement (red areas in Fig. 1), while grounded areas experience zero net uplift (purple areas). Individual pixels on the freely-floating part of the SMIS may show $\alpha$-values slightly above 100%. [c4]Flexure in the grounding zone can
* * *
[c6]

[c7]

[c1]

[c2]

[c3] *Text added.*

[c4] *Text added.*

include 'overshoot' for snapshots in time (Fricker and Padman, 2006)[c5], but these are averaged out during the calculation of the $\alpha$-map.

We now extract the $\alpha$-values along the profile from the $\alpha$-map. This $\alpha$-profile can be multiplied with the individual DInSAR measurements on its freely-floating end which results in empirically derived $\alpha$-predictions (Fig. 2 center). These mean predic-
5 tions do not perfectly replicate the DInSAR measurements (Fig. 2 top). Their misfits, however, show a very systematic pattern (Fig. 2 bottom). It is desirable to find a combination of offsets that have the least deviation from the $\alpha$-predictions. We therefore hypothesize that this rather systematic signal can be reconstructed using a least-squares strategy. We solve the underdetermined system simultaneously by finding the combination of offsets that result in a minimal sum of squares. The reconstructed offsets must then be removed from the $\alpha$-prediction for the times of SAR data acquisition (Fig. 3 top). The computed least-square
10 offsets generally replicate the pattern of the misfits (Fig. 3 center) and result in smooth displacement profiles in the ice-shelf grounding zone (Fig. 3 [c1]bottom).

**3 Results**

In this section we apply the workflow in two spatial dimensions to the Darwin Glacier ($79°53$' S, $159°00$' E). In this study area, we derived a total of 45 DInSAR images from 12 SAR scenes being acquired on the same satellite track (Tab. 1). [c2]To
15 ensure a balance of both SAR coherence between consecutive images and coverage of different tidal periods in different seasons, we acquired the SAR imagery in two blocks, separated by an 11 week gap. SAR image combinations were generally chosen consecutively so that a later image is always subtracted from an earlier image. For image triples, the central image was taken as a common reference/master image. Additionally the data gap between SAR 8 and 9 was taken into account (no 8-9 combination as loss of coherence over this relatively long interval). The advantage of using every other remaining combination
20 (Tab. 2) is that more double-differential measurements of tidal amplitude are available for the least-squares fitting algorithm than only using consecutive pairs alone. The system of linear equations is then overdetermined. A dedicated field campaign was conducted in the Darwin Glacier grounding zone in 2016 and *in-situ* data is available for numerical modelling and field validation purposes. In contrast to the simple geometry at the SMIS, the Darwin Glacier consists of a feature-rich embayment that is constrained by steep topography at its margins. Additionally a buttressing ice rise to the Ross Ice Shelf restricts outflow
25 in the North.

**3.1 Reconstruction of displacement maps during satellite overpasses**

From the interferogram dataset we identify a corridor of only about 2 km width along the centerline where the glacier can be assumed to be freely floating (Fig. 1). This area is expected to experience the full oscillation predicted from tide models. We run five tide models to predict the tidal oscillation at the GPS station 'Shirase' over the time span of SAR data acquisitions.
30 Here we use atmospheric pressure data from the automatic weather station 'Marilyn' which is located about 120 km away on
* * *
[c5] *Text added.*
[c1] *Text added.*
[c2] *Text added.*

the Ross Ice Shelf to correct for the inverse barometric effect. This record correlates well (Pearson's correlator 0.989) with a mean of seven barometers installed over 14 days across the Darwin Glacier during the TIDEx campaign. All tide-model predictions show a clear fortnightly occuring spring-neap tidal cycle which is superimposed by a dominant diurnal signal (Fig. 4). [c3]The approximately fortnightly spring/neap tidal cycle is primarily determined by the difference in frequency between the dominant K1 and O1 diurnal constituents.

We apply [c1]T_TIDE to our 16 day record of the 'Shirase' GPS to test the potential of short-term GPS surveys to improve current Antarctic tide models. The problem with using such a short window to determine a full set of tidal constituents results from the interplay of the lunar diurnal tide (K1, 23.93 h) with the solar diurnal tide (P1, 24.07 h) as they are close in frequency and P1 has an amplitude of $15 - 20$ % of K1. Without accounting for their inference, [c2]T_TIDE just extracts an apparent K1 from a 16 day record that is really K1+P1. As a consequence, the K1 tide from our harmonic analysis can vary by $30 - 40$ % over a 6-month period and its amplitude is only controlled by the exact time that the GPS data was acquired within the K1+P1 modulation cycle. Additionally, harmonic decomposition of GPS data is subject to inaccuracies itself with errors in both the extracted amplitudes and phases. These errors were found to be of the same magnitude as the K1+P1 inference. For this reason, we did not use inference to separate K1 and P1 (or similarly to separate the semi-diurnal S2 and K2 constituents), but perform a thorough analysis on the identified uncertainty range. While the analysis captures the dominant K1 constituent in the Ross Sea within a reasonable signal-to-noise ratio, fortnightly harmonics could not be retrieved adequately from this time series alone. The [c3]T_TIDE prediction is therefore the least accurate tide model and requires the largest adjustment to match DInSAR (Tab. 1). Although all the corrected tide model outputs now replicate our DInSAR measurements, their rate of tidal change is affected by the adjustment. Offsets computed for the Ross_Inv_2002 tide model are generally below 10 cm, whereas other tide models require adjustments of up to 13.3 cm (Tab. 1). This agrees well with the findings of Han et al. (2013), who find that the Ross_Inv_2002 model is the optimum tide model for the Terra Nova Bay with a 4.1 cm RMSE against 11 days of tide gauge data. We therefore choose Ross_Inv_2002 for numerical modelling purposes to minimize any effects on a viscoelastic model, but use TPXO7.2 to reconstruct vertical displacement at the times of satellite overpasses as it fits best to our GPS measurements. We refer to the Appendix for a validation of individual tide model output with GPS data from 'Shirase' (Fig. A1).

After the adjustment, modelled tidal amplitudes range from -0.966 m to 0.781 m over the whole SAR period (Fig. 4). Mean absolute residual error to the 45 DInSAR measurements at the tide-model location 'Shirase' is just 7 mm (Tab. 2), which can be explained by interferogram noise. We attribute this accuracy to the exceptionally high phase coherence of the TerraSAR-X data set. The reconstruction algorithm results in 12 smooth vertical displacement maps for the times of SAR data acquisition (Fig. 5).
* * *
[c3]

[c1] t_tide

[c2] t_tide

[c3] t_tide

**3.2 Validation with GPS measurements**

We now validate these reconstructions with available field data. As both GPS records overlap with the acquisition of SAR image 11, we extract the vertical displacement along the glacier's centerline and plot the profiles against the two GPS point measurements. The GPS measurement at 'Hillary' is 0.169 m, which is close to the reconstruction of 0.156 m. The 'Shirase' GPS measurement is 0.566 m, which is slightly above the reconstruction of 0.522 m. We attribute the deviations of +1.3 and +4.4 cm, respectively, to a combination of interpolation artefacts, temporal smoothing of the GPS data and residual errors of the least-squares algorithm. The overall shape of the vertical displacement is well reproduced as observed with both GPS measurements (Fig. 6).

**3.3 Applications**

**3.3.1 Tide-model refinement**

A map of tide-deflection ratio ($\alpha$-map) can be combined with the tide model to predict an average time series of vertical displacement between the times of SAR image acquisition. With this approach, the coarse grid of traditional tide models is refined to resolve small-scale features of vertical tidal displacement throughout the embayment. The $\alpha$-value for the pixel containing the 'Hillary' GPS station is 46.06%. We use this value and linearly scale the adjusted tide-model output for the location of the 'Shirase' GPS to predict vertical tidal motion within the flexure zone. This scaling maintains the tide model's high correlation (Pearson's correlator 0.95) with the 'Shirase' record, but improves the RMSE between the TPXO7.2 output and the 'Hillary' record from 21.4 cm to 5.6 cm, which corresponds to an improvement of [c1]74% to GPS data (Fig. 7). The primary reason for this large improvement, however, is that the tide model now takes the damping of the tidal signal by ice mechanics in the grounding zone into account.

**3.3.2 Ice heterogeneity**

With the 12 reconstructed displacement maps at hand, it is now possible to perform any image combination. We mosaic the 45 double differences corresponding to DInSAR combinations (Tab. 2) to allow a more direct comparison between measured and modelled interferograms. SAR image combinations were chosen so that the loss of coherence between SAR 8 and 9 was taken into account and that a maximum number of consecutive, double-differential interferograms was available for the least-squares fitting routine. The synthetic interferograms replicate not only simple tidal fringes as measured with DInSAR, but also show complicated viscoelastic signals within the grounding zone (Fig. 8). For an overall assessment of model performance we calculate again the misfit between each modelled and observed interferogram for every pixel, but this time after using the adjusted tide-model output. The standard deviation of these misfits is shown in Fig. 9, with the majority of the glacier surface below noise level of interferograms ($\sigma < 1.0$ cm). We identify a narrow band with higher standard deviations ($\sigma \approx 2.0$ cm)
* * *
[c1]

from the [c2] shear margin of the Darwin Glacier [c3]near Diamond Hill extending along flow direction onto its ice shelf. Standard deviations are largest [c4] in the shear zone of the fast-flowing Ross Ice Shelf and above steep rocky cliffs ($\sigma > 4.0$ cm) which is a result of poor phase coherence or layovers in SAR images in these areas.

**3.3.3 Detection of errors in phase unwrapping**

5   We now extend the earlier one dimensional analysis of the SMIS to a two dimensional re-analysis of the SMIS data set and calculate misfits of 9 DInSAR interferograms. Resulting standard deviations are generally smaller in this area ($\sigma < 0.3$ cm) and smoothly distributed throughout the map. We identify two regions of phase discontinuities between adjacent cells at the SMIS. Both extend from the center of an ice rise towards Black Island (cyan and green areas in Fig. 9) with $\sigma \approx 0.4$ cm and $\sigma \approx 0.5$ cm, respectively. We interpret these rapid [c1]spatial gradients as a proxy for errors in the DInSAR measurements, as the

10   modelled least-square interferograms originate from [c2]a bivariate spline approximation over a rectangular mesh which acts curvature minimizing. We re-evaluate the remote-sensing part of the analysis and find discontinuities in DInSAR interferograms ID:1 and ID:8 that match the [c3] two phase jumps in the standard deviation map. These discontinuities, in turn, occurred during phase unwrapping and can now be corrected.

**3.4 Finite-element modelling of viscoelasticity**

15   We hypothesize that any non-linear signal due to viscoelastic ice properties is significantly reduced or even completely lost during the averaging step to compute the $\alpha$-map. This signal can then be reconstructed by finding the offsets to match observations made with DInSAR. We therefore subtract the $\alpha$-prediction again from the 12 reconstructions to extract the theorised viscoelastic signal (Fig. 10). This signal is negligible at times during neap tide (SAR 4 and 9) but well pronounced for SAR images acquired during spring-tide periods (SAR 1,6,7,10 and 11).

20   In order to further explore this pattern, we now make use of the tiltmeter array and ApRES network of ice-thickness measurements at the Darwin Glacier (Fig. 11). We match the numerical solutions from two finite-element models to seven tiltmeter records, with the goal to derive information on the physical properties of Antarctic ice. Thereby, the Young's modulus, $E$, is a measure of ice stiffness and controls the width of the flexure region. The value for ice viscosity, $\nu$, influences the timing of the flexural response within the grounding zone (Wild et al., 2017). Two numerical models of ice-shelf flexure are employed. The

25   elastic approximation (Holdsworth, 1969; Vaughan, 1995; Schmeltz et al., 2002) as formulated by Walker et al. (2013):

$$kw + \nabla^2(D\nabla^2 w) = q, \tag{1}$$

where $w(t)$ is the time-dependent vertical deflection of the neutral layer in a plate, $\nabla^2$ is the Laplace operator in 2-D space and $k = 5$ MPa m$^{-1}$ a spring constant of the foundation which is zero for the floating part. The applied tidal force $q(t)$ is defined
* * *
[c2] inner

[c3] *Text added.*

[c4] out

[c1] increases

[c2] a curvature-minimizing polynomial interpolation

[c3] course of the

by:

$$q = \rho_{sw} g [A(t) - w], \tag{2}$$

with $g = 9.81\,\mathrm{m\,s^{-2}}$ the gravitational acceleration and $A(t)$ the time-dependent tidal amplitude given by the adjusted Ross_Inv_2002 tide model. We choose this model, in contrast to the TPXO7.2 model, for finite-element simulations to minimize any potential effects of tide-model adjustment on viscoelasticity (Tab. 1). The stiffness of the ice shelf is given by (Love, 1906, p. 443):

$$D = \frac{EH^3}{12(1 - \lambda^2)}, \tag{3}$$

where $E$ is the Young's modulus for ice, $H(x, y)$ our ice thickness map derived from ApRES point measurements and $\lambda = 0.4$ the Poisson's ratio. We compare the elastic model with the viscoelastic approach developed by Walker et al. (2013):

$$\frac{\partial}{\partial t} \left[ kw + \nabla^2 \left( D\nabla^2 w \right) \right] + \frac{Ek}{2\nu(1 - \lambda^2)} w = \frac{\partial}{\partial t} q + \frac{E}{2\nu(1 - \lambda^2)} q, \tag{4}$$

where $\nu$ is ice viscosity. The following boundary conditions are applied for both models: the upstream boundary of the model domains on the grounded portion are anchored rigidly ($w = 0, \nabla^2 w = 0$), the downstream boundaries on the freely-floating ice shelf are set free. The location of the tide model computation is constrained to be equal to the tidal oscillation ($w = A(t), \nabla w = 0$) and the grounding line is represented by a fulcrum ($w = 0$). Both models are implemented in two spatial dimensions to capture effects of complex grounding-line configuration on ice-shelf flexure (Wild et al., 2018). We then solve the models using the commercial finite-element software COMSOL Multiphysics. As tiltmeters measure slope, $w'$, along their longitudinal axis, we derive the models' solutions for vertical displacement, $w$, with respect to the $x$ and $y$ directions. This allows us to retrieve surface slopes components in easting and northing direction and to rotate them into the individual orientations of the tiltmeter sensors. With our 16 day tiltmeter records it is only possible to capture their diurnal components with confidence. Semi-diurnal, fortnightly and monthly harmonics have been removed from the tiltmeter time series and we focus further analysis only on the K1 component within the 16 day window. Therefore, we now make extensive use of the [c1]T_TIDE program to automatically extract the modelled K1 harmonics from the modelled surface slopes and compare them against the K1 constituents from the tiltmeters. We thereby take amplitude and phase errors that originate from the harmonic analysis of noisy tiltmeter records into account and find the best rheological parameters to match the elastic and viscoelastic models to these seven K1 components. Incorporating viscoelastic effects into the model simulations always improves the elastic fit to the tiltmeter data within the uncertainty range of K1 amplitude and phase (Tab. 3). We find that an average Young's modulus of $E = 1.0 \pm 0.56\,\mathrm{GPa}$ and an ice viscosity value of $\nu = 10 \pm 3.65\,\mathrm{TPa\,s}$ fits best to our measurements within uncertainty (Fig. 12). The viscoelastic model gives an average RMSE of [c2]0.00119 ° to the seven tiltmeters and improves on the elastic approximation with an average RMSE of [c3]0.00147 ° by [c4]≈ 20%.
* * *
[c1]

[c2]

[c3]

[c4]

**4 Discussion**

**4.1 Seasonal bias in $\alpha$-map**

Due to the alignment of the satellite overpasses with the dominant diurnal tidal constituents in the Ross Sea, the observed stage of the tidal oscillation varies only slowly throughout the year. In the austral winter months, TerraSAR-X images have been acquired during stages of low tide, whereas satellite overpasses concur with stages of high tide during the austral summer months. The first 8 snapshots of our SAR data acquisitions for the Darwin Glacier show conditions at low tide and only the last 4 are acquired during high tide. [c1]For this reason, 21 of our 45 DInSAR images result from low-tide SAR combinations and only 3 from purely high-tide SAR combinations. The remaining 21 DInSAR images result from a combination of low and high-tide SAR snapshots (Tab. 2). Our $\alpha$-map, as an average of all 45 DInSAR images, ignores this seasonality and may therefore have a low-tide bias. As a result, the contribution of a [c2]possible tide induced landward migration of the grounding line may be affected by the averaging process. The seasonal bias would then modify the scaling of the tide model within the flexure zone. This is supported by the finding that low-tide stages in the 'Hillary' GPS record are matched closely by the scaled tide model, but peaks during high-tide stages are still over[c3]-estimated (Fig. 4)

**4.2 Viscoelasticity between snapshots**

Similarly, the linear scaling using an $\alpha$-map only modifies the predicted tidal amplitude, but neglects a viscoelastic time delay in the flexural response towards the grounding line. Wild et al. (2017) found that viscosity is most pronounced in the diurnal tidal components. Harmonic analysis of our GPS records reveals that the diurnal K1 and O1 constituents at 'Hillary' are lagging approximately 20 mins behind 'Shirase'. This signal is currently disregarded in the scaling work flow as ice is treated as a perfect elastic material that transfers tidal forcing instantaneously in the flexure zone. This assumption, however, allowed us to improve the accuracy of the predicted displacement by [c4]74%. Currently, the viscoelastic signal can only be reconstructed for the times of SAR data acquisition. Including viscoelasticity between times of satellite overpasses offers a small, but systematic, opportunity for further refinement. In our study area, the rate of tidal change is up to 10 cm hr$^{-1}$ (Tab. 1) and the viscoelastic misfit corresponding to 20 minutes time delay is therefore up to about 3 cm.

When separating the viscoelastic contribution from the reconstructed maps of vertical displacement at times of satellite overpasses, we assume that an $\alpha$-prediction corresponds to an instantaneous elastic response. This is justified by viscoelasticity being most pronounced when rates of tidal change are maximal. By expressing the viscoelastic misfits in percent of prevailing tidal amplitude during the times of satellite overpasses, the areas of pronounced viscoelastic effects can be visualised. They are most pronounced within the Darwin Glacier's shear zone (Fig. 13).

When predicting rates of tidal change using the adjusted Ross_Inv_2002 tide model, we identify a threshold of $\dot{A} \approx \pm 0.05$ m h$^{-1}$ (SAR times 1, 2, 7, 8, 10, 11 in Tab. 1) above which viscoelasticity is well represented in the reconstructed
* * *
[c1] *Text added.*
[c2] *Text added.*
[c3] *Text added.*
[c4] -

vertical displacement maps (panels 1, 6, 7, 8, 10 and 11 in Fig. 10). Image 6 is thereby an exception, as the [c5] [c6]adjustment of the Ross_Inv_2002 tide model was large (-0.082 m), which affects the viscoelastic model. We find that the error due to viscoelasticity on the floating part of the ice shelf increases with the absolute rate of tidal change (Fig. 13). SAR images acquired during periods of spring tides at the Darwin Glacier show also a significant viscoelastic contribution that diminishes during

5 neap tide periods. These independent observations from satellite data alone support our suggested threshold of $\pm 0.05$ m h$^{-1}$ for the separation of elastic and viscoelastic signals, as derived from tiltmeter data on the Southern McMurdo Ice Shelf presented in an earlier study (Fig. 8 in Wild et al., 2017). The advantage of separating the elastic from the viscoelastic contribution to the tidal flexure pattern is the large potential for improving current inverse modelling techniques to determine grounding-zone ice thickness from DInSAR measurements alone. Hereby, an elastic model is currently employed to optimize grounding-zone

10 ice thickness to match the surface flexure from DInSAR. The applicability of an elastic model varies from location to location as effective viscosity is dependent on ice temperature and shear stress (Marsh et al., 2014). Our method to separate the two contributions to the flexure pattern may therefore help to remove the viscoelastic contamination and allow purely elastic inverse modelling. Such an analysis, however, goes beyond the scope of this manuscript and will be published elsewhere.

**4.3 Large-scale ice anisotropy**

15 Fast-moving glacial environments like the Darwin Glacier are subject to large deformation by flow convergence and divergence, ice compression and extension, lateral shearing at the margins accompanied by fracture under tension and rapid thinning by basal melt. With cumulative deformation, a crystallographic fabric evolves that reflects the glacier's flow history (Alley, 1988), and with it strain-dependent mechanical anisotropy of ice. [c1]The map of average error due to viscoelasticity, Fig. 13, shows a narrow band of larger [c2]errors extending from the Darwin Glacier's [c3] shear margin [c4]near Diamond Hill out towards the freely-

20 floating ice shelf. As preferred crystallographic orientation develops with strain, effective viscosity decreases of about a factor of ten compared to initially isotropic polycrystalline ice (Hudleston, 2015). Our analysis of tiltmeter data reveals a five-fold reduced viscosity at the very dynamic Darwin Glacier compared to an earlier study at the almost stagnant Southern McMurdo Ice Shelf (Wild et al., 2017). We hypothesize that this microscopic process explains the macroscopic response observed here, and accounts for the measured glacial heterogeneity within the embayment. Large scale observations of ice anisotropy, in

25 turn, are currently the missing key to improve parametrisations to account for polar ice anisotropy in ice-sheet flow modelling (Gagliardini et al., 2009).

Other processes have been proposed which lead to ice softening in areas with high strain rates. Thermomechanical modelling suggests that shear heating and consequent thermal softening reduces lateral drag in ice-stream margins (Perol and Rice, 2015). Fracture modelling implies that damage reduces ice viscosity along confined crevassed zones with consequences on ice-shelf

[c5] used
[c6] Ross_Inv_2002 tide model was largely adjusted
[c1] The standard deviation map, Fig. 9,
[c2] misfits
[c3] inner
[c4] *Text added.*

scale (Albrecht and Levermann, 2014). Full-Stokes viscoelastic modelling shows that Glen's non-linear flow law and tidal stresses in the ice-shelf flexure zone are sufficient to explain large-scale temporal variations in ice dynamics (Rosier and Gudmundsson, 2018). These processes, or a combination of them, might certainly be at play but they do not explain why a band of higher standard deviations can be observed in the shear zone of the Darwin Glacier which is absent in the flexure zone

5 of the SMIS (Fig. 9). We therefore attribute this difference to ice-fabric reorientation in the shear margin.

**4.4 Refining tidal constituents using DInSAR**

The idea of using SAR interferometry to derive a full set of tidal harmonics was first laid out in a study of tides in the Weddell Sea (Rignot et al., 2000). The authors discussed that DInSAR images cannot be transformed into individual displacement fields because of the nonuniqueness of the inversion. A large number of independent DInSAR images is required to overcome this

10 problem and to resolve the phase of tidal constituents that are close to the repeat-pass of the SAR satellite. For example, multiples of the lunar diurnal constituent K1 (23.93 h) are relatively close to the exact integer repeat-pass of TerraSAR-X (11 days)[c1], meaning that the observed amplitude of the K1 constituent is only varying once throughout the year. Consequently[c2], SAR images need to be acquired at least over the duration of one year to provide some redundancy for the inversion step of DInSAR images to tidal constituents. However, with an exact 12 h period, the stage of the semidiurnal solar tide, S2, will always be

15 the same at each satellite pass making[c3], TerraSAR-X and similar satellites with repeat [c4]passes of integer days blind to the S2 constituent. For example Minchew et al. (2017) needed a unique spatially and temporally dense SAR acquisition campaign as well as *a-priori* knowledge of the temporal basis functions from GPS data to empirically determine tidal constituents on Rutford Ice Stream. The four COSMO-SkyMed satellites in orbit, however, produce repeat-passes of 1, 3, 4 and 8 days and are blind to the S2 constituent as well even when using $> 1000$ available DInSAR images. Although other dominant tidal

20 constituents like M2 (12.4 h) and O1 (25.82 h) were inferred successfully, the method presented here can achieve a higher accuracy [c5]of the total tide with [c6]fewer DInSAR images. [c7]

**5 Conclusions and Outlook**

Accurate prediction of ocean tides in coastal areas [c8]around Antarctica is crucial as the majority of Antarctica's ice is discharged through large outlet glaciers. We presented a data fusion method between DInSAR and traditional Antarctic tide models to

25 predict spatial variability of tidal motion near the grounding line. The primary value of using DInSAR in conjunction with tide models lies in the spatio-temporal benefits of resolving complex grounding zone deformation. Their symbiosis not only
* * *
[c1] *Text added.*

[c2] *Text added.*

[c3] *Text added.*

[c4]

[c5]

[c6]

[c7]

[c8] *Text added.*

improves current accuracies of the predicted tidal amplitudes in coastal regions generally, but also [c9]eases issues related to the timing of the tidal wave and the sun-synchronous satellite orbit when attempting to derive tide-models from SAR data alone. In our study area, the method presented in this paper improves traditional tide modelling in average by [c10]22% from 11.8 cm to 9.3 cm RMSE against 16 days of GPS data. The GPS station 'Shirase' on the freely-floating part of the Darwin Glacier has proven invaluable to determine which tide model has to be used to best reconstruct the vertical displacement during satellite overpasses. For the Darwin Glacier, the TPXO7.2 tide model predicts best the tidal oscillation. With using DInSAR measurements to adjust the TPXO7.2 tidal prediction, its RMSE could be improved by [c11]39% from 10.8 cm to 6.7 cm.

Our GPS record from 'Shirase' is too short to improve already available Antarctic tide models. A longer record is required to adequately resolve a full set of tidal constituents. We produced an empirical displacement map from DInSAR for tidal deflection ($\alpha$-map). Comparison of a GPS record within the tidal flexure zone [c1]('Hillary'[c2]) with predicted vertical displacement from feeding the $\alpha$-map with the adjusted TPXO7.2 tide model shows a [c3]74% improvement over using the tide-model output alone. This independent validation supports the finding that our method for making use of DInSAR is very useful for refining tide models in Antarctic grounding zones.

Numerical modelling of ice dynamics in Antarctic grounding zones commonly assumes that ice is isotropic and homogeneous i.e. of the same density and rheological properties throughout. Our analysis reveals that this assumption is valid for the Southern McMurdo Ice Shelf, an almost stagnant area with a simple grounding line configuration, but invalid for the Darwin Glacier, a fast-flowing outlet glacier with complex shear margins causing non-negligible ice heterogeneity within the embayment.

Further work is required in order to improve tide models in a larger variety of grounding zones by including effects of grounding-line migration and variability of horizontal ice flow.

*Code availability.* The code is freely available to the scientific community. Collaboration is anticipated and desired.

*Data availability.* TerraSAR-X data presented in this paper are subject to license agreements. GPS/tiltmeter and ApRES data are available upon request.

*Code and data availability.* No data sets, nor software, are part of this study.
* * *
[c9]
[c10] _
[c11] _
[c1] *Text added.*
[c2] *Text added.*
[c3] _

*Sample availability.* No samples were collected for this study.

**Appendix A: GPS evaluation of tide models**

The quality of the used [c1]tide model to correctly reconstruct tidal displacement at the times of SAR data acquisitions, is also crucial to accurately predict spatial variability in tidal motion for all times. Here, we assume that a freely-floating area on the [c2]ice shelf experiences the full oscillation as predicted from a tide model. In this area, however, tide-model output deviates from our DInSAR measurements. This indicates either that the area under investigation is prevented from a freely-floating state by lateral stresses within the embayment, or that the tide-model prediction is inaccurate for this area. We circumvent this ambiguity by making use of the high vertical accuracy of DInSAR and correct the tide-model prediction to match our satellite measurements. This raises the question of whether the adjustment improves or worsens the match to a 'real' tidal motion ? We therefore independently evaluate the pre- and post adjustment tide-model predictions and calculate their RMSE to 16 days of GPS data from the freely-floating area (Tab. A1). The adjustment improves all traditional tide model predictions by up to -39% for TPXO7.2, and only worsens the RMSE for the [c3]T_TIDE output by +11%, indicating that a harmonic analysis of GPS data can not be improved by using DInSAR for correction purposes. We choose TPXO7.2 for further processing as it displays the overall smallest RMSE (6.7 cm) and replicates the small-scale variability observed during the neap-tide period in the second half of our GPS record.

*Author contributions.* All authors conceived the study and conducted fieldwork. CW developed the algorithm, performed the data analysis and wrote a first version of the paper. OM and WR processed the SAR images for the Darwin Glacier and Southern McMurdo Ice Shelf, respectively. All authors finalized and approved the manuscript.

*Competing interests.* The authors declare that the research was conducted in the absence of any commercial or financial relationships that could be construed as a potential conflict of interest.

*Disclaimer.*

*Acknowledgements.* We thank Antarctica New Zealand for logistical support and the Scott Base staff for their dedication to the Transantarctic Ice Deflection Experiment. The project was supported by the Royal Geographic Society (Marsden Fast Start, PI O. Marsh) and by the Past Antarctic Science and Future Implications Program (PACaFI). D. Price, M. Ryan, D. Floricioiu and E. Scheffler contributed to fieldwork.
* * *
[c1]

[c2]

[c3]

TerraSAR-X data were provided through DLR project HYD1421. Landsat-8 images courtesy of the US Geological Survey. AWS data from Scott Base were provided by NIWA, and for Marilyn station by AMRC, SSEC, UW-Madison. The authors enjoyed fruitful discussions with H. Purdie and M. Sellier. CW also thanks R. Mueller for sharing oceanographic expertise. The thorough comments of one anonymous reviewer and L. Padman's valuable insight into tidal dynamics largely improved the paper. We also thank A. Robinson for editing.

[revised manuscript text omitted]

```
%% Copernicus Publications Manuscript Preparation Template for LaTeX
Submissions
%% ---------------------------------
%% This template should be used for copernicus.cls
%% The class file and some style files are bundled in the Copernicus
Latex Package, which can be downloaded from the different journal
webpages.
%% For further assistance please contact Copernicus Publications at:
production@copernicus.org
%%
https://publications.copernicus.org/for_authors/manuscript_preparation.ht
ml

%% Please use the following documentclass and journal abbreviations for
discussion papers and final revised papers.

%% 2-column papers and discussion papers
\documentclass[tc, manuscript]{copernicus}

\graphicspath{{./Figures/}}
\usepackage[finalnew]{trackchanges}
%finalold  - Reject all edits.
%finalnew  - Accept all edits.
%footnotes - Display edits as footnotes.
%margins   - Display edits as margin notes.
%inline    - Display edits inline.

\usepackage{booktabs}

\begin{document}

\title{Differential InSAR for tide modelling in Antarctic ice-shelf
grounding zones}

% \Author[affil]{given_name}{surname}

\Author[1]{Christian T.}{Wild}
\Author[1,2]{Oliver J.}{Marsh}
\Author[1]{Wolfgang}{Rack}

\affil[1]{Gateway Antarctica, University of Canterbury, Private Bag 4800,
Christchurch 8140, New Zealand}
\affil[2]{British Antarctic Survey, High Cross, Madingley Road,
Cambridge, CB3 0ET, United Kingdom}

%% The [] brackets identify the author with the corresponding
affiliation. 1, 2, 3, etc. should be inserted.

\runningtitle{Differential InSAR and tide modelling}

\runningauthor{Wild et al.}
```

\correspondence{Christian T. Wild (christian.wild@canterbury.ac.nz)}

\received{}
\pubdiscuss{} %% only important for two-stage journals
\revised{}
\accepted{}
\published{}

%% These dates will be inserted by Copernicus Publications during the
typesetting process.

\firstpage{1}

\maketitle

\begin{abstract}
Differential interferometric synthetic aperture radar (DInSAR) is an
essential tool for detecting ice-sheet motion near Antarctica's oceanic
margin. These space-borne measurements have been used extensively in the
past to map the location and retreat of ice-shelf grounding lines as an
indicator for the onset of marine ice-sheet instability and to calculate
the mass balance of \change{ice-sheets}{ice sheets} and individual
catchments. The main difficulty in interpreting DInSAR is that images
originate from a combination of several SAR images and do not indicate
instantaneous ice deflection at the time of satellite data acquisition.
Here, we combine the sub-centimetre accuracy and spatial benefits of
DInSAR with the temporal benefits of tide models to infer the
spatiotemporal dynamics of ice-ocean interaction during the times of
satellite overpasses. We demonstrate the potential of this synergy with
TerraSAR-X data from the almost stagnant Southern McMurdo Ice Shelf. We
then validate our algorithm with GPS data from the fast-flowing Darwin
Glacier, draining the Antarctic Plateau through the Transantarctic
Mountains into the Ross Sea. We are able to match DInSAR \add{derived
vertical displacements} to \change{0.84~mm}{7~mm}; generally improve
traditional \change{tide models}{tide model output} by up to -39\% from
10.8~cm to 6.7~cm RMSE against GPS data from areas where ice is in local
hydrostatic equilibrium with the ocean; and up to -74\% from 21.4~cm to
5.6~cm RMSE against GPS data in feature-rich coastal areas where
\change{contemporary tide-models are most inaccurate}{tide models have
not been applicable before}. Numerical modelling then reveals a Young's
modulus of \add{$E=1.0\pm0.56$~GPa} and an ice viscosity of
\add{$\nu=10\pm3.65$~TPa~s} when finite-element simulations of tidal
flexure are matched to 16 days of tiltmeter data; supporting the
\change{theory}{hypothesis} that strain dependent anisotropy may
significantly decrease effective viscosity compared to isotropic
polycrystalline ice on large spatial scales. Applications of our method
\change{range from}{include:} (i) refining coarsely-gridded tide models to resolve
small-scale features at the spatial resolution and vertical accuracy of
SAR imagery\change{, to}{} (ii) separating elastic and viscoelastic contributions
in the satellite derived flexure measurement; and (iii) gaining
information about large-scale ice \change{heterogenity}{heterogeneity} in
Antarctic ice-shelf grounding zones, the missing key to improve current

**Commented [L1]:** Relative to what? I really don't understand this "accuracy" metric

**Commented [L2]:** don't need these: you don't refer to list items by this code

ice-sheet flow models. The reconstruction of the individual components forming DInSAR images has the potential to become a standard remote-sensing method in polar tide modelling. Unlocking the algorithm's full potential to answer multi-disciplinary research questions is desired and demands collaboration within the scientific community.
\end{abstract}

\copyrightstatement{TEXT}

\introduction  %% \introduction[modified heading if necessary]
\label{Intro}
The periodic rise and fall of the ocean's surface is caused by the gravitational interplay of the Earth-Moon-Sun system and our planet's rotation. Knowledge of ocean tides is fundamental to fully understand oceanic processes, sedimentation rates and behaviour of marine ecosystems. In Antarctica, the tidal oscillation also controls the motion of ice sheets near the coastline and ocean mixing in the sub ice-shelf cavity which modifies heat transport to the ice-ocean interface \citep{padman2018ocean}.

SAR satellites repeatedly illuminate Earth's surface and record the backscattered radar wave. While the SAR signal's amplitude depends on reflection intensity and is mainly characterized by the surface, the recorded phase holds information about the distance travelled by the signal \citep{massom2006polar}. Two-pass interferometry (InSAR) can be used to determine lateral surface motion with sub-centimetre accuracy over vast remote areas, \change{Recently,}{and} InSAR has been applied to measure surface velocity of floating ice \citep[e.g.][]{tong2018multi} and to observe tidal strain of landfast sea ice \citep{han2018glacial}. In grounding zone areas, where an \change{ice-sheet}{ice sheet} comes in contact with the ocean for the first time and forms floating glaciers and \change{ice-shelves}{ice shelves}, InSAR has become the state-of-the-art practice to measure the flux divergence of ice-flow velocity \citep{mouginot2014sustained, han2015tide} and thus the mass balance of many \change{ice-shelves}{ice shelves} around Antarctica \citep{rignot2013ice}. InSAR can also be used to identify vertical deflection due to ocean tides. Horizontal \remove{motion} and vertical motion cannot be distinguished \change{at this stage}{in single interferograms} but the unsteady tidal contribution can be extracted by \change{differencing two separate InSAR pairs that originate from a}{DInSAR using } triple or quadruple combination\add{s} of \remove{three or four} SAR images. This \change{assumes that gravitational flow due to steady ice creep}{is based on the assumption that horizontal flow} is \change{time-invariant}{time invariant}, and that its phase contribution \change{can therefore be removed}{therefore cancels out}. The double-differential measurement of vertical displacement only is known as Differential InSAR (DInSAR). While DInSAR has often been applied to detect the grounding line movement around Antarctica \citep{konrad2018net} the signal can also be used to measure spatial variability of ocean tides at very high ground resolution. This second application is complicated by the fact that DInSAR interferograms show a combination of multiple stages of the tidal oscillation. Tidal migration of the grounding line as well as viscoelastic time delays in ice

**Commented [L3]:** what aspect of the surface are you referring to here?

**Commented [L4]:** ??? Can't get height changes from two-pass InSAR, right?

**Commented [L5]:** This doesn't seem like a fundamental cite for this statement

**Commented [L6]:** You use "DInSAR" before you define it. It *is* defined in the Abstract, but should be defined (again) in Main Text at first use.

**Commented [L7]:** Feels like you need a cite here. Maybe Minchew et al., Baek and Shum, or both

displacement, tidally-induced velocity variations and geometric effects
on the surface flexure also complicate the correct interpretation of
DInSAR interferograms to date
\citep{rack2017analysis,wild2018unraveling}.

Present-day displacement measurements \change{associated with
interferometric phase suffer from two limitations:}{by interferometry are
exacerbated by the requirement of phase unwrapping, which is the most
crucial processing step in any InSAR method.} \remove{As the absolute
number of waves in the received SAR signal cannot be measured, the phase
can only serve as a measure of relative distance change between two
images. Phase is, by definition, expressed as the fraction of a full wave
cycle that has elapsed relative to the origin with values between 0-
2$\pi$ in radians. Measurements of relative ground displacements between
satellite overpasses therefore require smooth phase unwrapping, the most
crucial processing step in DInSAR. Leaps between adjacent cells above 1
$\pi$, e.g. introduced by layover or}
\change{discontinuities}{Discontinuities} in the \change{initial SAR
images}{fringe pattern} can cause jumps in the unwrapped phase and may
therefore bias the continuous motion field.

Due to these complications, only very few studies have attempted to
derive a tide model from DInSAR.÷ \cite{minchew2017tidally} developed an
unprecedented spatially and temporally dense SAR data acquisition
campaign for the Rutford Ice Stream, Weddell Sea. Their novel Bayesian
method to unequivocally separate a complete set of \remove{energetic}
tidal harmonics from nontidal ice-surface variability is unique, but
beyond the data availability for the remainder of Antarctica.
\cite{baek2011antarctic} \change{failed to develop a full tide model from
the ERS-1/2 tandem mission, but succeeded in detecting the dominant tidal
constituent in Sulzberger Bay, Ross Sea.}{succeeded in using data from
the ERS-1/2 tandem mission to map the dominant tidal constituent (O1) in
Sulzberger Bay, Ross Sea, but data limitations prevented them from
developing a full tidal model.} In this case, too short a time span
(71~days) eliminated a change of the observed tidal amplitudes as the
repeat-pass cycle of the SAR satellites \change{masks}{masked} the
sensor's sensitivity to tidal variability. \add{However, even identifying
only the dominant dominant single tidal constituent is valuable÷ as it
indicates ways in which tide models need to be changed, and these changes
will ultimately filter into improve modeling of other constituents.} In
the Ross Sea, the tidal oscillation is dominated by diurnal harmonics
\citep{padman2018ocean}. An accurate inversion of TerraSAR-X data with an
exact integer number of \change{repeat-passes}{repeat passes} to a
complete set of tidal constituents is therefore not possible from DInSAR
measurements alone.

Tide models can be consulted to predict both the timing and magnitude of
the dominant harmonics. Numerous tide models of various spatial scales
(global vs regional) and complexity have been developed \citep[see][for
an overview]{stammer2014accuracy}. While forward models integrate the
equations of fluid motion subjected to a gravitational forcing over time,
inverse models assimilate measurements of vertical displacement from
laser altimetry, tide gauges and GPS \citep{egbert2002efficient,
padman2003tides, padman2008improving}. Since the modelled physics is

**Commented [L8]:** Remove paragraph break here?

**Commented [L9]:** This won't make sense to most readers. You will need to explain why this dataset is so special.

**Commented [L10]:** I really don't understand. (1) You haven't told us what the 71-day limit applies to (the record that Baek and Shum used? But now we don't need to be limited by that.) (2) Even in 71 days, tidal "amplitudes" (as in "instantaneous tide height) will vary; it is just that many constituents will have their variability poorly sampled.

**Commented [L11]:** This doesn't follow from the previous sentence.

And why mention TerraSAR-X here? You haven't yet explained that/whether this is the SAR dataset you intend to use.

generally simple and gravitational forces are well known, tide model
predictions are of high quality in areas where ice is \change{freely-
floating}{freely floating} on the ocean. \remove{(error = $\pm0.9$~cm,
Stammer et al., 2014)}. In coastal areas,  tide models are prone
to inaccuracies due to errors in model bathymetry, grounding-line
location and insufficient knowledge of the ice water drag coefficient
\citep{padman2018ocean}. Another source of error arises from the
conversion of ice-shelf draft to ice-shelf thickness and subsequent
estimation of water-column thickness. This freeboard conversion assumes
that ice near the coastline is in local hydrostatic equilibrium, whereas
stresses from the grounded ice clearly prevent a freely-floating state. A
bias of the hydrostatic solution towards thicker ice
\citep{marsh2014grounding}, and therefore a thinning water-column
thickness, may negatively affect the tidal prediction. In summary, the
relatively coarse spatial resolution and underlying assumptions of
contemporary tide models introduce inaccuracies especially in feature-
rich coastal areas such as fjord-type outlet glaciers. Although average
tide model accuracy has improved markedly in coastal areas over one
decade, from about $\pm10$~cm \citep{padman2002new} to $\pm6.5$~cm
\citep{stammer2014accuracy}, they are still an order of magnitude larger
than the sub-centimetre accuracy of DInSAR \citep{rignot2011antarctic}.
\add{For this reason, we consider DInSAR as the absolute truth and use
these space-borne measurements to correct tide-model output.}

In this manuscript, we show how the spatial benefits and high accuracy of
DInSAR can be used to refine coarse resolution tide models to adequately
resolve ocean tides along the feature-rich Antarctic coastline. First we
introduce the necessary data set, describe the preprocessing and guide
through the work flow. Second we test the algorithm for the Southern
McMurdo Ice Shelf (SMIS), a small and almost stagnant ice shelf with a
simple grounding-zone geometry, and expand the study to the Darwin
Glacier, a relatively fast-flowing outlet glacier within a complex fjord-
like embayment. We validate our results with dedicated field measurements
taken within the Transantarctic Ice Deflection Experiment (TIDEx) in
2016. We then demonstrate how this exercise can also be applied to reveal
errors in interferometric phase unwrapping and answer fundamental
questions about the physical properties of ice in Antarctic glaciology.

\section{Methodology}
\label{Meths}

\subsection{Summary of SAR image processing}
To develop the method, we use 11-day repeat-pass TerraSAR-X data in
StripMap imaging mode. The satellite acquires X-band radar data
(wavelength 3.1 cm, frequency 9.6 GHz) with a ground resolution of
slightly below 3x3~m and images covering an area of 30x50 km. We
calculate vertical surface displacement due to ocean tides using the
Gamma software package \citep{werner2000gamma}. \add{InSAR and DInSAR
image combinations are generally chosen so that a later image is always
subtracted from an earlier image. For image triplets, the central SAR
image serves as a common reference during the co-registration.} We then
correct the resulting DInSAR interferograms for apparent vertical

Commented [L12]: "in turn" isn't correct here.

Commented [L13]: poorly written

Commented [L14]: 1) what are the different dimensions? one along-track and the other cross-track?

2) are these image sizes just the way the along-track swath data are packaged, or what you can get for a target or opportunity, or ???

displacement due to horizontal surface motion \citep{rack2017analysis}
using the method presented  by \cite{wild2018unraveling}.

\subsection{Tide models}
The predictions of five tide models are validated: the regional
barotropic models (1) Circum-Antarctic Tidal Solution (CATS2008)
developed by \cite{padman2008improving}, (2) Ross Sea Height-Based Tidal
Inverse Model (Ross\_Inv\_2002) developed by \cite{padman2003tides}, (3)
Ross Sea assimilation model (Ross\_VMADCP\_9cm), (4) the fully global
barotropic assimilation model (TPXO7.2) from Oregon State University
developed by \cite{egbert2002efficient}, and (5) the (t\_tide) prediction
of GPS data from freely-floating areas following the harmonic analysis of
\cite{pawlowicz2002classical} \add{which is based on} FROTRAN codes
developed by \cite{foreman1977manual}. The t\_tide software is a widely-
used toolbox for performing classical harmonic analysis of ocean tides.
It can be sued to analyse any time series record and outputs the
amplitude and phase of its dominant harmonics \add{with error estimates},
along with a tidal prediction that is free from non-tidal effects. The
isostatic deformation of the Earth's lithosphere underneath the moving
water masses is modelled using TPXO7.2 load tide model
\citep{egbert2002efficient}, which itself is based on 13 tidal
constituents and added to all tide model predictions except t\_tide. In
addition to the tidal motion underneath the floating ice, much of the
ice-surface variability can be attributed to the Inverse Barometric
Effect \cite[IBE,][]{padman2003tides}. A +1~hPa anomaly of atmospheric
pressure translates to \change{a}{an instantaneous} -1~cm
\change{drop}{change} on the ice-shelf surface. \add{~~It is noteworthy to
mention~~Note that we did not apply a running mean to the pressure records,
as the application of any window length worsened the fit to available GPS
data.} To correct for the IBE \add{outside the GPS period}, we make use
of atmospheric pressure records obtained by nearby automatic weather
stations on Ross Island (Scott Base AWS) and the Ross Ice Shelf (Marilyn
AWS). We validate these records with separate barometric measurements
taken within the TIDEx campaign and find very good agreement.

\add{In this paper, we use the terms 'traditional tide modelling' or
'tide model' to refer to the sum of ocean-tide, load-tide model outputs
and the IBE. Freely-floating areas of ice shelves and glaciers are
expected to experience the full oscillation of this tide model.
Traditional tide modelling, however, neglects ice mechanics in grounding
zones where tidal flexure significanly affects the surface elevation
signal in reality. Other signals that change sea-level height such as
mean dynamic topography and storm surges are also excluded from this type
of tide model.}

\subsection{In-situ data}
We set up a number of GPS receivers to measure ice-surface motion at
millimetre accuracy and high temporal resolution. \change{Here we
use}{Although we used} GPS data from the freely-floating parts \change{of
the ice surface and}{to} develop local tide models using t\_tide,
\add{all GPS data was only used for validation purposes and did not feed
into the algorithm}. \remove{GPS measurements from within the tidal
flexure regions are only used as validation data sets.} \remove{All} GPS
measurements were differentially corrected using static base stations

**Commented [L15]:** My preference: use "by" to give credit to the authors for ideas, "in" when it's just where you find a fact. "Table 1 in Wild et al., …"

**Commented [L16]:** This is what we call it now

**Commented [L17]:** The Pawlowicz paper uses "T_TIDE" for style formatting.

**Commented [L18]:** Not exactly; has to be evenly spaced and adequately resolved.

**Commented [L19]:** absolutely not true! Non-tidal signals can get misrepresented as tides, especially on shorter records or where the non-tidal signal has frequencies near tides.

**Commented [L20]:** The Padman et al. 2003 papers need to be assigned 'a' and 'b' suffixes.

**Commented [L21]:** not hyphenated in this form. Might be hyphenated in, e.g., "tide-modelling output"

**Commented [L22]:** Really? Cite for this accuracy?

**Commented [L23]:** "T_TIDE"? see earlier comment. Pawlowicz gets to decide.

\add{to increase their spatial accuracy}. We also
\change{install}{installed} an array of seven tiltmeters
\change{recording}{to record} surface flexure over 16 days across the
grounding zone to confine the physical properties of Antarctic ice.
\change{This is}{The tiltmeters were} complemented by a dense network of
point measurements of ice thickness using a the new autonomous phase-
sensitive radar echo sounder \cite[ApRES,][]{nicholls2015ground}.

\subsection{Combining DInSAR and tide models}
\change{A tide model must perfectly predict the}{To allow a correct
interpretation of DInSAR images covering grounding zones, it is desirable
that tide models replicate} DInSAR observations \change{in an area that
can be expected to experience the full tidal forcing.}{in freely-floating
areas.} We first adjust the tide-model output to match the highly-
accurate DInSAR measurements using a least sum of squares routine
\citep{wild2018unraveling}. \add{By doing so, we consider just the tide
model amplitude to contain errors. Possible tide-model phase errors are
then accounted for by adjusting the absolute amplitude and thus the rate
of tidal change during the times of SAR data acquisition.} Second we
build on earlier work by \cite{han2014tide} and to develop an empirical
displacement map showing tide-deflection ratio throughout the satellite
image ($\alpha$-map). By feeding the $\alpha$-map with the adjusted tide
model output, the \change{'point-forecast'}{'point forecast'} is then
spatially extended to predict the mean vertical displacement for every
pixel at the times of SAR data acquisition. We then perform the
\change{double-differences}{double differences} of the empirical model
corresponding to the SAR image combinations used to generate the DInSAR
images. The original DInSAR satellite measurements are subsequently
removed from the mean DInSAR images to calculate their misfits, $\mu$. We
now compute the least-squares solution to the equation $Ax=b$ such that
the 2-norm $|b-Ax|$ is minimized. Here, $A$ is the $m\times n$ DInSAR
matrix of SAR image combinations with $m$ rows of SAR images and $n$
columns of coherent DInSAR interferograms; $b$ is a vector of $\alpha$-
prediction misfits and $x$ the least-squares solution of this
\change{over-determined system}{system of linear equations}. The values
of $x$ correspond to how much an $\alpha$-prediction deviates from the
'real' vertical displacement at the times of SAR data acquisition. We
therefore subtract these offsets, \change{$\theta$}{$\Delta A$}, from the
$\alpha$-prediction maps.

We \add{now} demonstrate the workflow in one spatial dimension with an
example of the Southern McMurdo Ice Shelf ($78^{\circ}15\textrm{' S}$,
$167^{\circ}7\textrm{' E}$, SMIS). In this study area, we derived 9
DInSAR images from 12 TerraSAR-X scenes in 2014 \citep{rack2017analysis}.
\add{The low number of DInSAR images is a consequence of the SAR scenes
being acquired on 3 different satellite tracks. The resulting system of
linear equations is therefore underdetermined as there are more offsets,
$\Delta A$, than misfits, $\mu$, to constrain the least-squares
solutions.} We choose a pixel on the freely-floating end of a profile
through the ice-shelf grounding zone to represent the unrestricted ice
shelf movement and calculate the percentage vertical displacement of
every other pixel from this location. Averaged over the 9 DInSAR
interferograms this pixel retains 100\% vertical displacement (red areas
in Fig. \ref{fig:alpha_maps}), while grounded areas experience zero net

Commented [L24]: Wrong word. Maybe "constrain", but really "estimate"

Commented [L25]: Not "new" any more.

Commented [L26]: We've discussed this before. This method doesn't really "account for" phase errors. Take an extreme case: K1 in a model is 180 degrees out of phase, even though the amplitude is right. How will amplitude scaling help?

Note that, because phase errors are smallish, the amplitude scaling is probably an okay first step. But I still don't see why this fitting can't be done in complex space to get amp and phase sorted at the same time.

Commented [L27]: you haven't defined '{\Delta}A'

Commented [L28]: I have to trust you on this; I really don't follow.

Commented [L29]: I don't understand this either. Surely the primary control on "low number" is SAR mission duration and repeat interval. Here, I don't even understand if the number of images would go up or down if you had more or fewer tracks.

Commented [L30]: Flexure in the GZ can include overshoot (alpha >100%); see, e.g., Fricker and Padman 2006

uplift (purple areas). Individual pixels on the freely-floating part of the SMIS may show $\alpha$-values slightly above 100\%.

We now extract the $\alpha$-values along the profile from the $\alpha$-map. This $\alpha$-profile can be multiplied with the individual DInSAR measurements on its freely-floating end which results in empirically derived $\alpha$-predictions (Fig. \ref{fig:ls_process_DInSAR_SMIS} center). These mean predictions do not perfectly replicate the DInSAR measurements (Fig. \ref{fig:ls_process_DInSAR_SMIS} top). Their misfits, however, show a very systematic pattern (Fig. \ref{fig:ls_process_DInSAR_SMIS} bottom). It is desirable to find a combination of offsets that have the least deviation from the $\alpha$-predictions. We therefore hypothesize that this rather systematic signal can be reconstructed using a least-squares strategy. \remove{Here, the linear system is under-determined with 9 DInSAR equations and 12 unknown SAR offsets.} We solve the \add{underdetermined} system simultaneously by finding the combination of offsets that result in a minimal sum of squares. The reconstructed offsets must then be removed from the $\alpha$-prediction for the times of SAR data acquisition (Fig. \ref{fig:ls_process_SAR_SMIS} top). The computed least-square offsets generally replicate the pattern of the misfits (Fig. \ref{fig:ls_process_SAR_SMIS} center) and result in smooth displacement profiles in the ice-shelf grounding zone (Fig. \ref{fig:ls_process_SAR_SMIS}).

\section{Results}
\label{Res}
In this section we apply the workflow in two spatial dimensions to the Darwin Glacier ($79^{\circ}53\textrm{' S}, 159^{\circ}00\textrm{' E}$). \add{In this study area, we derived a total of 45 DInSAR images from 12 SAR scenes being acquired on the same satellite track. SAR image combinations were generally chosen consecutively so that a later image is always subtracted from an earlier image. For image triples, the central image was taken as a common reference/master image. Additionally the data gap between SAR 8 and 9 was taken into account (no 8-9 combination as loss of coherence over this relatively long interval). The advantage of using every other remaining combination} (Tab. \ref{tab:DInSAR_table}) \add{is that more double-differential measurements of tidal amplitude are available for the least-squares fitting algorithm than only using consecutive pairs alone. The system of linear equations is then overdetermined.} A dedicated field campaign was conducted in \change{its}{the Darwin Glacier} grounding zone in 2016 and \textit{in-situ} data is available \add{for numerical modelling and field validation purposes}. In contrast to the simple geometry at the SMIS, the Darwin Glacier consists of a feature-rich embayment that is constrained by steep topography at its margins. Additionally a buttressing ice rise to the Ross Ice Shelf restricts outflow in the North.

\subsection{Reconstruction of displacement maps during satellite overpasses}
From the interferogram dataset we identify a corridor of only about 2~km width along the centerline where the glacier can be assumed to be freely floating (Fig. \ref{fig:alpha_maps}). This area is expected to experience the full oscillation predicted from tide models. We run five tide models

Commented [L31]: callouts to specific panels are good policy; however, it's much easier with 'a','b' etc panel IDs.

Commented [L32]: This is the first time you've really discussed this gap. The thing a reader wants to know is not so much about coherence loss, but why the gap is there in the first place.

Commented [L33]: Figure 1 should have "North" arrows. It isn't obvious whether North is up or down.

to predict the tidal oscillation at the GPS station 'Shirase' over the \change{time-span}{time span} of SAR data acquisitions. Here we use atmospheric pressure data from the automatic weather station 'Marilyn' which is located about 120~km away on the Ross Ice Shelf to correct for the inverse barometric effect. This record correlates well (Pearson's correlator 0.989) with a mean of seven barometers installed over 14 days across the Darwin Glacier during the TIDEx campaign. All tide-model predictions show a clear fortnightly occuring spring-neap tidal cycle which is superimposed by a dominant diurnal signal (Fig. \ref{fig:forcing_Darwin}). The approximately fortnightly  spring/neap tides  the difference in  frequency between the dominant K1 and O1 diurnal constituents. \remove{The K1 tide has a period of 23.93~hours and is the dominant tidal constituent in the Ross Sea (Padman et al., 2018).}

We apply t\_tide to our 16~day record of the 'Shirase' GPS to test the potential of short-term GPS surveys to improve current Antarctic \change{tide-models}{tide models}. \add{The problem with using such a short window to determine a full set of tidal constituents results from the interplay of the lunar diurnal tide (K1, 23.93~h) with the solar diurnal tide (P1, 24.07~h) as they are close in frequency and P1 has an amplitude of $15-20$~\% of K1. Without accounting for their inference, t\_tide just extracts an apparent K1 from a 16~day record that is really K1+P1. As a consequence, the K1 tide from our harmonic analysis can vary by $30-40$~\% over a 6-month period and its amplitude is only controlled by the exact time that the GPS data was acquired within the K1+P1 modulation cycle. Additionally, harmonic decomposition of GPS data is subject to inaccuracies itself with errors in both the extracted amplitudes and phases. These errors were found to be of the same magnitude as the K1+P1 inference. For this reason, we did not use inference to separate K1 and P1 (or similarly to separate the semi-diurnal S2 and K2 constituents), but perform a thorough analysis on the identified uncertainty range.} While the analysis captures the dominant \change{diurnal}{K1} constituent \add{in the Ross Sea within a reasonable signal-to-noise ratio}, fortnightly harmonics could not be retrieved adequately from this time series alone. The t\_tide prediction is therefore the least accurate tide model and requires the largest adjustment to match DInSAR (Tab. \ref{tab:SAR_table}). Although all the corrected tide model outputs now replicate our DInSAR measurements, their rate of tidal change is affected by the adjustment. Offsets computed for the Ross\_Inv\_2002 tide model are generally below 10~cm, whereas other tide models require adjustments of up to 13.3~cm (Tab. \ref{tab:SAR_table}). This agrees well with the findings of \cite{han2013accuracy}, who find that the Ross\_Inv\_2002 model is the optimum tide model for the Terra Nova Bay with a 4.1~cm RMSE against 11 days of tide gauge data. We therefore choose Ross\_Inv\_2002 for numerical modelling purposes to minimize any effects on a viscoelastic model, but use TPXO7.2 to reconstruct vertical displacement at the times of satellite overpasses as it fits best \add{to} our GPS measurements. We refer to the Appendix for a validation of individual tide model output with GPS data from 'Shirase' (Fig. \ref{fig:GPS_validation}).

After the adjustment, modelled tidal amplitudes range from -0.966~m to 0.781~m over the whole SAR period (Fig. \ref{fig:forcing_Darwin}). Mean

\add{absolute} residual error to the 45 DInSAR measurements at the tide-model location 'Shirase' is just \change{0.84~mm}{7~mm} (Tab. \ref{tab:DInSAR_table}), \change{which is within}{which can be explained by} interferogram noise. We attribute this accuracy to the exceptionally high phase coherence of the TerraSAR-X data set. The reconstruction algorithm results in 12 smooth vertical displacement maps for the times of SAR data acquisition (Fig. \ref{fig:coeff_maps_Darwin}).

\subsection{Validation with GPS measurements}
We now validate these reconstructions with available field data. As both GPS records overlap with the acquisition of SAR image 11, we extract the vertical displacement along the glacier's centerline and plot the profiles against the two GPS point measurements. The GPS measurement at 'Hillary' is 0.169~m, which is close to the reconstruction of 0.156~m. The 'Shirase' GPS measurement is 0.566~m, which is slightly above the reconstruction of 0.522~m. We attribute the deviations of +1.3 and +4.4~cm, respectively, to a combination of interpolation artefacts, temporal smoothing of the GPS data and residual errors of the least-squares algorithm. The overall shape of the vertical displacement is well reproduced as observed with both GPS measurements (Fig. \ref{fig:LS_flexure_curves_Darwin}).

\subsection{Applications}
\subsubsection{Tide-model refinement}
A map of tide-deflection ratio ($\alpha$-map) can be combined with the tide model to predict an average \change{time-series}{time series} of vertical displacement between the times of SAR image acquisition. With this approach, the coarse grid of \add{traditional} tide models is refined to resolve small-scale features of vertical tidal displacement throughout the embayment. The $\alpha$-value for the pixel containing the 'Hillary' GPS station is 46.06\%. We use this value and linearly scale the \add{adjusted} tide-model output for the location of the 'Shirase' GPS to predict vertical tidal motion within the flexure zone. This scaling maintains the tide model's high correlation (Pearson's correlator 0.95) with the 'Shirase' record, but improves the RMSE between the TPXO7.2 output and the 'Hillary' record from 21.4~cm to 5.6~cm, which corresponds to an improvement of -74\% \add{to GPS data} (Fig. \ref{fig:Hillary_validation}). \add{The primary reason for this large improvement, however, is that the tide model now takes the damping of the tidal signal by ice mechanics in the grounding zone into account.}

\subsubsection{Ice \change{heterogenity}{heterogeneity}}
With the \change{twelve}{12} reconstructed displacement maps at hand, it is now possible to perform any image combination. We mosaic the 45 double differences corresponding to DInSAR combinations (Tab. \ref{tab:DInSAR_table}) \add{to allow a more direct comparison between measured and modelled interferograms}. \add{SAR image combinations were chosen so that the loss of coherence between SAR 8 and 9 was taken into account and that a maximum number of consecutive, double-differential interferograms was available for the least-squares fitting routine.} \add{The synthetic interferograms replicate not only simple tidal fringes as measured with DInSAR, but also show complicated viscoelastic signals within the grounding zone} (Fig. \ref{fig:Darwin_DInSARs}). \add{For an overall assessment of model performance we} \remove{and then} calculate

again the misfit between each modelled and observed interferogram for every pixel, but this time after using the adjusted tide-model output. The standard deviation of these misfits is shown in Fig. \ref{fig:LS_STD_maps}, with the majority of the glacier surface below noise level of interferograms ($\sigma < 1.0$~cm). We identify a narrow band with higher standard deviations ($\sigma \approx 2.0$~cm) from the inner shear margin of the Darwin Glacier extending along \change{flow-direction}{flow direction} onto its ice shelf. Standard deviations are largest out \change{on the}{in the shear zone of the fast-flowing} Ross Ice Shelf \add{and above steep rocky cliffs} ($\sigma > 4.0$~cm) \change{and a}{which is a} result of poor \change{coherence between}{phase coherence or layovers in} SAR images in \change{this area}{these areas} .

\subsubsection{Detection of errors in phase unwrapping}
We now \change{apply the algorithm to the}{extend the earlier one dimensional analysis of the SMIS to a two dimensional re-analysis of the} SMIS data set and calculate misfits of 9 DInSAR interferograms. Resulting standard deviations are generally smaller in this area ($\sigma < 0.3$~cm) and smoothly distributed throughout the map. We identify two regions of \change{jumps}{phase discontinuities} between adjacent cells at the SMIS. Both extend from the center of an ice rise towards \change{the dry land}{Black Island} (cyan and green areas in Fig. \ref{fig:LS_STD_maps}) with $\sigma \approx 0.4$~cm and $\sigma \approx 0.5$~cm, respectively. We interpret these rapid increases as a proxy for errors in the DInSAR measurements, as the modelled least-square interferograms originate from a curvature-minimizing polynomial interpolation. We re-evaluate the remote-sensing part of the analysis and find discontinuities in DInSAR interferograms ID:1 and ID:8 that match the course of the two \add{phase} jumps in the standard deviation map. These discontinuities, in turn, \change{are a result of using a minimum cost-flow algorithm on a triangular network for unwrapping}{occurred during phase unwrapping} \remove{interferometric phase differences to relative surface displacement} \add{and can now be corrected}.

\subsection{Finite-element modelling of viscoelasticity}
We hypothesize that any non-linear signal due to viscoelastic ice properties is significantly reduced or even completely lost during the averaging step to compute the $\alpha$-map. This signal can then be reconstructed by finding the offsets to match observations made with DInSAR. We therefore subtract the $\alpha$-prediction again from the 12 reconstructions to extract the theorised viscoelastic signal (Fig. \ref{fig:visco_maps_Darwin}). This signal is negligible at times during neap tide (SAR 4 and 9) but well pronounced for SAR images acquired during spring-tide periods (SAR 1,6,7,10 and 11).

In order to further explore this pattern, we now make use of the tiltmeter array and ApRES network of ice-thickness measurements at the Darwin Glacier (Fig. \ref{fig:Darwin_field_overview}). We match the numerical solutions from two finite-element models to seven tiltmeter records, with the goal to derive information on the physical properties of Antarctic ice. Thereby, the Young's modulus, $E$, is a measure of ice stiffness and controls the width of the flexure region. The value for ice viscosity, $\nu$, influences the timing of the flexural response within

**Commented [L36]:** not clear what counts as the "inner" shear margin.

**Commented [L37]:** I don't know the term "layovers" in this context.

**Commented [L38]:** I don't know what "increases" refers to. Sounds like time-domain, but I guess you mean "spatial gradients to the locations of maxima" or something like that.

**Commented [L39]:** So, we don't really have much information about this "curvature-minimizing polynomial interpolation". Can I repeat your study by following your "explanation" ?

**Commented [L40]:** what's the "course of the two phase jumps" ?

**Commented [L41]:** Note that my understanding of the ice rheology stuff is limited.

**Commented [L42]:** Have we seen Fig. 11 yet? Looks like you went straight from Fig. 10 to Fig. 12

the grounding zone \citep{wild2017viscosity}. Two numerical models of ice-shelf flexure are employed. The elastic approximation \citep{holdsworth1969flexure, vaughan1995tidal,schmeltz2002tidal} as formulated by \cite{walker2013ice}:
\begin{equation}\label{elastic_model}
 kw + \nabla^{2} (D \nabla^{2} w) = q,
\end{equation}
where $w(t)$ is the \change{time-dependant}{time-dependent} vertical deflection of the neutral layer in a plate, $\nabla^{2}$ is the Laplace operator in 2-D space and $k=5$~MPa~m$^{-1}$ a spring constant of the foundation which is zero for the floating part. The applied tidal force $q(t)$ is defined by:
\begin{equation}
 q = \rho_{sw} g [A(t) - w],
\end{equation}
with $g=9.81$~m~s$^{-2}$ the gravitational acceleration and $A(t)$ the \change{time-dependant}{time-dependent} tidal amplitude given by the adjusted Ross\_Inv\_2002 tide model. We choose this model, in contrast to the TPXO7.2 model, for finite-element simulations to minimize any potential effects of tide-model adjustment on viscoelasticity (Tab. \ref{tab:SAR_table}). The stiffness of the ice shelf is given by \cite[p. 443]{love1906treatise}:
\begin{equation}\label{03}
 D = \frac{E H^{3}}{12(1-\lambda^{2})},
\end{equation}
where $E$ is the Young's modulus for ice, $H(x,y)$ our ice thickness map derived from ApRES point measurements and $\lambda=0.4$ the Poisson's ratio. We compare the elastic model with the viscoelastic approach developed by \cite{walker2013ice}:
\begin{equation}\label{visco_model}
  \frac{\partial}{\partial t} \left[ kw + \nabla^{2} \left( D \nabla^{2} w \right) \right] + \frac{Ek}{2 \nu(1-\lambda^{2})} w
   = \frac{\partial}{\partial t} q + \frac{E}{2 \nu (1-\lambda^{2})} q,
\end{equation}
where $\nu$ is ice viscosity. The following boundary conditions are applied for both models: the upstream boundary of the model domains on the grounded portion are anchored rigidly ($w=0, \nabla^{2} w=0$), the downstream boundaries on the freely-floating ice shelf are set free. The location of the tide model computation is constrained to be equal to the tidal oscillation ($w=A(t), \nabla w = 0$) and the grounding line is represented by a fulcrum ($w=0$). Both models are implemented in two spatial dimensions to capture effects of complex grounding-line configuration on ice-shelf flexure \citep{wild2018unraveling}. We then solve the models using the commercial finite-element software COMSOL Multiphysics. As tiltmeters measure slope, $w'$,  along their longitudinal axis, we derive the models' solutions for vertical displacement, $w$, with respect to the $x$ and $y$ directions. This allows us to retrieve surface slopes components in easting and northing direction and to rotate them into the individual orientations of the tiltmeter sensors. With our 16~day tiltmeter records it is only possible to capture their diurnal \remove{and semi-diurnal} components with confidence. \change{Fortnightly}{Semi-diurnal, fortnightly} and monthly harmonics have been removed from the tiltmeter time series and we focus further analysis only on the K1 component \add{within the 16~day window}.

Therefore, we now make extensive use of the t\_tide program to automatically extract the modelled K1 harmonics from the modelled surface slopes and compare them against the K1 constituents from the tiltmeters. \add{We thereby take amplitude and phase errors that originate from the harmonic analysis of noisy tiltmeter records into account and find the best rheological parameters to match the elastic and viscoelastic models to these seven K1 components. Incorporating viscoelastic effects into the model simulations always improves the elastic fit to the tiltmeter data within the uncertainty range of K1 amplitude and phase} (Tab. \ref{tab:K1_table}). \change{A Young's modulus of $E=1.0$~GPa and an ice viscosity value of $\nu=10$~TPa~s fits best our measurements}{We find that an average Young's modulus of $E=1.0\pm0.56$~GPa and an ice viscosity value of $\nu=10\pm3.65$~TPa~s fits best to our measurements within uncertainty} (Fig. \ref{fig:K1_components}). The viscoelastic model gives an average RMSE of 0.00118845~$^{\circ}$ to the seven tiltmeters and improves on the elastic approximation with an average RMSE of 0.00147136~$^{\circ}$ by $\approx -20$\%.

\section{Discussion}
\label{Dis}
\subsection{Seasonal bias in $\alpha$-map}
Due to the alignment of the satellite overpasses with the dominant diurnal tidal constituents in the Ross Sea, the observed stage of the tidal oscillation varies only slowly throughout the year. In the austral winter months, \change{SAR}{TerraSAR-X} images \change{are}{have been} acquired during stages of low tide, whereas satellite overpasses concur with stages of high tide during the austral summer months. The first 8 snapshots of our \change{TerraSAR-X data set}{SAR data acquisitions} for the Darwin Glacier show conditions at low tide and only the last 4 are acquired during high tide. Our $\alpha$-map, in turn, ignores this seasonality and may therefore have a low-tide bias. As a result, the contribution of a tide induced landward migration of the grounding line may be affected by the averaging process. The seasonal bias would then modify the scaling of the \change{tide-model}{tide model} within the flexure zone. This \remove{theory} is supported by the finding that low-tide stages in the 'Hillary' GPS record are matched closely by the scaled tide model, but peaks during high-tide stages are still over-\-estimated (Fig. \ref{fig:forcing_Darwin})

\subsection{Viscoelasticity between snapshots}
Similarly, the linear scaling using an $\alpha$-map only modifies the predicted tidal amplitude, but neglects a viscoelastic time delay in the flexural response towards the grounding line. \cite{wild2017viscosity} found that viscosity is most pronounced in the diurnal tidal components. Harmonic analysis of our GPS records reveals that the diurnal K1 and O1 constituents at 'Hillary' are lagging approximately 20 mins behind 'Shirase'. This signal is currently disregarded in the scaling work flow as ice is treated as a perfect elastic material that transfers tidal \change{motion}{forcing} instantaneously in the flexure zone. This assumption, however, allowed us to improve the accuracy of the \change{tide-model prediction}{predicted displacement} by -74\%. Currently, the viscoelastic signal can only be reconstructed for the times of SAR data acquisition. Including viscoelasticity between times of satellite overpasses \change{may therefore be only}{offers} a small, but

systematic, opportunity for further refinement. \add{In our study area, the rate of tidal change is up to $10$~cm~hr$^{-1}$} (Tab. \ref{tab:SAR_table}) \add{and the viscoelastic misfit corresponding to 20~minutes time delay is therefore up to about 3~cm.}

When separating the viscoelastic contribution from the reconstructed maps of vertical displacement at times of satellite overpasses, we assume that an $\alpha$-prediction corresponds to an instantaneous elastic response. This is justified by viscoelasticity being most pronounced when rates of tidal change are maximal. \add{By expressing the viscoelastic misfits in percent of prevailing tidal amplitude during the times of satellite overpasses, the areas of pronounced viscoelastic effects can be visualised. They are most pronounced within the Darwin Glacier's shear zone} (Fig. \ref{fig:visco_errors_Darwin}). \remove{SAR images acquired during periods of spring tides at the Darwin Glacier show also a significant viscoelastic contribution that diminishes during neap tide periods.}

When predicting rates of tidal change using the adjusted Ross\_Inv\_2002 tide model, we identify a threshold of $\dot{A}\approx \pm0.05$~m~h$^{-1}$ (SAR times 1, 2, 7, 8, 10, 11 in Tab. \ref{tab:SAR_table}) above which viscoelasticity is well represented in the reconstructed vertical displacement maps (panels 1, 6, 7, 8, 10 and 11 in Fig. \ref{fig:visco_maps_Darwin}). Image 6 is thereby an exception, as the used Ross\_Inv\_2002 tide model was \change{adjusted largely}{largely adjusted} (-0.082 m), which affects the viscoelastic model. \add{We find that the error due to viscoelasticity on the floating part of the ice shelf increases with the absolute rate of tidal change} (Fig. \ref{fig:visco_errors_Darwin}). \add{SAR images acquired during periods of spring tides at the Darwin Glacier show \add{also} a significant viscoelastic contribution that diminishes during neap tide periods.} These independent observations \add{from satellite data alone} support our suggested threshold of $\pm0.05$~m~h$^{-1}$ for the separation of elastic and viscoelastic signals, as derived from tiltmeter data on the Southern McMurdo Ice Shelf presented in an earlier study \citep[Fig. 8 in][]{wild2017viscosity}. The advantage of separating the elastic from the viscoelastic contribution to the tidal flexure pattern is the large potential for improving current inverse modelling techniques to determine grounding-zone ice thickness from DInSAR measurements alone. Hereby, an elastic model is currently employed to optimize grounding-zone ice thickness to match the surface flexure from DInSAR. \remove{This is because an elastic model for tidal flexure is only forced by the 'apparent' tidal amplitude (Eq.), which can be measured directly from the interferogram on the freely-floating area. A viscoelastic model additionally incorporates the time derivative of the tidal forcing (Eq.) and hence captures the rate of tidal change. This information, in turn, can not be deduced directly from the interferogram which makes the usage of auxiliary tide models inevitable. Tide models, however, have shown to be prone to large inaccuracies around Antarctica making a successful inversion of viscoelastic flexure models highly elusive.} The applicability of an elastic model varies from location to location as effective viscosity is dependent on ice temperature and shear stress \citep{marsh2014grounding}. Our method to separate the two contributions to the flexure pattern may therefore help to remove the viscoelastic

Commented [L46]: I think Figs. 11 and 12 are assigned and cited incorrectly.

Commented [L47]: I don't understand what you mean here. Maybe that the adjustment was large, rather than the model being "largely adjusted" ?

contamination and allow purely elastic inverse modelling.
\remove{Furthermore, the threshold of $\pm0.05$~m~h$^{-1}$ is invaluable
to determine which satellite data acquisitions should be used for this
calculation.} \change{This}{Such an} analysis, however, goes beyond the
scope of this manuscript and will be published elsewhere.

\subsection{Large-scale ice anisotropy}
Fast-moving glacial environments like the Darwin Glacier are subject to
large deformation by flow convergence and divergence, ice compression and
extension, lateral shearing at the margins accompanied by fracture under
tension and rapid thinning \change{at the ice-ocean interface}{by basal
melt}. With cumulative deformation, a crystallographic fabric evolves
that reflects the glacier's flow history \citep{alley1988fabrics}, and
with it strain-dependent mechanical anisotropy of ice. The standard
deviation map, Fig. \ref{fig:LS_STD_maps}, shows a narrow band of larger
misfits extending from the Darwin Glacier's inner shear margin out
towards the freely-floating ice shelf. As preferred crystallographic
orientation develops with strain, effective viscosity decreases of about
a factor of ten compared to initially isotropic polycrystalline ice
\citep{hudleston2015structures}. Our analysis of tiltmeter data reveals a
five-fold reduced viscosity at the very dynamic Darwin Glacier compared
to an earlier study at the almost stagnant Southern McMurdo Ice Shelf
\citep{wild2017viscosity}. We \change{theorise}{hypothesize} that this
microscopic process explains the macroscopic response observed here, and
accounts for the measured glacial \change{heterogenity}{heterogeneity}
within the embayment. Large scale observations of ice anisotropy, in
turn, are currently the missing key to improve parametrisations to
account for polar ice anisotropy in ice-sheet flow modelling
\citep{gagliardini2009review}. \add{Other processes have been proposed
which lead to ice softening in areas with high strain rates.
Thermomechanical modelling suggests that shear heating and consequent
thermal softening reduces lateral drag in ice-stream margins}
\citep{perol2015shear}. \add{Fracture modelling implies that damage
reduces ice viscosity along confined crevassed zones with consequences on
ice-shelf scale} \citep{albrecht2014fracture}. \add{Full-Stokes
viscoelastic modelling shows that Glen's non-linear flow law and tidal
stresses in the ice-shelf flexure zone are sufficient to explain large-
scale temporal variations in ice dynamics} \citep{rosier2018tidal}.
\add{These processes, or a combination of them, might certainly be at
play but they do not explain why a band of higher standard deviations can
be observed in the shear zone of the Darwin Glacier which is absent in
the flexure zone of the SMIS} (Fig. \ref{fig:LS_STD_maps}). \add{We
therefore attribute this difference to ice-fabric reorientation in the
shear margin.}

\subsection{Refining tidal constituents using DInSAR}
\add{The idea of using SAR interferometry to derive a full set of tidal
harmonics was first laid out in a study of tides in the Weddell Sea}
\citep{rignot2000observation}. \add{The authors discussed that DInSAR
images cannot be transformed into individual displacement fields because
of the nonuniqueness of the inversion. A large number of independent
DInSAR images is required to overcome this problem and to resolve the
phase of tidal constituents that are close to the repeat-pass of the SAR
satellite. For example, multiples of the lunar diurnal constituent K1

**Commented [L48]:** Again, not really clear what "inner" means Darwin Glacier.

**Commented [L49]:** Maybe a paragraph break here

(23.93~h) are relatively close to the exact integer repeat-pass of TerraSAR-X (11~days), meaning that the observed amplitude of the K1 constituent is only varying once throughout the year. Consequently, SAR images need to be acquired at least over the duration of one year to provide some redundancy for the inversion step of DInSAR images to tidal constituents. However, with an exact 12~h period, the stage of the semidiurnal solar tide, S2, will always be the same at each satellite pass, making TerraSAR-X and similar satellites with repeat -passes of integer days blind to the S2 constituent.} \add{For example,} \cite{minchew2017tidally} \add{needed a unique spatially and temporally dense SAR acquisition campaign as well as \textit{a-priori} knowledge of the temporal basis functions from GPS data to empirically determine tidal constituents on Rutford Ice Stream. The four COSMO-SkyMed satellites in orbit, however, produce repeat-passes of 1, 3, 4 and 8 days and are blind to the S2 constituent as well even when using $>1000$ available DInSAR images. Although other dominant tidal constituents like M2 (12.4~h) and O1 (25.82~h) were inferred successfully, the method presented here can achieve a higher accuracy offo the total tide with fewerless DInSAR images. From another perspective, the inclusion of an auxiliary tide model eases the requirement of a very large number of DInSAR images.}

\conclusions[Conclusions and Outlook]  %% \conclusions[modified heading if necessary]
\label{Conc}
\add{Accurate prediction of ocean tides in coastal areas around Antarctica is crucial as the majority of Antarctica's ice is discharged through large outlet glaciers.} \change{Here we present the first data fusion of DInSAR with traditional Antarctic tide-modelling to predict spatial variability of tidal motion. The principal value of using DInSAR and tide models in tandem lies in the spatio-temporal benefits of resolving small features over large regions.}{We presented a data fusion method between DInSAR and traditional Antarctic tide models to predict spatial variability of tidal motion near the grounding line. The primary value of using DInSAR in conjunction with tide models lies in the spatio-temporal benefits of resolving complex grounding zone deformation.} Their symbiosis not only improves current accuracies of the predicted tidal amplitudes in coastal regions generally, but also avoids issues related to the timing of the tidal wave and the sun-synchronous satellite orbit when attempting to derive tide-models from SAR data alone. \add{In our study area,} the method presented in this paper improves traditional tide modelling in average by -22\% from 11.8~cm to 9.3~cm RMSE against 16 days of GPS data. The GPS station 'Shirase' on the freely-floating part of the Darwin Glacier has proven invaluable to determine which \change{tide-model}{tide model} has to be used to best reconstruct the vertical displacement during satellite overpasses. For the Darwin Glacier, the TPXO7.2 tide model predicts best the tidal oscillation. With using DInSAR measurements to adjust the TPXO7.2 tidal prediction, its RMSE could be improved by -39\% from 10.8~cm to 6.7~cm. \remove{which exceeds the average tide model improvement of -35\% within the last decade (Stammer et al., 2014).}

Our GPS record from 'Shirase' is too short to \change{develop a local tide model that improves}{improve} already available Antarctic tide models. A longer record is required to adequately resolve a full set of

**Commented [L50]:** A casual reader might get confused here: how is COSMO-SkyMed related to TerraSAR-X (Not at all, right?)  SO, first time Minchew comes up, explain this, and why you can't use the same satellite for your SMIS and Darwin Glacier studies but have to use TerraSAR-X instead.

**Commented [L51]:** This is wrong, and probably easiest to delete it.

**Commented [L52]:** I don't think this is right. It does *not* avoid issues with timing (phase); it simply works to minimize problems that arise from phase errors. The earlier comment about the "thought experiment" about a major harmonic with an extreme phase error seems relevant here.

tidal constituents. \remove{The GPS record from 'Hillary' could not be used for this purpose as it was recorded within the tidal flexure zone.} \add{We produced an empirical displacement map from DInSAR for tidal deflection ($\alpha$-map).} Comparison of \change{its measurements}{a GPS record within the tidal flexure zone ('Hillary')} with predicted vertical displacement from feeding the $\alpha$-map with the adjusted TPXO7.2 tide model shows a -74\% improvement over using the tide-model output alone. This independent validation supports the finding that \add{our method for making use of} DInSAR is very useful for refining tide models in Antarctic \change{grounding-zones}{grounding zones}. \remove{Accurate prediction of ocean tides in coastal areas is crucial as the majority of Antarctica's ice is discharged through large outlet glaciers.}

Numerical modelling of ice dynamics in Antarctic grounding zones commonly assumes that ice is isotropic and homogeneous i.e. of \add{the} same density and rheological properties throughout. Our analysis reveals that this assumption is valid for the Southern McMurdo Ice Shelf, an almost stagnant area with a simple grounding line \add{configuration}, but invalid for the Darwin Glacier, a fast-flowing outlet glacier with complex shear margins causing non-negligible ice \change{heterogenity}{heterogeneity} within the embayment.

Further work is required \change{(1) to incorporate viscoelasticity to continue refining predictions of tidal motion between times of satellite overpasses, (2) to develop an automated method to monitor grounding-line migration due to ocean tides and (3) to perform inverse modelling of tidal elastic flexure to indirectly measure ice thickness from SAR data.}{in order to improve tide models in a larger variety of grounding zones by including effects of grounding-line migration and variability of horizontal ice flow.}

%% The following commands are for the statements about the availability of data sets and/or software code corresponding to the manuscript.
%% It is strongly recommended to make use of these sections in case data sets and/or software code have been part of your research the article is based on.

\codeavailability{The code is freely available to the scientific community. Collaboration is anticipated and desired.
} %% use this section when having only software code available

\dataavailability{TerraSAR-X data presented in this paper are subject to license agreements. GPS/tiltmeter and ApRES data are available upon request.} %% use this section when having only data sets available

\codedataavailability{No data sets, nor software, are part of this study.} %% use this section when having data sets and software code available

\sampleavailability{No samples were collected for this study.} %% use this section when having geoscientific samples available

**Commented [L53]:** a negative improvement is not an improvement!

```
\appendix
\section{GPS evaluation of tide models}     %% Appendix A
The quality of the used \change{tide-model}{tide model} to correctly
reconstruct tidal displacement at the times of SAR data acquisitions, is
also crucial to accurately predict spatial variability in tidal motion
for all times. Here, we assume that a freely-floating area on the
\change{ice-shelf}{ice shelf} experiences the full oscillation as
predicted from a tide model. In this area, however, tide-model output
deviates from our DInSAR measurements. This indicates either that the
area under investigation is prevented from a freely-floating state by
lateral stresses within the embayment, or that the tide-model prediction
is inaccurate for this area. We circumvent this ambiguity by making use
of the high vertical accuracy of DInSAR and correct the tide-model
prediction to match our satellite measurements. This raises the question
of whether the adjustment improves or worsens the match to a 'real' tidal
motion ? We therefore independently evaluate the pre- and post adjustment
tide-model predictions and calculate their RMSE to 16 days of GPS data
from the freely-floating area (Tab. \ref{tab:GPS_table}). The adjustment
improves all traditional tide model predictions by up to -39\% for
TPXO7.2, and only worsens the RMSE for the t\_tide output by +11\%,
indicating that a harmonic analysis of GPS data can not be improved by
using DInSAR for correction purposes. We choose TPXO7.2 for further
processing as it displays the overall smallest RMSE (6.7~cm) and
replicates the small-scale variability observed during the neap-tide
period in the second half of our GPS record.
%\subsection{}      %% Appendix A1, A2, etc.

\noappendix        %% use this to mark the end of the appendix section

%% Regarding figures and tables in appendices, the following two options
are possible depending on your general handling of figures and tables in
the manuscript environment:

%% Option 1: If you sorted all figures and tables into the sections of
the text, please also sort the appendix figures and appendix tables into
the respective appendix sections.
%% They will be correctly named automatically.

%% Option 2: If you put all figures after the reference list, please
insert appendix tables and figures after the normal tables and figures.
%% To rename them correctly to A1, A2, etc., please add the following
commands in front of them:

%% Please add \clearpage between each table and/or figure. Further
guidelines on figures and tables can be found below.

\authorcontribution{All authors conceived the study and conducted
fieldwork. CW developed the algorithm, performed the data analysis and
```

wrote a first version of the paper. OM and WR processed the SAR images for the Darwin Glacier and Southern McMurdo Ice Shelf, respectively. All authors finalized and approved the manuscript.
} %% it is strongly recommended to make use of this section

\competinginterests{The authors declare that the research was conducted in the absence of any commercial or financial relationships that could be construed as a potential conflict of interest.
} %% this section is mandatory even if you declare that no competing interests are present

\disclaimer{} %% optional section

\begin{acknowledgements}
We thank Antarctica New Zealand for logistical support and the Scott Base staff for their dedication to the Transantarctic Ice Deflection Experiment. The project was supported by the Royal Geographic Society (Marsden Fast Start, PI O. Marsh) and by the Past Antarctic Science and Future Implications Program (PACaFI). D. Price, M. Ryan, D. Floricioiu and E. Scheffler contributed to fieldwork. TerraSAR-X data were provided through DLR project HYD1421. Landsat-8 images courtesy of the US Geological Survey. AWS data from Scott Base were provided by NIWA, and for Marilyn station by AMRC, SSEC, UW-Madison. The authors enjoyed fruitful discussions with H. Purdie and M. Sellier. CW also thanks R. Mueller for sharing oceanographic expertise. The thorough comments of one anonymous reviewer and L. Padman's valuable insight into tidal dynamics largely improved the paper. We also thank A. Robinson for editing.
\end{acknowledgements}

%% REFERENCES

%% The reference list is compiled as follows:

%\begin{thebibliography}{}

%\bibitem[AUTHOR(YEAR)]{LABEL1}
%REFERENCE 1

%\bibitem[AUTHOR(YEAR)]{LABEL2}
%REFERENCE 2

%\end{thebibliography}

%% Since the Copernicus LaTeX package includes the BibTeX style file copernicus.bst,
%% authors experienced with BibTeX only have to include the following two lines:
%%
\bibliographystyle{copernicus}
\bibliography{./JabRef/WC_database}

```
%%
%% URLs and DOIs can be entered in your BibTeX file as:
%%
%% URL = {http://www.xyz.org/~jones/idx_g.htm}
%% DOI = {10.5194/xyz}

%% LITERATURE CITATIONS
%%
%% command                        & example result
%% \citet{jones90}|               & Jones et al. (1990)
%% \citep{jones90}|               & (Jones et al., 1990)
%% \citep{jones90,jones93}|       & (Jones et al., 1990, 1993)
%% \citep[p.~32]{jones90}|        & (Jones et al., 1990, p.~32)
%% \citep[e.g.,][]{jones90}|      & (e.g., Jones et al., 1990)
%% \citep[e.g.,][p.~32]{jones90}| & (e.g., Jones et al., 1990, p.~32)
%% \citeauthor{jones90}|          & Jones et al.
%% \citeyear{jones90}|            & 1990

\clearpage

\begin{figure}[ht!]
\centering{\includegraphics[width=\textwidth]{alpha_maps_v2.png}}
\caption{$\alpha$-maps of percentage vertical displacement due to ocean
tides. Red colors highlight areas that can be assumed to be
\change{freely-floating}{freely floating}. The white crosses show the
tide-model locations that also serve as a common reference point across
the images. The solid black line is the location of the profiles shown in
Figures \ref{fig:ls_process_DInSAR_SMIS} and
\ref{fig:ls_process_SAR_SMIS} on the Southern McMurdo IceShelf (left).
The dashed black line shows the location of the profiles along the Darwin
Glacier's centerline shown in Fig. \ref{fig:LS_flexure_curves_Darwin}
(right). The GPS station 'Shirase' and and 'Hillary' in the tidal-flexure
zone. White contours delineate areas of constant vertical displacement.
The map background is contrast-stretched Landsat 8 panchromatic imagery.
The geographic projection is Antarctic Polar Stereographic with easting
and northing coordinates shown in kilometers.}
\label{fig:alpha_maps}
\end{figure}
\clearpage

\begin{figure}[ht!]
\centering{\includegraphics[width=86mm]{ls_process_DInSAR_v2.png}}
\caption{Vertical displacements along a profile through the grounding
zone of the Southern McMurdo Ice Shelf, as (top) measured with 9 DInSAR
interferograms, (center) predicted from an empirical displacement model
($\alpha$-map) and (bottom) their difference.}
\label{fig:ls_process_DInSAR_SMIS}
\end{figure}
\clearpage

\begin{figure}[ht!]
\centering{\includegraphics[width=86mm]{ls_process_SAR_v2.png}}
```

```latex
\caption{Reconstruction of vertical displacement along the profile during
the 12 times of satellite overpasses on the Southern McMurdo Ice Shelf.
(Top) a combination of an empirical displacement model with adjusted CATS
tide-model output, (center) their least-square adjustment and (bottom)
the final vertical displacement profiles during the times of SAR data
acquisition.}
\label{fig:ls_process_SAR_SMIS}
\end{figure}
\clearpage

\begin{figure}[ht!]
\centering{\includegraphics[width=\textwidth]{forcing_v2.png}}
\caption{The tidal oscillation at the Darwin Glacier as predicted by four
tide models and a harmonic analysis of GPS data from the freely-floating
area. The tide-model outputs are adjusted to match DInSAR observations
using a least-squares fitting technique published in
\cite{wild2018unraveling}. Black vertical lines coincide with times of
SAR data acquisitions. Values for the prevailing tidal amplitudes and
their adjustment at these times are given in Table \ref{tab:SAR_table}.
Gray shaded areas delineate the duration of the TIDEx campaign, when GPS
data was acquired for validation (Figs. \ref{fig:Hillary_validation} and
\ref{fig:GPS_validation}).}
\label{fig:forcing_Darwin}
\end{figure}
\clearpage

\begin{table*}[ht!]
\centering
\caption{SAR imagery used for the Darwin Glacier, least-squares
adjustment (\change{$\theta$}{$\Delta A$} in m) for 5 tide models, tidal
amplitude ($A$ in m) as predicted with the TPXO7.2 tide model and rate of
tidal change ($\dot{A}$ in m~h$^{-1}$) as predicted with the
Ross\_Inv\_2002 tide model.}
\label{tab:SAR_table}
\begin{tabular}{lc|rrrrr|rr}\toprule
SAR: & Date 13:57 (UTC) & $\Delta A_{\text{CATS}}$ & $\Delta
A_{\text{RossInv}}$ & $\Delta A_{\text{Ross9cm}}$ & $\Delta
A_{\text{TPXO7.2}}$ & $\Delta A_{\text{t\_tide}}$ & $A_{\text{TPXO7.2}}$
& $\dot{A}_{\text{RossInv}}$ \\\midrule
1           & 25/05/16      & 0.109          & 0.098              &
0.133          & 0.101              & -0.004 & -0.341  & -0.059        \\
2           & 05/06/16      & -0.061         & -0.066             & -
0.097          & -0.039             & -0.030 & -0.666  & -0.057        \\
3           & 16/06/16      & 0.029          & 0.035              & -
0.050          & 0.034              & 0.013  & -0.409  & -0.007        \\
4           & 27/06/16      & 0.032          & -0.012             & -
0.049          & -0.009             & -0.111 & 0.002   & -0.022        \\
5           & 08/07/16      & 0.054          & -0.022             &
0.107          & 0.008              & 0.122  & -0.271  & -0.037        \\
6           & 19/07/16      & -0.086         & -0.082             &
0.035          & -0.088             & 0.206  & -0.661  & -0.005        \\
7           & 30/07/16      & -0.091         & 0.015              & -
0.026          & -0.045             & 0.074  & -0.687  & 0.080         \\
```

```
8               & 10/08/16        & -0.078         & 0.001           & -
0.035           & -0.040          & -0.080 & -0.276  & 0.072      \\
9               & 26/10/16        & -0.073         & -0.023          & -
0.053           & -0.046          & -0.244 & -0.132  & 0.011      \\
10              & 06/11/16        & -0.044         & -0.009          & -
0.088           & -0.023          & -0.172 & 0.087   & 0.096      \\
11              & 17/11/16        & 0.099          & 0.025           & -
0.002           & 0.084           & 0.059  & 0.522   & 0.052      \\
12              & 28/11/16        & 0.109          & 0.041           &
0.124            & 0.062           & 0.168  & 0.398   & -0.029
\\\midrule
\multicolumn{2}{r|}{mean absolute $\Delta A$} & 0.072 & 0.036 & 0.067 &
0.048 & 0.107 & - & - \\\bottomrule
\end{tabular}
\end{table*}

\clearpage

\begin{table*}[ht!]
\centering
\caption{DInSAR images of the Darwin Glacier. The SAR combination from 12
available SAR images, the tidal amplitude ($A$ in m) as measured at the
Shirase location as well as predicted with the TPXO7.2 tide model.}
\label{tab:DInSAR_table}
%\begin{adjustbox}{totalheight=\textheight-2\baselineskip}
\begin{minipage}[b]{0.45\linewidth}\centering
 \begin{tabular}{lr|rr|r}\toprule
ID & SAR combination & $A_{\text{TerraSAR-X}}$ & $A_{\text{TPXO7.2}}$ &
$\Delta$    \\\midrule
1  & (1-2)-(2-3)     & 0.581     & 0.582     & -0.001 \\
2  & (1-2)-(3-4)     & 0.740     & 0.735     & 0.004  \\
3  & (1-2)-(4-5)     & 0.057     & 0.052     & 0.005  \\
4  & (1-2)-(5-6)     & -0.061    & -0.065    & 0.004  \\
5  & (1-2)-(6-7)     & 0.298     & 0.299     & -0.001 \\
6  & (1-2)-(7-8)     & 0.734     & 0.736     & -0.002 \\
7  & (1-2)-(9-10)    & 0.552     & 0.544     & 0.008  \\
8  & (1-2)-(10-11)   & 0.736     & 0.760     & -0.024 \\
9  & (1-2)-(11-12)   & 0.207     & 0.201     & 0.006  \\
10 & (2-3)-(3-4)     & 0.154     & 0.154     & 0.001  \\
11 & (2-3)-(4-5)     & -0.529    & -0.530    & 0.001  \\
12 & (2-3)-(5-6)     & -0.646    & -0.647    & 0.001  \\
13 & (2-3)-(6-7)     & -0.288    & -0.283    & -0.005 \\
14 & (2-3)-(7-8)     & 0.137     & 0.154     & -0.017 \\
15 & (2-3)-(9-10)    & -0.034    & -0.038    & 0.004  \\
16 & (2-3)-(10-11)   & 0.190     & 0.178     & 0.012  \\
17 & (2-3)-(10-11)   & -0.376    & -0.381    & 0.004  \\
18 & (3-4)-(4-5)     & -0.688    & -0.683    & -0.004 \\
19 & (3-4)-(5-6)     & -0.805    & -0.801    & -0.005 \\
20 & (3-4)-(6-7)     & -0.446    & -0.436    & -0.009 \\
21 & (3-4)-(7-8)     & -0.014    & 0.001     & -0.015 \\
22 & (3-4)-(9-10)    & -0.192    & -0.192    & -0.000 \\
23 & (3-4)-(10-11)   & 0.055     & 0.025     & 0.030  \\
\vdots & \vdots & \vdots & \vdots & \vdots \\\bottomrule
\end{tabular}
```

```latex
\end{minipage}
\hspace{0.5cm}
\begin{minipage}[b]{0.45\linewidth}
%\end{minipage} \hfill
%\begin{minipage}{0.5\textwidth}
\begin{tabular}{lr|rr|r}\toprule
ID & SAR combination & $A_{\text{TerraSAR-X}}$ & $A_{\text{TPXO7.2}}$ &
$\Delta$    \\\midrule
\vdots & \vdots & \vdots & \vdots & \vdots \\
24 & (3-4)-(11-12)  & -0.536    & -0.534    & -0.002 \\
25 & (4-5)-(5-6)    & -0.112    & -0.117    & 0.005  \\
26 & (4-5)-(6-7)    & 0.246     & 0.247     & -0.001 \\
27 & (4-5)-(7-8)    & 0.679     & 0.684     & -0.005 \\
28 & (4-5)-(9-10)   & 0.496     & 0.492     & 0.004  \\
29 & (4-5)-(10-11)  & 0.690     & 0.708     & -0.018 \\
30 & (4-5)-(11-12)  & 0.155     & 0.149     & 0.006  \\
31 & (5-6)-(6-7)    & 0.363     & 0.364     & -0.001 \\
32 & (5-6)-(7-8)    & 0.811     & 0.801     & 0.010  \\
33 & (5-6)-(9-10)   & 0.620     & 0.609     & 0.011  \\
34 & (5-6)-(10-11)  & 0.803     & 0.825     & -0.023 \\
35 & (5-6)-(11-12)  & 0.274     & 0.266     & 0.008  \\
36 & (6-7)-(7-8)    & 0.430     & 0.437     & -0.007 \\
37 & (6-7)-(9-10)   & 0.237     & 0.244     & -0.007 \\
38 & (6-7)-(10-11)  & 0.466     & 0.461     & 0.005  \\
39 & (6-7)-(11-12)  & -0.107    & -0.098    & -0.009 \\
40 & (7-8)-(9-10)   & -0.202    & -0.192    & -0.009 \\
41 & (7-8)-(10-11)  & 0.025     & 0.024     & 0.000  \\
42 & (7-8)-(11-12)  & -0.541    & -0.535    & -0.006 \\
43 & (9-10)-(10-11) & 0.228     & 0.217     & 0.011  \\
44 & (9-10)-(11-12) & -0.343    & -0.342    & -0.001 \\
45 & (10-11)-(11-12) & -0.565   & -0.559    & -0.006 \\\hline
\multicolumn{4}{r|}{mean absolute error}& 0.007 \\\bottomrule
\end{tabular}
\end{minipage}
%\end{minipage}
%\end{adjustbox}
\end{table*}

\clearpage

\begin{figure}[ht!]
\centering{\includegraphics[width=0.75\textwidth]{Darwin_DInSARs.png}}
\caption{Selection of three measured and modelled images from 45
available DInSAR combinations. The top panels show conditions at a
relatively large double-differential tidal amplitude (ID 37), the center
panels display a pronounced visoelastic signal in the Darwin
\change{Glacier's}{Glaciers} grounding zone (ID 39) and the bottom panels
show a complex flexural pattern (ID 15).}
\label{fig:Darwin_DInSARs}
\end{figure}
\clearpage

\begin{figure}[ht!]
```

[revised manuscript text omitted]

\begin{table*}[ht!]
\centering
\caption{Amplitude and phase of the K1 tidal constituents from harmonic
analysis of tiltmeter measurements and values of the rheological
parameters to minimize the average RMSE. Amplitudes are given in degrees,
phases are the phase lag of the K1 constituent with respect to the
equilibrium tide on Greenwich longitude.}
\label{tab:K1_table}
\begin{tabular}{lr|lcl|lcr}\toprule
                          &  & &K1 amplitude $\pm$ error ($^{\circ}$)&
&          & K1 phase $\pm$ error ($^{\circ}$) &            \\\midrule
&tiltmeter 1                            & -0.001 & 0.0033  & +0.001 & -
24.34     & 206.22          & +24.34     \\
&tiltmeter 2                            & -0.001 & 0.0044  & +0.001 & -
8.02      & 215.35          & +8.02     \\
&tiltmeter 3                            & -0.001 & 0.0044  & +0.001 & -
7.46      & 218.64          & +7.46      \\
&tiltmeter 4                            & -0.002 & 0.0065  & +0.002 & -
13.63     & 219.34          & +13.63     \\
&tiltmeter 5                            & -0.001 & 0.0055  & +0.001 & -
12.36     & 198.09          & +12.36     \\
&tiltmeter 6                            & -0.001 & 0.0014  & +0.001 & -
14.95     & 207.34          & +14.95     \\
&tiltmeter 7                            & -0.001 & 0.0025  & +0.001 & -
18.06     & 181.97          & +18.06
\\\midrule
elastic & \begin{tabular}{@{}r@{}} best $E$ (GPa) \\  average RMSE
($^{\circ}$) \end{tabular} & \begin{tabular}{@{}l@{}}1.5 \\ 0.00098
\end{tabular} & \begin{tabular}{@{}c@{}}1.0 \\ 0.00147 \end{tabular} &
\begin{tabular}{@{}r@{}}0.5 \\ 0.00198 \end{tabular} &
\begin{tabular}{@{}l@{}}0.5 \\ 0.00127 \end{tabular} &
\begin{tabular}{@{}c@{}}1.0 \\ 0.00147 \end{tabular} &
\begin{tabular}{@{}r@{}}2.0 \\ 0.00182 \end{tabular}\\\midrule
&best $E$ (GPa)                & 1.5      & 1.0 & 1.0
& 0.5      & 1.0                      & 2.0          \\
viscoelastic &best $\nu$ (TPa s)              & 19.9    & 10.0 & 10.0
& 12.6      & 10.0                      & 7.9         \\
&average RMSE ($^{\circ}$)    & 0.00077        & 0.00119 & 0.00170
& 0.00122 & 0.00119                 & 0.00128 \\\bottomrule
\end{tabular}
\end{table*}
```

```
\appendixfigures  %% needs to be added in front of appendix figures
\begin{figure}[ht!]
\centering{\includegraphics[width=172mm]{GPS_validation_v2.png}}
\caption{Validation of the tidal predictions of 5 tide models with a GPS
record from the freely-floating 'Shirase' station. The tide-model outputs
are adjusted to match DInSAR observations using a least-squares fitting
technique published in \cite{wild2018unraveling}. Root-mean-square-errors
before and after this adjustment are presented in Tab.
\ref{tab:GPS_table}. The length of the records corresponds to the gray-
shaded area in Fig. \ref{fig:forcing_Darwin}.}
\label{fig:GPS_validation}
\end{figure}

\appendixtables   %% needs to be added in front of appendix tables
\begin{table}[ht!]
\centering
\caption{Root-mean-square-errors in m between tide-model output and GPS
data from 'Shirase' before and after the adjustment to match DInSAR.}
\label{tab:GPS_table}
\begin{tabular}{lrr}\toprule
Tide-model:           & RMSE before:     & RMSE after:  \\\midrule
CATS2008a\_opt          & 0.117 & 0.087               \\
Ross\_Inv\_2002        & 0.112 & 0.091             \\
Ross\_VMADCP\_9cm      & 0.135 & 0.127             \\
TPXO7.2                & 0.108 & 0.067                 \\\midrule
mean & 0.118 & 0.093 \\\midrule
t\_tide              & 0.127 & 0.141 \\\bottomrule
\end{tabular}
\end{table}

%% FIGURES

%% When figures and tables are placed at the end of the MS (article in
one-column style), please add \clearpage
%% between bibliography and first table and/or figure as well as between
each table and/or figure.

%% ONE-COLUMN FIGURES

%%f

%
%%% TWO-COLUMN FIGURES
%
%%f

%
%
%%% TABLES
%%%
```

```
%%% The different columns must be seperated with a & command and should
%%% end with \\ to identify the column brake.
%
%%% ONE-COLUMN TABLE
%
%%t
%\begin{table}[t]
%\caption{TEXT}
%\begin{tabular}{column = lcr}
%\tophline
%
%\middlehline
%
%\bottomhline
%\end{tabular}
%\belowtable{} % Table Footnotes
%\end{table}
%
%%% TWO-COLUMN TABLE
%
%%t
%\begin{table*}[t]
%\caption{TEXT}
%\begin{tabular}{column = lcr}
%\tophline
%
%\middlehline
%
%\bottomhline
%\end{tabular}
%\belowtable{} % Table Footnotes
%\end{table*}
%
%%% LANDSCAPE TABLE
%
%%t
%\begin{sidewaystable*}[t]
%\caption{TEXT}
%\begin{tabular}{column = lcr}
%\tophline
%
%\middlehline
%
%\bottomhline
%\end{tabular}
%\belowtable{} % Table Footnotes
%\end{sidewaystable*}
%
%
%%% MATHEMATICAL EXPRESSIONS
%
%%% All papers typeset by Copernicus Publications follow the math
typesetting regulations
```

```
%%% given by the IUPAC Green Book (IUPAC: Quantities, Units and Symbols
in Physical Chemistry,
%%% 2nd Edn., Blackwell Science, available at:
http://old.iupac.org/publications/books/gbook/green_book_2ed.pdf, 1993).
%%%
%%% Physical quantities/variables are typeset in italic font (t for time,
T for Temperature)
%%% Indices which are not defined are typeset in italic font (x, y, z, a,
b, c)
%%% Items/objects which are defined are typeset in roman font (Car A, Car
B)
%%% Descriptions/specifications which are defined by itself are typeset
in roman font (abs, rel, ref, tot, net, ice)
%%% Abbreviations from 2 letters are typeset in roman font (RH, LAI)
%%% Vectors are identified in bold italic font using \vec{x}
%%% Matrices are identified in bold roman font
%%% Multiplication signs are typeset using the LaTeX commands \times (for
vector products, grids, and exponential notations) or \cdot
%%% The character * should not be applied as mutliplication sign
%
%
%%% EQUATIONS
%
%%% Single-row equation
%
%\begin{equation}
%
%\end{equation}
%
%%% Multiline equation
%
%\begin{align}
%& 3 + 5 = 8\\
%& 3 + 5 = 8\\
%& 3 + 5 = 8
%\end{align}
%
%
%%% MATRICES
%
%\begin{matrix}
%x & y & z\\
%x & y & z\\
%x & y & z\\
%\end{matrix}
%
%
%%% ALGORITHM
%
%\begin{algorithm}
%\caption{...}
%\label{a1}
%\begin{algorithmic}
%...
```

```
%\end{algorithmic}
%\end{algorithm}
%
%
%%% CHEMICAL FORMULAS AND REACTIONS
%
%%% For formulas embedded in the text, please use \chem{}
%
%%% The reaction environment creates labels including the letter R, i.e.
(R1), (R2), etc.
%
%\begin{reaction}
%%% \rightarrow should be used for normal (one-way) chemical reactions
%%% \rightleftharpoons should be used for equilibria
%%% \leftrightarrow should be used for resonance structures
%\end{reaction}
%
%
%%% PHYSICAL UNITS
%%%
%%% Please use \unit{} and apply the exponential notation

\end{document}
```